# MIND THE GAP: ALIGNING THE BRAIN WITH LANGUAGE MODELS REQUIRES A NONLINEAR AND MULTIMODAL APPROACH

## ABSTRACT

Speech comprehension involves complex, nonlinear, and multimodal processes within the brain, integrating auditory signals with linguistic and semantic information across widespread brain networks. Traditional brain encoding models, often relying on linear mappings from unimodal features, fall short in representing these intricate mechanisms. In this study, we introduce a nonlinear, multimodal encoding model that combines audio and linguistic features extracted from pretrained deep learning models (e.g., LLAMA and Whisper). These nonlinear architectures and early fusion mechanisms significantly enhance cross-modal integration, achieving a 14.4% increase in average normalized correlation coefficient and 7.7% increase in average single-story correlation compared to the previous state-of-the-art model relying on weighted averaging of linear unimodal predictions. Moreover, this improved performance reveals novel insights into the brain's functional organization, demonstrating how auditory and semantic information are nonlinearly fused within regions linked to motor control, somatosensory processing, and higher-level semantic representation. Our findings provide empirical support for foundational neurolinguistic theories, including the Motor Theory of Speech Perception, embodied semantic memory, and the Convergence Zone model, revealing novel insights into neural mechanisms otherwise impossible with simpler encoder models. By emphasizing the critical role of nonlinearity and multimodality in brain encoding models, our work bridges neural mechanisms and computational modeling, paving the way for the development of more biologically inspired, brain-aligned artificial intelligence systems.

## 1 INTRODUCTION

The human brain effortlessly deciphers the complex symphony of spoken language, seamlessly integrating auditory signals with linguistic and semantic meaning. This remarkable feat of speech comprehension involves a dynamic interplay of multiple brain regions. The brain relies heavily on the integration of information from different modalities (McGettigan et al., 2012; Calvert, 2001; Ghazanfar & Schroeder, 2006). Furthermore, this process is inherently nonlinear, with complex transformations and interactions occurring across various levels of neural representation (Tuller et al., 2011). Understanding these intricate neural mechanisms is not only crucial for advancing our knowledge of human cognition but also holds immense potential for developing more human-like artificial intelligence and creating more effective therapies for individuals with language disorders.

Language encoding models, which predict brain activity from speech stimuli, provide a powerful tool for unraveling the neural processes underlying speech comprehension (Naselaris et al., 2011; Tang et al., 2023; Vaidya et al., 2022; de Heer et al., 2017; LeBel et al., 2021; Jain & Huth, 2018; Toneva & Wehbe, 2019; Goldstein et al., 2022; Schrimpf et al., 2021). Unlike traditional contrast-based paradigms that rely on carefully controlled experiments, encoding models inherently capitalize on naturalistic stimuli, such as spoken language, to better capture the brain's real-world processing. This allows for a more comprehensive understanding of the brain's activity in response to complex and ecologically valid tasks, offering greater generalizability compared to simplified, contrast-based tasks (Jain et al., 2024). Moreover, they are increasingly being utilized for in-silico experiments: computational simulations of neural activity that enable researchers to test hypotheses

about brain function without the need for new data collection (Jain et al., 2024; Yamins & DiCarlo, 2016; Ratan Murty et al., 2021; Bashivan et al., 2019; Wehbe et al., 2016).

Initially, encoding models mapped simple acoustic features (e.g., spectograms) onto brain activity (de Heer et al., 2017). The introduction of word embeddings (Mikolov, 2013) enabled incorporation of semantic information, revealing how meaning is represented across the brain (Huth et al., 2016). Building on these foundations, recent advances in large language models (LLMs) and sophisticated audio models have enabled even richer and more context-sensitive representations of speech. These innovations have led to substantial improvements in prediction accuracy, as demonstrated by studies utilizing these advanced feature extractors (Antonello et al., 2024; Vaidya et al., 2022).

Despite significant progress, most prior work has been limited to unimodal features and linear mappings, which fail to address two critical aspects of neural processing. First, the brain relies on nonlinear computations (Friston et al., 2000; Deco et al., 2008; Beniaguev et al., 2021; Tuller et al., 2011), enabling complex interactions across neural systems. Second, speech comprehension is a multimodal process, requiring the integration of diverse information sources (voice, gesture, linguistic) across distributed neural networks (McGettigan et al., 2012). These principles, along with the Motor Theory of Speech Perception (Liberman et al., 1967) and the Convergence Divergence Zone model (Damasio, 1989), highlight the importance of encoding models capable of capturing both nonlinear dynamics and multimodal integration to reveal the brain's functional organization. Therefore, addressing these limitations is critical for building encoding models that more accurately reflect the brain's complexity and provide deeper insights into its multimodal and nonlinear processes.

While some studies explored multimodal models that combine linguistic features with visual information (Oota et al., 2022; Wang et al., 2022; Scotti et al., 2024), the integration of auditory features into language encoding remains largely unexplored. This gap in research is particularly significant, as auditory features are central to speech comprehension. Recent work by Oota et al. (2023) demonstrates that speech models, unlike text-based language models, capture brain activity patterns in auditory regions that cannot be explained by low-level stimulus features, underscoring the complementary nature of auditory and linguistic representations. Investigating how auditory and linguistic information interact in the brain could provide key insights into neural mechanisms underlying speech processing and advance the development of more brain aligned encoding models.

In this study, we address these limitations by introducing a nonlinear and multimodal encoding models that leverage both audio and linguistic features extracted from advanced deep learning models like LLAMA and Whisper. Our work makes the following key contributions:

- **We demonstrate a substantial improvement in prediction accuracy. Compared to the state-of-the-art stacked linear regression model, we achieved a 14.4% increase in mean normalized correlation coefficient (Appendix A.3). We also saw a 17.2% increase in mean correlation across the cortex compared to widely used linear encoding models (Antonello et al., 2024).** This significant performance boost highlights the importance of incorporating nonlinearity and multimodality together for capturing the full capacity of the brain's language processing mechanisms, showing that these complex interactions are crucial for accurately modeling brain activity during speech comprehension.
- **We provide novel evidence for nonlinear multimodal fusion in motor, sensory, and visual brain regions, supporting existing theories in neurolinguistics while revealing new insights into the neural basis of speech comprehension.** Our findings demonstrate enhanced prediction accuracy in motor regions when incorporating both auditory and semantic features, supporting the long-lasting Motor Theory of Speech Perception (Liberman et al., 1967; 1952; Poeppel & Assaneo, 2020), suggesting the involvement of motor regions during speech comprehension in simulating articulatory movements. Also, we observe improved predictions in visual areas bordering the visual cortex when adding semantic information, supporting the Convergence Zone model (Damasio, 1989; Damasio et al., 1996; 2004) and providing evidence for a distributed, multimodal representation of semantic concepts. Our results reveal that incorporating semantic features enhances predictions in somatosensory regions, suggesting a broader involvement of these areas in processing language, consistent with the concept of embodied semantics (Davis & Yee, 2021).
- **We establish a foundation for future research by demonstrating the potential of deep learning techniques for modeling the complexity of speech comprehension in the brain.** Our work paves the way for developing more sophisticated encoding models that in-

Table 1: Performance of encoding models across different modalities and architectures. This table presents the average voxelwise $r^2$ and normalized correlation coefficient ($CC_{norm}$) values for various encoding models, comparing their ability to predict fMRI responses across different input modalities *semantic, audio* or *multimodal* and encoder architectures *Linear, MLLinear, DIMLP* and *MLP*. Notably, MLP encoders consistently outperform linear models and their variants, highlighting the importance of incorporating nonlinearity for accurate fMRI prediction. $r^2$ is computed as $|r| * r$.

| modality 1 | modality 2 | encoder | response | Avg $r^2$ | Avg $CC_{norm}$ |
|---|---|---|---|---|---|
| semantic | audio | MLP | PCA | **4.29% (+17.2%)** | **34.32% (+17.9%)** |
| semantic | audio | DIMLP | PCA | 4.18% (+14.2%) | 32.59% (+11.9%) |
| semantic | audio | MLLinear | PCA | 4.10% (+12.0%) | 32.41% (+11.3%) |
| semantic | audio | Linear | all voxels | 4.10% (+12.0%) | 31.36% (+7.7%) |
| semantic | audio | Linear | PCA | 3.87% (+5.7%) | 28.92% (-0.7%) |
| semantic | audio | MLP | all voxels | 3.83% (+4.6%) | 31.11% (+6.8%) |
| semantic | - | MLP | PCA | 3.79% (+3.6%) | 30.89% (+6.1%) |
| semantic | - | MLLinear | PCA | 3.67% (+0.3%) | 29.95% (+2.8%) |
| semantic | - | Linear | all voxels | 3.66% (Baseline) | 29.12% (Baseline) |
| semantic | - | Linear | PCA | 3.56% (-2.7%) | 26.88% (-7.7%) |
| semantic | - | MLP | all voxels | 3.36% (-8.2%) | 27.45% (-5.7%) |
| audio | - | MLP | PCA | 3.01% (-17.8%) | 29.01% (-0.4%) |
| audio | - | MLP | all voxels | 2.89% (-21.0%) | 28.21% (-3.1%) |
| audio | - | MLLinear | PCA | 2.89% (-21.0%) | 27.50% (-5.6%) |
| audio | - | Linear | PCA | 2.81% (-23.2%) | 26.71% (-8.3%) |
| audio | - | Linear | all voxels | 2.77% (-24.3%) | 25.20% (-13.5%) |

corporate mechanisms like attention, memory, and brain-inspired computations, ultimately leading to a deeper understanding of how the brain creates meaning from language.

## 2 METHOD

### 2.1 MRI DATA

We used a publicly available fMRI dataset (LeBel et al., 2023; Tang et al., 2023) of three subjects listening to approximately 20 hours of English language podcasts. The training data comprised 95 stories across 20 scanning sessions (approximately 33,000 time points). For testing, we used three held-out stories: one averaged across ten repetitions and two averaged across five repetitions each, with no session containing repeated test stimuli. Each voxel was normalized to zero mean and unit variance across time, ensuring consistent training and testing data (Antonello et al., 2024).

### 2.2 FEATURE EXTRACTION

We extracted the features from the stimuli by taking the intermediate hidden layer representations of various large language models (LLMs) and audio models exposed to the same stimuli as the subject. For semantic feature extraction, we utilized LLAMA-1 (Touvron et al., 2023a) models with 7B, 13B, 33B, and 65B parameters, LLAMA-2 7B (Touvron et al., 2023b), and LLAMA-3 8B (Dubey et al., 2024). For audio feature extraction, we employed Whisper (Radford et al., 2023) models, including Tiny, Base, Small, Large, and Large v2 and v3. All models were obtained from Hugging Face (Wolf, 2019) and computed using half-precision (float16) for efficiency.

For LLAMA models, the stimuli were presented using a dynamically sized context window strategy (Antonello et al., 2024) to balance computational efficiency and contextual coherence. Initially, the context window grew incrementally as tokens were added, up to a maximum of 512 tokens, after which the window was reset to a new context of 256 tokens. This approach avoided memory overheads associated with processing the entire tokenized text while maintaining sufficient contextual information for accurate semantic representation. For Whisper models, the audio stimuli (waveform) were processed using a sliding-window approach with a fixed window size of 16 seconds and a stride of 0.1 seconds. Features were extracted exclusively from the encoder portion of the encoder-decoder

architecture, as it processes only the raw audio waveform input. This ensured that the extracted features accurately captured audio-specific representations relevant to the stimuli. Refer to Antonello et al. (2024) for further details on the process and the handling of model contexts.

The following process adheres to Antonello et al. (2024) for fair comparison. We temporally aligned the hidden states from the $l^{\text{th}}$ layer of the language or audio models with fMRI acquisition times using Lanczos interpolation. To account for temporal delays between stimulus presentation and neural responses, we concatenated representations from the four preceding timepoints (2, 4, 6, and 8 seconds prior) for each fMRI acquisition timepoint, yielding a feature vector for each TR (see Appendix A.2.1). Unless stated otherwise, we extracted semantic features from the 12th layer of LLAMA-7B and audio features from the final encoder layer of Whisper Large V1. The layers were selected based on our findings that performance scaling with increasing LLM size, as reported in Antonello et al. (2024), does not hold for LLAMA models of size $\geq$7B (see Appendix A.5).

### 2.3 REPRESENTATIONS FOR FMRI DATA

We predicted PCA-reduced fMRI representations, rather than the full voxel space, motivated by three benefits. First, PCA is a common dimensionality reduction method in fMRI analysis that helps prevent overfitting and has been widely applied in neuroimaging studies (Wang et al., 2010; Mourao-Miranda et al., 2005; López et al., 2011; Koutsouleris et al., 2009). This was crucial for our study, as speech comprehension engages the whole cortex and hence a vast number of voxels $80 - 90$k (LeBel et al., 2023), far more than in tasks like vision encoding $\approx 15$k (Allen et al., 2022). In fact, linearly mapping the semantic stimulus ($4 \times 4096$) to subject S1 require 1.3 billion parameters, whereas utilizing PCA (512 components) reduced this to 8.4 million, preventing overfitting. Second, PCA is effective at untangling the redundancy in brain data, as fMRI voxels are highly correlated. Studies such as Jabakhanji et al. (2022); Lin et al. (2022) have shown that masking up to 90% of voxels does not significantly impact fMRI decoding or reconstruction performance, suggesting that information is distributed and redundant. Lin et al. (2022) further showed that when training a three-layer autoencoder on fMRI data, 99% of the variance in latent features and reconstructed voxels could be attributed to their top 100 PCA components, further highlighting the brain's redundancy. Lastly, PCA allows for reverse transformation to recover the original voxel space from predicted PCA embeddings, maintaining the interpretability of the model's predictions.

In detail, we applied Principal Component Analysis (PCA) to the aggregate fMRI response matrix $Y_{\text{org}} \in \mathbb{R}^{N_{\text{TR}} \times N_{\text{voxels}}}$, reducing its dimensionality to $Y_{\text{PCA}} \in \mathbb{R}^{N_{\text{TR}} \times N_{\text{PCA}}}$. $N_{\text{TR}}$, $N_{\text{voxels}}$ each refers to the number of time points (TRs) and voxels respectively, and $N_{\text{PCA}}$ was set to 512. The encoding model was trained to predict these PCA-reduced representations, and during evaluation, the predicted $\hat{Y}_{\text{PCA}}^{\text{test}}$ was reconstructed back to the original voxel space using inverse projection. This reconstructed output was then compared to the ground truth fMRI responses, $Y^{\text{test}} \in \mathbb{R}^{N_{\text{TR-test}} \times N_{\text{voxels}}}$. More implementation details are provided in Appendix A.2.2.

### 2.4 ENCODING MODEL

Previous research primarily employed linear regression to predict voxel responses from unimodal features (audio or semantic) (Tang et al., 2023; Huth et al., 2016; de Heer et al., 2017; LeBel et al., 2021; Jain & Huth, 2018; Toneva & Wehbe, 2019; Goldstein et al., 2022; Schrimpf et al., 2021). Our study expands upon this approach by systematically investigating a range of encoding models varying in complexity and input modality to capture more nuanced relationships between stimulus representations and brain responses. We explored combinations of different stimulus representations, encoding model architectures, and response representations (Details are in Appendix A.2.3) :

**Stimulus representations (model input)** included audio and semantic features extracted from various versions of LLAMA and Whisper models, as explained in Section 2.2. For multimodal encoders, we concatenated audio and semantic features to capture the combined effect of both modalities.

**Encoding model architectures** assessed the impact of complexity and nonlinearity:

- *Linear Regression (Linear):* Following Antonello et al. (2024), we used ridge regression.
- *Multi-Layer Perceptron (MLP):* MLP with a single hidden layer of 256 units.

- *Multi-Layer Linear (MLLinear):* MLP but without dropout, batch normalization, and with the identity activation function. This model serves as a reduced-rank linear regression, helping to isolate the effects of dimensionality reduction from nonlinearity.
- *Delayed Interaction MLP (DIMLP):* Used for multimodal cases, this MLP variant processes each modality through separate 256-unit hidden layers before concatenation and final linear projection. This allows nonlinear processing within each modality while limiting cross-modal interaction to be linear, revealing the effects of nonlinear fusion of modalities.

**Response representations (model output)** included two types: the full voxel space and a 512-dimensional PCA-reduced representation of the voxel responses, as explained in Section 2.3.

## 2.5 Noise ceiling and normalized correlation coeffiicnet

Due to the inherent noise in fMRI data, there is a theoretical upper limit to the amount of explainable variance an ideal encoding model can achieve, known as the noise ceiling. The noise ceiling for each voxel was estimated by applying an existing method (Schoppe et al., 2016) on ten responses of the same test story. For each voxel, the maximum correlation coefficient is estimated as $CC_{max} = (\sqrt{1 + \frac{NP}{SP \times N}})^{-1}$, where $N$ is the number of repeats (10 in our case), $NP$ is the noise power or unexplainable variance, and $SP$ is the amount of variance that could be explained by an ideal predictive model. Afterwards, by dividing $CC_{abs}$, the correlation coefficient of the encoding model's prediction with the ground truth fMRI signals (estimated using the average of test responses to the same stimuli), with $CC_{max}$, we obtain $CC_{norm}$, the normalized correlation coefficient ($CC_{norm} = CC_{abs}/CC_{max}$). Due to the large number of voxels ($\approx 80,000$), random noise can cause certain voxels to have $CC_{abs} > CC_{max}$, resulting in $CC_{norm} > 1$. To prevent this, we regularized for noisy voxels by setting those with $CC_{max}$ values smaller than 0.25 to 0.25 when computing $CC_{norm}$.

## 3 Results

Incorporating nonlinearity into the encoding models improved performance across all modalities. As demonstrated in Table 1, utilizing a Multi-Layer Perceptron (MLP) resulted in a notable performance increase: 3.6% for semantic models, 8.7% for audio models, and 4.1% for multimodal models, compared to traditional linear models. Furthermore, multimodality (combining audio and semantic features) yielded a substantial 12.0% gain over unimodal models, highlighting the importance of capturing multimodal interactions. Combining both nonlinearity and multimodality yielded the greatest improvement, with a 17.2% increase in mean correlation across the cortex. These results not only confirm that the brain employs nonlinear, multimodal processing for speech comprehension but also provide a powerful tool for uncovering previously hidden neural mechanisms, as we will demonstrate in the following sections. (correlation was computed as $r^2 = |r| * r$)

### 3.1 Nonlinear encoders

#### 3.1.1 Nonlinear encoders improve prediction performance

To examine the benefits of nonlinearity across feature hierarchies, we compared MLP and linear encoders across different layers of LLAMA and Whisper models (Figure 24 (Appendix)). MLP consistently outperformed linear encoding at every layer depth for both modalities, suggesting that nonlinear processing is advantageous regardless of feature complexity and supporting the brain's use of nonlinear computations throughout the speech processing hierarchy.

#### 3.1.2 Nonlinearity, not reduced dimension, improves the encoding performance

To confirm that MLP performance gains stem from nonlinear processing rather than reduced dimensionality, we compared the MLP encoder with two linear control models: "Linear," which uses linear regression on PCA-reduced data, and "MLLinear," which mirrors the MLP architecture without nonlinear activation functions. As shown in Table 1, both Linear and MLLinear models performed similarly to or worse than linear regression on the full voxel space (baseline model), particularly for

semantic and multimodal cases. While these models slightly outperformed the baseline for audio-only data, their accuracy remained below that of the MLP encoder. These findings highlight the MLP's ability to capture nonlinear relationships drives its superior performance, not merely dimensionality reduction. Additionally, PCA proves essential for leveraging the benefits of nonlinear models. MLP models predicting all voxels directly performed poorly, likely due to overfitting from the large voxel space (80–90k voxels compared to 512 PCA components).

### 3.1.3 NONLINEARITY IMPROVES BRAIN-WIDE PREDICTIONS AND SPATIO-TEMPORAL CLUSTERING

Nonlinear models, such as MLPs, provide a crucial advantage over linear models by effectively capturing the complex and nonlinear relationships inherent in brain activity during speech comprehension. Comparative brain maps in Figure 25, 26, and 27 (Appendix) illustrate the superior performance of MLP encoders over linear encoders, with improvements in prediction accuracy distributed broadly across the cortex. The MLP model shows significant gains in regions associated with semantic and auditory processing, such as the medial prefrontal cortex (mPFC), precuneus (PrCu), and lateral temporal cortex (LTC). These gains highlight the role of nonlinear interactions in accurately modeling brain activity, particularly in areas involved in higher-order language processing. Furthermore, our hierarchical clustering and modularity analyses (Appendix A.9.4) based on the relative error difference (RED) between Whisper and LLAMA encoding models reveal that nonlinear (MLP) encoders outperform linear encoders in functionally grouping the brain regions according to their spatial and temporal response profiles. This result suggests that our nonlinear encoding model can reveal a more refined picture of the brain's functional compartmentalization.

Our results of the widespread enhancement of prediction accuracy across the cortex and of the MLP encoder's ability to effectively compartmentalize brain regions based on their spatio-temporal dynamics together point towards a fundamental characteristic of brain function: the distributed nature of speech comprehension, which involves coordinated activity across multiple (likely parallel) brain networks. This coordinated activity of the distributed brain networks is not only crucial for language processing but also plays a key role in other high-level cognitive functions (Bassett & Sporns, 2017; Medaglia et al., 2015; Da Rocha et al., 2011). Our findings suggest that nonlinear models are well-suited for capturing this complex interplay of distributed neural activity, providing a more accurate and insightful representation of the brain's functional organization compared to linear models.

## 3.2 MULTIMODAL ENCODING MODELS IMPROVE BRAIN PREDICTIONS

### 3.2.1 MULTIMODALITY IMPROVES BRAIN-WIDE PREDICTIONS

Improvements from multimodality are cortex-wide and not restricted to specific brain regions. Voxelwise plots in Figure 1 (a,b) show that adding audio features improves not only auditory areas but also regions such as the paracentral lobule (located on the medial surface between the mPFC and PrCu) and OC (Occipital Cortex). Similarly, Figure 1 (c,d) shows that the addition of semantic features leads to improvements in most cortical areas, except certain parts of the auditory cortex (AC). These patterns are more pronounced in $CC_{norm}$ visualizations (Appendix A.10.2).

This widespread improvement from multimodality is further amplified by nonlinearity. Comparing Figure 1 (b) with (a), and (d) with (c) reveals that using MLP models not only enhances performance in regions that were already well predicted by multimodal linear encoders, but also in regions not initially benefiting from the added modality, such as the LTC when adding audio features, and LTC, mPFC, and occipital regions when adding semantic features. This suggests multimodal MLP models can better exploit the additional modality through nonlinearity, which we discuss in Section 3.3.

This cortex-wide improvement differs from Antonello et al. (2024), who reported localized enhancements primarily in the AC and M1M. Methodological differences may explain this discrepancy. First, Antonello et al. (2024) utilized features from multiple Whisper layers, likely introducing redundancy, whereas our approach focuses on the final layer. Second, their encoding model, (linear) stacked regression, averaged predictions from separate audio and semantic linear encoders, limiting the interaction between modalities to a small set of weights ("attributions") for each modality. Whereas our concatenation approach, allows for more direct interactions between modalities, both in linear and nonlinear encoders (refer to Appendix A.3 for direct comparison). Our findings align with

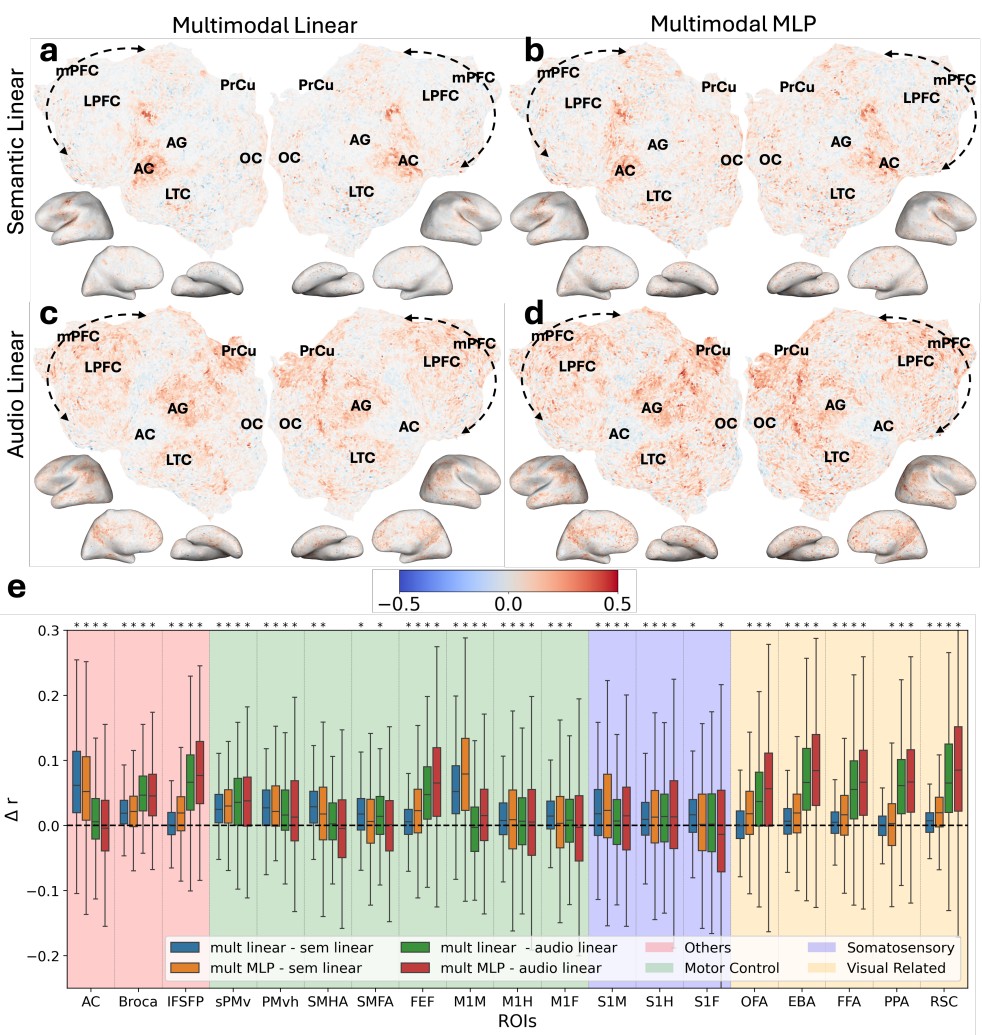

Figure 1: Multimodality improvement in encoding models. Panels (a)-(d) display voxelwise $\Delta r$ values of a single subject, where warmer colors indicate regions where the multimodal models outperform linear models. Each panel corresponds to the difference between the voxel-wise predictions of the model in the corresponding column and the model in the corresponding row. E.g., panel (a) shows the difference between the Multimodal Linear and Semantic Linear models. (e) Box plot showing $\Delta r$ across different regions of interest (ROIs), where the $\Delta r$ values are aggregated over all subjects. *multi* refers to multimodal, and *sem* refers to semantic encoders. Asterisk* indicate regions where $\Delta r > 0$ with a statistically significant difference (p < 0.05). ROIs are grouped and color-coded by their functions. (A complete list of ROI abbreviations are at Appendix A.1. Voxel-wise and ROI-wise plots for each subject are in Figure 32, 33, and 36 in the Appendix).

the convergence-divergence-zone theory (Damasio, 1989; Damasio et al., 1996; 2004) which posits a distributed cortical representation of semantic information integrated from multiple modalities. Variance partitioning analysis in Appendix A.7 further supports this.

### 3.2.2 MULTIMODAL ENCODING ENHANCES MOTOR AND SENSORY AREA PREDICTIONS

Our study revealed that incorporating multimodality significantly enhances fMRI prediction accuracy, particularly within motor and sensory regions crucial for speech comprehension. The addition of either audio or semantic features consistently improved predictions in cortical regions associated with motor control (green) and somatosensory processing (blue) (Figure 1 (e)). These improvements

vary across ROIs: some benefit from the addition of semantic features (e.g., frontal eye field (FEF)), others from audio features (e.g., primary mouth motor cortex (M1M)), and some from both.

This finding aligns with the long-standing Motor Theory of Speech Perception (Liberman et al., 1967; 1952; Poeppel & Assaneo, 2020), which posits that motor regions actively participate in simulating the articulatory movements necessary for speech production, thereby aiding comprehension. In particular, improvements seen with the addition of auditory features align with research that discovered a tight coupling between auditory and motor-sensory processing (Skipper et al., 2005; Wu et al., 2014; Wilson et al., 2004).

Furthermore, the improvement of somatosensory prediction from the addition of either modalities also suggests that semantic information plays a critical role in shaping activity within somatosensory regions. This suggests a broader involvement of these areas in speech comprehension than previously recognized. This aligns with the concept of embodiment of semantic memory, where the understanding of concepts is grounded in sensory and motor experiences and their memory in the neocortex (Binder & Desai, 2011). Our results align with Nagata et al. (2022), who showed that the sensorimotor cortex is engaged in processing both concrete and abstract word semantics.

These enhancements in motor and sensory areas are more pronounced with MLP models, underscoring the importance of nonlinear interactions between auditory and semantic information. We explore this in more details in Section 3.3. See Appendix A.10 for subject-wise plots.

### 3.2.3 FUSION OF AUDITORY AND SEMANTIC INFORMATION ACROSS BRAIN REGIONS

Our analysis reveals that fusing auditory and semantic information significantly enhances fMRI prediction accuracy, even in regions traditionally associated with high-level visual processing. Specifically, incorporating semantic features leads to improved predictions in areas situated at the border of the visual cortex, including the Occipital Face Area (OFA) (Pitcher et al., 2011), Extrastriate Body Area (EBA) (Downing et al., 2001), Fusiform Face Area (FFA) (Kanwisher et al., 1997), Posterior Parietal Area (PPA) (Epstein & Kanwisher, 1998), and Retrosplenial Cortex (RSC) (Vann et al., 2009) (Figure 1(e)). This suggests that these regions, while primarily involved in processing visual stimuli, also play a role in representing linguistic semantics.

This finding aligns with prior studies demonstrating that visual and linguistic stimuli with similar semantic content elicit similar brain responses (Tang et al., 2024; Deniz et al., 2019; Devereux et al., 2013; Fairhall & Caramazza, 2013; Popham et al., 2021; Huth et al., 2012; 2016). These studies, along with our results, support the convergence-divergence-zone theory (Popham et al., 2021; Damasio et al., 1996; 2004; Damasio, 1989), which posits semantic information from multiple modalities is integrated at points across the cortex, leading to a unified representation of semantic meaning. This model suggests that the brain constructs a modality-independent representation of semantics, drawing on information from vision, language, and other senses (Binder & Desai, 2011; Tang et al., 2023; 2024; Devereux et al., 2013; Fairhall & Caramazza, 2013; Martin, 2016). Our variance partitioning analysis in Appendix A.7 further supports this.

Crucially, our study provides novel evidence for the auditory modality's contribution to this unified semantic representation. We found that adding audio features to our multimodal models resulted in a small, but statistically significant performance increase in those visual border ROIs (yellow). This finding suggests that auditory information, such as tone of voice and environmental sounds, provides unique semantic context not fully captured by visual or linguistic features alone.

Furthermore, our analysis of Broca's area and the Inferior Frontal Sulcus Face Patch (IFSFP) reinforces the notion that these regions are involved in multimodal processing. We found that incorporating audio or semantic information significantly improved prediction accuracy in these areas, particularly when using the MLP encoder. This is consistent with the growing evidence that Broca's area and IFS play a role in integrating phonological, semantic, and motor information during speech comprehension (Gough et al., 2005; Nixon et al., 2004; Papoutsi et al., 2009; Bohland & Guenther, 2006; Papoutsi et al., 2009; Bohland & Guenther, 2006; Guenther et al., 2015).

The consistent observation that multimodal fusion, particularly with nonlinear models, enhances prediction accuracy emphasizes the brain's use of complex, nonlinear computations to combine information from different modalities for a holistic understanding of language. For a visualization of subject-wise differences in ROI prediction performance, see Figure 36 (Appendix).

### 3.3 Nonlinear interactions between modalities enhance fMRI predictions

#### 3.3.1 Nonlinear interactions between modalities are important for motor and visual areas

To investigate the role of nonlinear interactions between modalities, we developed a novel model called the Delayed Interaction MLP (DIMLP). While a standard MLP allows complex nonlinear interactions between all features, DIMLP processes each modality (audio and semantic) independently through separate nonlinear layers before a final linear fusion stage. This design enables us to directly compare the contribution of within-modality nonlinearity (captured by DIMLP) to the contribution of cross-modal nonlinear interactions (captured by the standard MLP).

We find that both DIMLP and MLP outperform linear models (Table 1). DIMLP, incorporating only within-modality nonlinearity, yields a 2.0% average r² improvement over the linear model (from 4.10% average r² to 4.18%). But the standard MLP, allowing full nonlinear interactions, achieves a further 2.6% gain (from 4.18% to 4.29%). This suggests that both types of nonlinearity contribute to prediction accuracy, but cross-modal nonlinear interactions are particularly crucial.

Voxelwise analysis (Figure 37 and 38 (in Appendix)) further supports this finding. While DIMLP improves prediction accuracy across various brain regions compared to the linear model, the transition to a standard MLP leads to further, cortex-wide enhancements. This pattern suggests that nonlinear interactions between audio and semantic features are essential for modeling the complex, distributed neural representations underlying speech comprehension (see Appendix A.12).

ROI-wise analysis (Figure 2) provides a nuanced view. While the benefits of nonlinearity vary across ROIs, multimodal MLP consistently outperforms or matches DIMLP, and often improves over linear models. Notably, motor regions (e.g., M1M, FEF) and somatosensory regions (e.g., S1M) consistently show gains with statistical significance when allowing nonlinear cross-modal interactions. This suggests that nonlinear fusion of modalities is particularly critical for these regions, supporting their role in complex multimodal processing during speech comprehension.

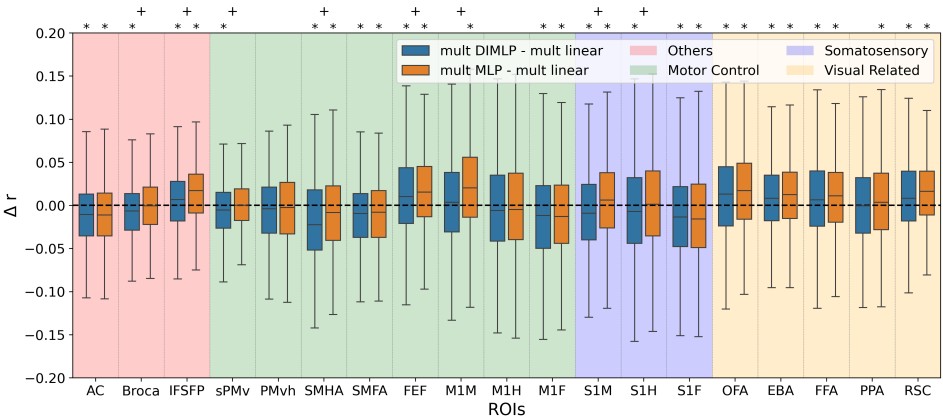

Figure 2: Box plot showing $\Delta r$ across ROIs, where the $\Delta r$ values are aggregated over all subjects. *multi* refers to multimodal, and *sem* refers to semantic encoders, and *DIMLP* refers to Delayed Interaction MLP, where only a *linear* interaction between modalities is allowed. The ROIs are color-coded by function. Regions where $\Delta r > 0$ with a p-value less than 0.05 are indicated by * symbols. Additionally, + symbols denote ROIs where there is a statistically significant difference (p-value < 0.05) between the two models based on a pairwise t-test. Voxel-wise and ROI-wise plots for each subjects can be found in Figure 37 (Appendix), and Figure 39 (Appendix), respectively.

## 4 Discussion and Conclusion

This study underscores the transformative potential of nonlinear, multimodal encoding models for advancing our understanding of speech comprehension in the brain. By introducing a nonlinear

Multi-layer Perceptron (MLP) and integrating audio and linguistic features, we achieved a 14.4% increase in mean normalized correlation across the cortex compared to previous state-of-the-art (Antonello et al., 2024), predicting 34.3% of the brain's explainable variance.

A significant finding is the enhanced prediction accuracy in motor, sensory, and semantic brain regions, demonstrating the brain's capacity for multimodal integration, in line with neurolinguistic theories. Notably, motor regions align with the Motor Theory of Speech Perception (Liberman et al., 1967; 1952; Poeppel & Assaneo, 2020), suggesting active articulatory simulation during comprehension. Performance gains in visual border areas, likely incorporating semantic information, provide evidence for the convergence-divergence-zone theory (Damasio, 1989), suggesting distributed, multimodal representation of semantic concepts. Variance partitioning analysis (Appendix A.7) supports this, showing that the majority of explained variance comes from joint audio-semantic processing rather than from either modality alone. Also, semantic feature integration improved accuracy in somatosensory regions, supporting embodied semantics (Davis & Yee, 2021). These findings show how nonlinear models effectively capture complex spatial interactions essential for multimodal language processing, revealing deeper insights into the brain's mechanisms for speech comprehension.

A key limitation in current language fMRI research is the lack of sufficiently large datasets. Our experiments revealed that increasing model complexity through additional hidden layers or sophisticated architectures like RNNs and Transformers led to overfitting (Appendix A.4). This highlights the urgent need for larger, richer language fMRI datasets to fully harness the potential of deep learning and drive further advancements. Similar to how large-scale datasets like the Natural Scenes Dataset (NSD) (Allen et al., 2022) catalyzed breakthroughs in vision encoding and decoding (Adeli et al., 2023; Yang et al., 2023; Chen et al., 2023; Scotti et al., 2024), comparable efforts in language fMRI could enable robust model training, improve generalization, and foster significant advancements in understanding the neural basis of language.

Developing inherently multimodal LLMs is also essential. Our findings suggest current multimodal LLM architectures may need rethinking. Models like LLAMA 3.1 405B (Dubey et al., 2024) adopt a text-centric approach: pretraining a text model alongside modality-specific encoders, then finetuning by freezing the text model and training only the encoders and adapters. This contrasts with our observations of brain processing in two ways. First, current models force other modalities to align with text representations, whereas our results show auditory information provides unique semantic contributions. Second, sensory inputs in these architectures are processed bottom-up, while our findings suggest rich, bidirectional interactions between modalities across multiple levels of processing. Future architectures could benefit from treating modalities as equal contributors to meaning construction, enabling bidirectional interactions throughout the processing hierarchy. Such brain-aligned multimodal LLMs may also help uncover the brain's multimodal integration. Recent research using multimodal transformers (Tang et al., 2024) support this potential, demonstrating their capacity to capture shared semantic representations across language and vision in the brain.

While nonlinear encoders offer significant benefits, they also pose challenges for precisely attributing predictive power to individual features or modalities. Although we conduct variance partitioning analysis A.7, further work is still required. Addressing this will require a multi-faceted approach to interpretability. First, at the multimodal level, feature attribution techniques like SHAP (Lundberg, 2017) and LIME (Ribeiro et al., 2016) could provide insights into how the model fuses audio and linguistic representations. Second, at the unimodal level, leveraging methods like those demonstrated by Oota et al. (2023) can help identify the unique contributions of the specific features within the audio and language model representations. Third, traditional neuroimaging methods such as representational similarity analysis (Kriegeskorte et al., 2008) remain valuable for determining which components of language and speech representations are most predictive of brain activity across regions of interest. Nonlinear models also open new avenues for interpretation that are inaccessible to linear approaches as Yang et al. (2023) showed with vision encoding. For example, exploring what the hidden layers encode could uncover brain-relevant multimodal features that current LLMs are missing. Together, these interpretability strategies could help elucidate the "black box" nature of nonlinear encoding models, offering deeper insights into their alignment with brain processes.

In conclusion, our study demonstrates the value of nonlinear, multimodal encoding models in understanding the brain's speech comprehension mechanisms. Addressing limitations in dataset size and model interpretability will be pivotal for future advancements, paving the way for biologically aligned AI systems with transformative potential in research and real-world applications.

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

# A  APPENDIX

## A.1  ABBREVIATIONS OF BRAIN AREAS AND REGIONS OF INTEREST (ROIS)

Brain Areas are abbreviated as follows :

- **AC**: Auditory Cortex
- **AG**: Angular Gyrus
- **LPFC**: Lateral Prefrontal Cortex
- **LTC**: Lateral Temporal Cortex
- **mPFC**: Medial Prefrontal Cortex
- **OC**: Occipital Cortex
- **PrCu**: Precuneus

The ROIs are abbreviated as follows :

- **AC**: Auditory Cortex
- **AG**: Angular Gyrus
- **Broca**: Broca's Area
- **EBA**: Extrastriate Body Area
- **FFA**: Fusiform Face Area
- **FEF**: Frontal Eye Field

- **IFSFP**: Inferior Frontal Sulcus Face Patch
- **LPFC**: Lateral Prefrontal Cortex
- **LTC**: Lateral Temporal Cortex
- **M1F**: Primary Motor Cortex - Foot
- **M1H**: Primary Motor Cortex - Hand
- **M1M**: Primary Motor Cortex - Mouth
- **mPFC**: Medial Prefrontal Cortex
- **OC**: Occipital Cortex
- **OFA**: Occipital Face Area
- **PMvh**: Ventral Premotor Hand Area
- **PPA**: Parahippocampal Place Area
- **PrCu**: Precuneus
- **RSC**: Retrosplenial Cortex
- **S1F**: Primary Somatosensory Cortex - Foot
- **S1H**: Primary Somatosensory Cortex - Hand
- **S1M**: Primary Somatosensory Cortex - Mouth
- **sPMv**: Superior Ventral Premotor Speech Area
- **SMFA**: Supplementary Motor Foot Area
- **SMHA**: Supplementary Motor Hand Area

## A.2 DETAILS OF IMPLEMENTATION

### A.2.1 RESAMPLING THE HIDDEN STATE OF LLMs TO FMRI TIME POINTS

After giving the language/audio model the same input as the subject, we temporally aligned the hidden states of its $l^{\text{th}}$ layer corresponding to a given $i^{\text{th}}$ token (last token of the $i^{\text{th}}$ word for language models), $H_l^i(S_{\{k|k\leq i\}}) \in \mathbb{R}^{d_{\text{model}}^l}$ (aggregate shape of $\mathbb{R}^{N_{\text{token}} \times d_{\text{model}}^l}$ for the whole story where $N_{\text{token}}$ is the number of tokens/words), to the fMRI acquisition times (TR times) using Lanczos interpolation, obtaining an extracted feature of size $\mathbb{R}^{N_{\text{TR}} \times d_{\text{model}}^l}$, where $N_{\text{TR}}$ is the number of tokens (or number of words for language models) for each story and $d_{\text{model}}^l$ is the dimension of the $l^{\text{th}}$ hidden layer. We constructed the feature corresponding to a given $n^{\text{th}}$ TR ($2n$ seconds in physical time) by concatenating the representations from four previous TRs (2, 4, 6, 8 seconds before $t$ in physical time) to get a vector of shape $\mathbb{R}^{4d_{\text{model}}^l}$ for every $n^{\text{th}}$ TR, which we denote as $H'^n_l(S_{\{t|t\leq 2n\}})$. $H'$ denotes the additional resampling and concatenation done after applying the model, $H$. We used four previous time delays (2, 4, 6, 8 seconds) to account for the delay between the stimuli and brain response and to provide past stimuli information to the model.

### A.2.2 REPRESENTATIONS FOR FMRI RESPONSE USING PCA

To an aggregate fMRI response, $Y_{\text{org}} \in \mathbb{R}^{N_{\text{TR}} \times N_{\text{voxels}}}$, we applied PCA with 8192 maximum components along the voxel dimension using scikit-learn (Pedregosa et al., 2011), yielding an approximate projection matrix, $W \in \mathbb{R}^{N_{\text{voxels}} \times N_{8192}}$. Given $N_{\text{PCA}}$ number of principal components to consider, we take the top $N_{\text{PCA}}$ components to get $W_{\text{PCA}} \in \mathbb{R}^{N_{\text{voxels}} \times N_{\text{PCA}}}$, and train the encoding model to predict the reduced dimension PCA projection of the data, $Y_{\text{PCA}} = Y_{\text{org}} W_{\text{PCA}} \in \mathbb{R}^{N_{\text{TR}} \times N_{\text{PCA}}}$. During evaluation, the trained model outputs a reduced dimension representation of the data, $\hat{Y}_{\text{PCA}}^{\text{test}} \in \mathbb{R}^{N_{\text{TR-test}} \times N_{\text{PCA}}}$, where $N_{\text{TR-test}}$ denotes the number of timepoints (TRs) in the test story. This is reconstructed back the the original voxel space by applying an inverse of the projection matrix, $\hat{Y}^{\text{test}} = \hat{Y}_{\text{PCA}}^{\text{test}} W_{PCA}^T \in \mathbb{R}^{N_{\text{TR-test}} \times N_{\text{voxels}}}$, which is later compared with the ground truth, $Y^{\text{test}} \in \mathbb{R}^{N_{\text{TR-test}} \times N_{\text{voxels}}}$.

It should be noted that due to the high dimensionality of the data, incremental PCA was used, in place of regular PCA.

### A.2.3 DETAILS OF ENCODING MODELS

The encoding model architecture is as follows:

- *Linear Regression (Linear):* Ridge regression. Following Antonello et al. (2024), ridge regression with bootstrapping ($n = 3$) was used to estimate the optimal regularization

parameters (alphas) for each voxel. The training data was divided into chunks of length 20, with 25% used for held-out validation in each bootstrap iteration. The best alpha values were averaged across iterations, and the final model was trained on the full training dataset using these alphas.

- *Multi-Layer Perceptron (MLP):* MLP with a single hidden layer of 256 units, applying batch normalization and dropout to prevent overfitting. The hyperbolic tangent (tanh) was used as the activation function.
- *Multi-Layer Linear (MLLinear):* MLP but without dropout, batch normalization, and with the identity activation function.
- *Delayed Interaction MLP (DIMLP):* MLP variant processes. Each modality through separate 256-unit hidden layers before concatenation and final linear projection.

We implemented encoding models using PyTorch. We employed the AdamW optimizer (Loshchilov, 2017) with a batch size of 128 and Mean Absolute Error (MAE) as the loss function to mitigate excessively penalizing random signal fluctuations. Our training regime consisted of 200 epochs with early stopping (patience = 10) based on validation loss, and we applied batch normalization with a momentum of 0.1. For robust evaluation, we implemented 5-fold cross-validation, averaging predictions across the five models for our final results. Hyperparameter optimization was conducted using Optuna (Akiba et al., 2019), which performed 70 trials to determine optimal values for the dropout rate (0.1 to 0.3), learning rate ($10^{-5}$ to $10^{-1}$), and weight decay ($5 \times 10^{-5}$ to $10^{-1}$).

Ridge regression was performed using a CPU node with 96 cores (Intel(R) Xeon(R) Gold 6240R CPU @ 2.40GHz) and 512 GB of RAM. Running the audio and language models and training encoding models was done using a GPU node with 8 H100 80GB GPUs.

## A.3 COMPARISON WITH STACKED REGRESSION MODEL OF ANTONELLO ET AL. (2024)

To establish the effectiveness of our nonlinear multimodal approach, we conduct a detailed comparison with the current state-of-the-art stacked regression model (Antonello et al., 2024). Their method combines semantic and audio predictions through stacked regression followed by voxel-selection, where they decide what model to use (stacked regression or semantic linear) for each voxel based on a validation dataset. Their results are compared here and not in Table 1 due to their use of parts of the test stories as validation, barring computation of the "**Avg** $r^2$" value in Table 1. For accurate comparison, we obtain and use their published model weights and features.

The evaluation protocols differ specifically for the stacked regression (SR) model: while all models (including those in Antonello et al. (2024)) primarily report performance using three test stories (Table 1), SR uniquely requires using two of these test stories for validation-based voxel selection and only using the story "wheretheressmoke" for final testing.

Also, following the identification of an error in the original evaluation protocol through community feedback, we corrected the methodology for fair comparison. Note that $CC_{norm}$ values remain consistent with Table 1 as they were originally computed using only the "wheretheressmoke" story due to the unavailability of test repeats for the other two stories.

To ensure fair comparison with SR, we additionally evaluate all models using their single-story protocol in Table 2, reporting both $CC_{norm}$ and a story-specific **Avg** $r^2$ **(story)** metric to distinguish from our three-story evaluation. We found $CC_{norm}$ provides more stable comparisons than $r^2$ in this context, as the reduced number of timepoints (251 versus 790) makes $r^2$ more susceptible to noisy voxels compared to $CC_{norm}$ that accounts for these noisy voxels. This stability is reflected in the closer alignment between $CC_{norm}$ and $r^2$ rankings in Table 1 compared to Table 2. Therefore, we sort Table 2 with respect to the $CC_{norm}$.

Also, while their approach uses LLAMA-30B's 18th layer (denoted as semantic$_A$), we demonstrate competitive performance using LLAMA-7B features, consistent with our finding that encoding performance roughly plateaus beyond 7B parameters (Appendix A.5). For comprehensive comparison, we implement both their pre-computed validation-based voxel selection mask ("mask$_A$", created using an unspecified significance threshold) and our simpler approach ("mask") that retains voxels showing any validation set improvement.

Table 2: Comparing encoding performance across different models using the single test story evaluation protocol. Values show normalized correlation coefficient ($CC_{norm}$) and story-specific $r^2$ (**Avg $r^2$ (story)**)(distinguishing from Table 1's three-story evaluation (**Avg $r^2$**)). SR refers to the stacked regression model (Antonello et al., 2024), which combines LLM and audio predictions through weighted averaging. Two masking approaches are used: 1) "mask$_A$" - their pre-computed validation-based voxel selection mask, and 2) "mask" - our computed masks that retain voxels showing validation improvements. For "mask", Linear+Mask indicates creating and applying a mask based on multimodal linear vs semantic linear performance, while MLP+Mask does the same using MLP models. semantic$_A$ denotes features from LLAMA-30B's 18th layer used in SR, while our models uses features from the 12th layer of LLAMA-7B. All approaches are evaluated using identical test data for fair comparison and $r^2$ is computed as $|r| * r$.

| modality 1 | modality 2 | encoder | response | Avg $CC_{norm}$ | Avg $r^2$ (story) |
|---|---|---|---|---|---|
| semantic | audio | MLP | PCA | **34.32%** (+14.4%) | **5.13%** (+7.7%) |
| semantic | audio | MLP + mask | PCA | 33.33% (+11.0%) | 5.02% (+5.5%) |
| semantic | audio | DIMLP | PCA | 32.59% (+8.6%) | 4.93% (+3.6%) |
| semantic | audio | MLLinear | PCA | 32.41% (+8.0%) | 5.00% (+5.1%) |
| semantic | audio | MLP + mask$_A$ | PCA | 31.70% (+5.6%) | 4.77% (+0.2%) |
| semantic | audio | Linear | all voxels | 31.36% (+4.5%) | 4.92% (+3.4%) |
| semantic | audio | MLP | all voxels | 31.11% (+3.6%) | 4.54% (-4.5%) |
| semantic | audio | Linear + mask | all voxels | 31.09% (+3.6%) | 4.90% (+2.9%) |
| semantic$_A$ | audio | SR + mask$_A$ | all voxels | 30.02% (Baseline) | 4.76% (Baseline) |
| semantic | audio | Linear | PCA | 28.92% (-3.7%) | 4.48% (-5.8%) |
| semantic | - | MLP | PCA | 30.89% (+2.9%) | 4.58% (-3.7%) |
| semantic | - | MLLinear | PCA | 29.95% (-0.2%) | 4.59% (-3.6%) |
| semantic$_A$ | - | Linear | all voxels | 29.84% (-0.6%) | 4.60% (-3.3%) |
| semantic | - | Linear | all voxels | 29.12% (-3.0%) | 4.50% (-5.4%) |
| semantic | - | MLP | all voxels | 27.45% (-8.6%) | 3.97% (-16.6%) |
| semantic | - | Linear | PCA | 26.88% (-10.4%) | 4.15% (-12.8%) |
| audio | - | MLP | PCA | 29.01% (-3.4%) | 3.83% (-19.6%) |
| audio | - | MLP | all voxels | 28.21% (-6.0%) | 3.67% (-22.8%) |
| audio | - | MLLinear | PCA | 27.50% (-8.4%) | 3.66% (-23.1%) |
| audio | - | Linear | PCA | 26.71% (-11.0%) | 3.54% (-25.6%) |
| audio | - | Linear | all voxels | 25.20% (-16.0%) | 3.46% (-27.3%) |

Table 2 demonstrates several key results about our multimodal nonlinear approach. Our multimodal MLP achieves 34.32% CC$_{norm}$ without masking, representing a 14.4% improvement over the baseline stacked regression model, though the Avg $r^2$ (story) improvement is more modest at 7.7%.

Our multimodal linear encoder also outperforms stacked regression by 4.5%, supporting our hypothesis that direct concatenation enables more effective modality interaction compared to weighted averaging of unimodal predictions. The performance hierarchy (MLP > Linear > SR) suggests that both architectural choices - direct multimodal fusion and nonlinearity - contribute independently to improved predictions.

Interestingly, validation-based masking did not improve performance for either our linear or MLP models, regardless of whether using our mask or the precomputed mask$_A$ from previous work. This suggests our models learn effective feature selection implicitly, determining when to leverage or ignore audio features for specific voxels without explicit masking. The benefit of removing masking also likely stems from our models' ability to learn voxel-specific feature importance through direct access to input data, combined with the inherent noise in validation masks due to the limited number of timepoints.

These results demonstrate that enabling direct interaction between modalities through concatenation, combined with nonlinear processing, provides a more robust approach than previous methods relying on weighted averaging and explicit feature selection.

A.4 RESULTS OF MORE COMPLEX NONLINEAR MODELS

We explored a range of more complex nonlinear models, as detailed in Table 3. Specifically, we evaluated LSTM, GRU, RNN, and Transformer architectures, each configured with a single layer. The hidden dimensions for these models were determined by experimenting with sizes of 256, 512, 768, and 1024, selecting the dimension that yielded the best performance.

All models received inputs consisting of four timepoints, consistent with the MLP model, which concatenates these timepoints. For the recurrent models (LSTM, GRU, RNN), the final predictions were generated by applying a linear projection to a weighted pooling of the outputs corresponding to the four input timepoints. In the case of the Transformer model, we utilized learnable positional embeddings along with full self-attention mechanisms, and the final prediction was obtained by linearly projecting the output of the last token.

Additionally, we examined the DeepMLP model, an extension of the standard MLP with two hidden layers instead of one.

Our results indicate that while the MLP with a single hidden layer outperforms linear models, introducing greater complexity—such as recurrent models or additional hidden layers—leads to over-fitting and decreased performance.

Table 3: Encoding performance of various nonlinear semantic encoders compared to other models. The table presents the average $r^2$ and normalized correlation coefficients ($CC_{norm}$) along with percentage changes relative to the baseline Linear model. Deep MLP refers to an MLP with two hidden layers, while MLP is an MLP with one hidden layer.

| Modality 1 | Modality 2 | Encoder | Response | Avg $r^2$ | Avg $CC_{norm}$ |
|---|---|---|---|---|---|
| semantic | - | MLP | PCA | 3.79% (+3.6%) | 30.89% (+6.1%) |
| **semantic** | - | **Linear** | **all voxels** | **3.66% (Baseline)** | **29.12% (Baseline)** |
| semantic | - | LSTM | PCA | 3.33% (-9.0%) | 26.95% (-7.46%) |
| semantic | - | GRU | PCA | 3.21% (-12.3%) | 26.15% (-10.2%) |
| semantic | - | DeepMLP | PCA | 3.05% (-16.7%) | 27.45% (-5.73%) |
| semantic | - | RNN | PCA | 2.99% (-18.0%) | 25.42% (-12.7%) |
| semantic | - | Transformer | PCA | 2.82% (-23.0%) | 27.97% (-3.95%) |

A.5 SCALING LLM AND AUDIO MODELS DOES NOT NECESSARILY LEAD TO BETTER ENCODERS

Previous research by Antonello et al. (2024) found that increasing the size of large language models (LLMs) and audio models, such as scaling OPT from 125M to 175B parameters or Whisper from 8M to 637M parameters, enhanced encoding performance. However, performance gains plateaued for larger models like LLAMA-33B and OPT-175B, which they attributed to overfitting from larger hidden sizes.

Building on these findings, our study delves deeper into the scaling trends and offers a refined perspective on their implications for brain encoding models. For audio models, we confirm a positive correlation between model size and performance, as shown in Figure 3 (d). However, this scaling effect does not hold for language models. Specifically, LLAMA-7B, LLAMA-13B, LLAMA-33B, and LLAMA-65B exhibit comparable encoding performance, as shown in Figure 3 (b). This suggests diminishing returns beyond 7 billion parameters, a finding consistent with prior work by Bonnasse-Gahot & Pallier (2024), which reported performance plateaus for LLMs larger than 3 billion parameters.

We also evaluated the impact of scaling training data by examining newer versions of LLAMA and Whisper (e.g., LLAMA-1, LLAMA-2, LLAMA-3; Whisper v1, v2, v3). Despite larger datasets, newer versions did not yield significant performance improvements for either audio or semantic encoding models. This indicates that advancements in self-supervised learning (SSL) tasks, such as better next-token prediction, do not necessarily translate to more effective features for brain encoding. In essence, SSL improvements do not directly enhance brain-aligned representations.

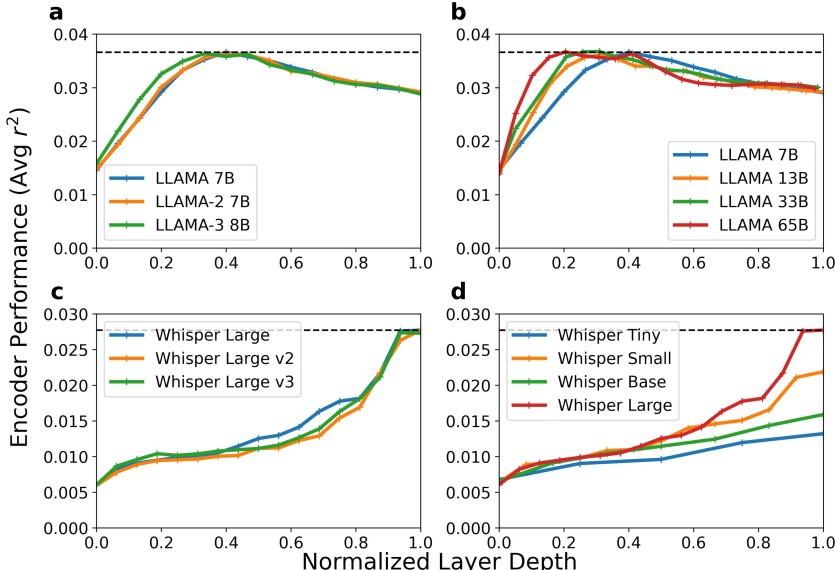

Figure 3: Encoder performance across different LLAMA and Whisper model variants, using linear regression applied to the full set of voxels. Panel (a) compares LLAMA models of various architectures (LLAMA-2 and LLAMA-3) with 7B and 8B parameters. Panel (b) presents performance across different LLAMA models of increasing sizes, from 7B to 65B. Panels (c) and (d) show the performance for different Whisper model variants, including comparisons between Whisper Large versions (c) and different model sizes (d), from Whisper Tiny to Whisper Large. Performance is measured in terms of average $r^2$, plotted against normalized layer depth.

In conclusion, our findings highlight two key points: (1) scaling language models beyond 7 billion parameters does not substantially improve encoding performance, and (2) increasing training data or using newer model versions does not enhance brain encoding feature extractors. These results challenge the assumption that simply scaling feature extractors, as proposed by Antonello et al. (2024), will lead to better encoding models.

## A.6    CONTEXT SIZE SPEECH MODELS INFLUENCE ENCODER PERFORMANCE

Figure 4 illustrates the impact of varying the context size (window size) of the Whisper model on encoding performance when using linear encoders, as explored in Oota et al. (2023). The results indicate that a 16-second window size, which was used as the default throughout our study, delivers the best performance. This outcome aligns with expectations, as the selected window size is consistent with the recommendations from Antonello et al. (2024).

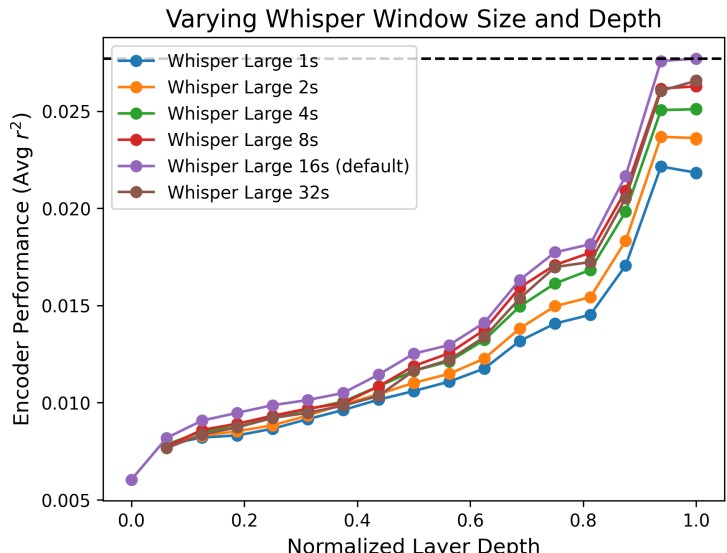

Figure 4: Encoder performance across different Whisper Large models with varying window size, using linear regression applied to the full set of voxels.

## A.7 VARIANCE PARTITIONING ANALYSIS

To quantify the unique contributions of different feature spaces in our nonlinear multimodal encoding models, we employed a variance partitioning analysis similar to de Heer et al. (2017). This approach allowed us to determine how much variance could be uniquely explained by each feature versus that explained by a multiple features. We estimated both the fraction of variance explained by each feature space individually and the fraction that might be equally well explained by combinations of feature spaces.

We show our variance partitioning analysis results in three complementary ways: 1) voxel-wise variance partition results (Appendix A.7.2), 2) voxel-wise plots showing the largest variance partition for each voxel (Appendix A.7.3), and 3) ROI-wise Venn diagrams illustrating the distribution of variance explained across different brain regions (Appendix A.7.4).

For this analysis, we fit models with all possible combinations of feature spaces: two single-feature models (audio and semantic), one model combining both features (semantic-audio), and examined the distribution of variance explained within brain regions. This allowed us to decompose the total explained variance into three components: variance uniquely explained by audio features, variance uniquely explained by semantic features, and variance jointly explained by both feature spaces.

### A.7.1 SUMMARY OF VARIANCE PARTITIONING RESULTS

Looking at the results of Appendix A.7.2, we observe that joint variance dominates across most cortical regions, contrasting with de Heer et al. (2017) where semantic only features showed greater dominance. This difference likely stems from our feature choices - whereas de Heer et al. (2017) used spectral and articulatory features that primarily contained information relevant mostly only to auditory cortex, our use of Whisper features provides richer auditory representations that enable better predictions beyond traditional auditory regions. This finding aligns with our earlier argument (Section 3.2.3) that multiple modalities jointly contribute to neural computations across the cortex rather than having one modality dominate.

The dominance pattern of joint variance is consistent both within and near AC, with a notable exception in early auditory regions where audio features show unique contributions. This hierarchical organization suggests that while early AC predominantly processes pure acoustic information, later AC regions integrate both semantic and auditory features for higher-level speech processing.

The unique contribution of audio features in early AC is noteworthy as it suggests preservation of modality-specific processing at early sensory stages despite using rich Whisper features.

Also, Appendix A.7.3 reveals distinct spatial patterns in feature representation across cortical regions. The prefrontal cortex exhibits mixed dominance patterns, showing both joint semantic-audio representation and semantic-only areas. While early auditory cortex shows expected unique audio contributions, we also observe audio-specific representation in motor-sensory mouth areas (M1M, S1M), though this pattern varies across subjects. Notably, for subject S3, these audio-specific contributions in M1M and S1M become more pronounced when using MLP encoders (Figure 13), suggesting nonlinear encoding may better capture modality-specific processing in these regions.

The ROI-wise analysis in Appendix A.7.4 reveals that joint semantic-audio features dominate cortical representation, accounting for approximately 65% of significantly predicted voxels across the entire cortex. Core language-processing regions (AC, Broca's area, sPMv) show particularly strong joint representation (around 80 to 90%), supporting our hypothesis that speech comprehension relies on integrated multimodal processing. This integration is consistently observed across subjects, though some ROIs (e.g., PMvh in Subject S2 with only 14 voxels) have insufficient data for reliable interpretation. The transition from linear to MLP encoders increases the total number of significantly predicted voxels while maintaining similar representation patterns, indicating that nonlinear encoding primarily enhances prediction accuracy rather than fundamentally altering feature representation structure.

### A.7.2 VARIANCE PARTITIONING OF VARIOUS MODELS

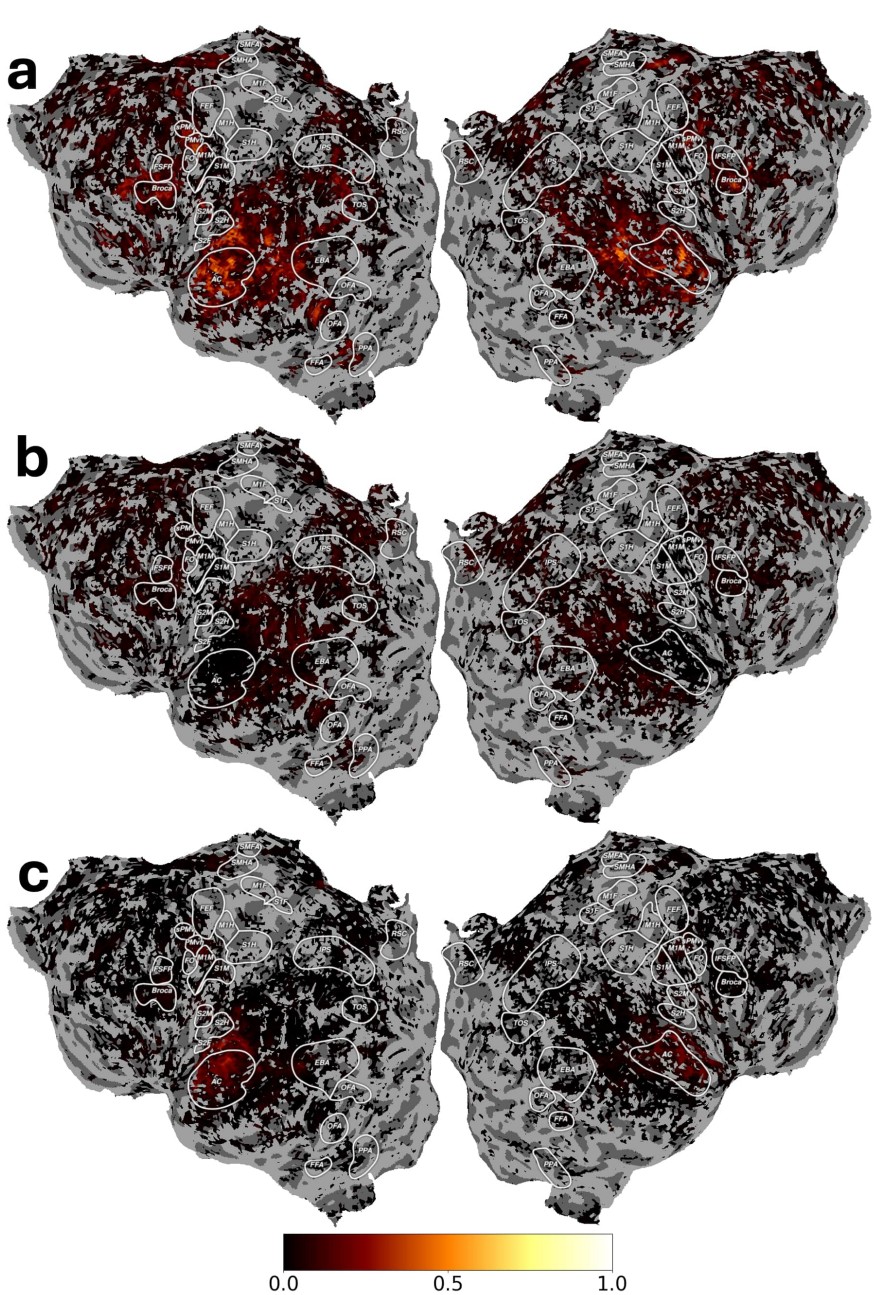

Figure 5: Voxelwise variance partitioning analysis showing the contributions of different feature types to prediction accuracy for a subject S1 using linear models. The flatmaps display (a) variance jointly explained by audio and semantic features, (b) variance uniquely explained by semantic features, and (c) variance uniquely explained by audio features. Values shown are normalized correlations ($CC_{norm}$) for voxels where the joint model achieved significant prediction ($q(\text{FDR}) < 0.01$).

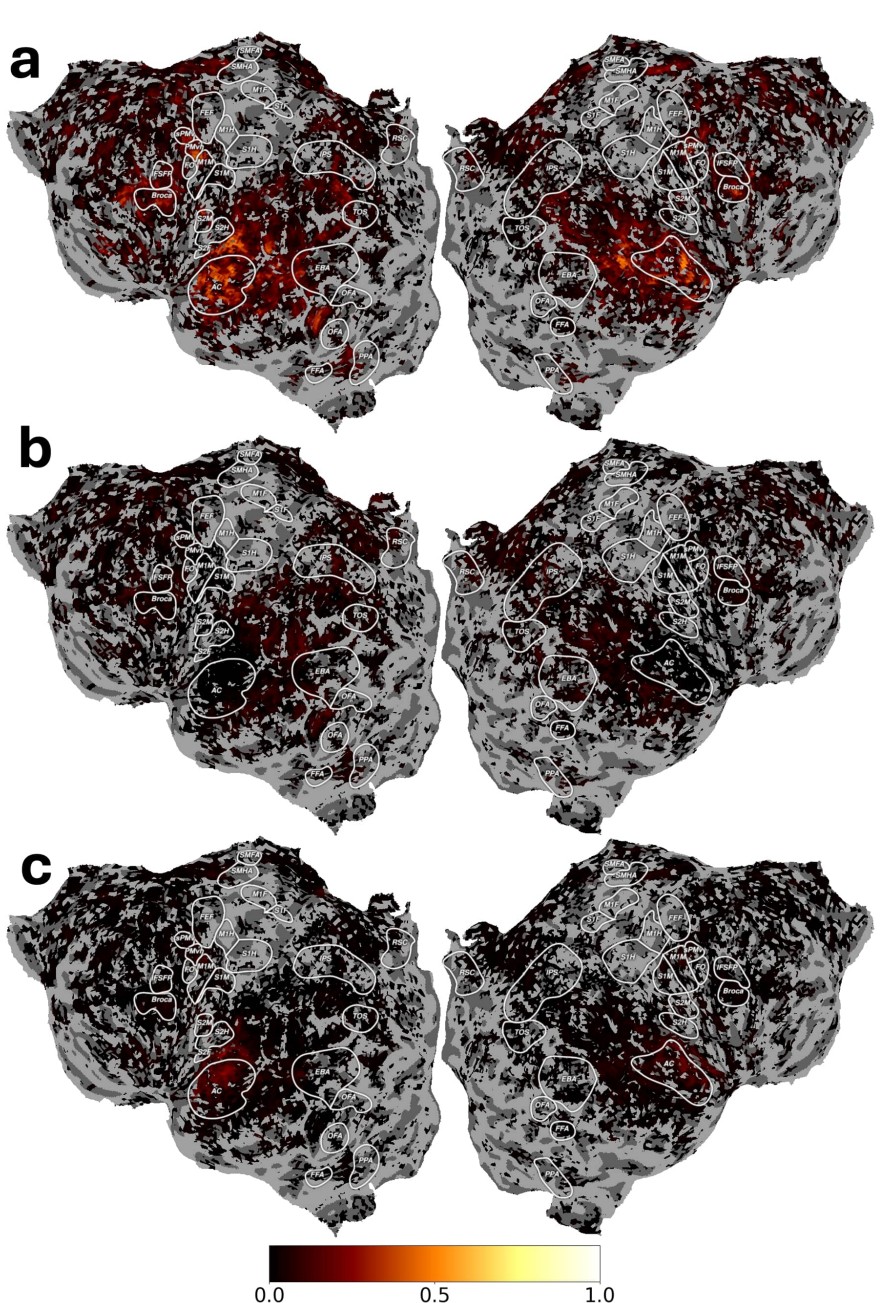

Figure 6: Voxelwise variance partitioning analysis showing the contributions of different feature types to prediction accuracy for a subject S1 using MLP models. The flatmaps display (a) variance jointly explained by audio and semantic features, (b) variance uniquely explained by semantic features, and (c) variance uniquely explained by audio features. Values shown are normalized correlations ($CC_{norm}$) for voxels where the joint model achieved significant prediction ($q(\text{FDR}) < 0.01$).

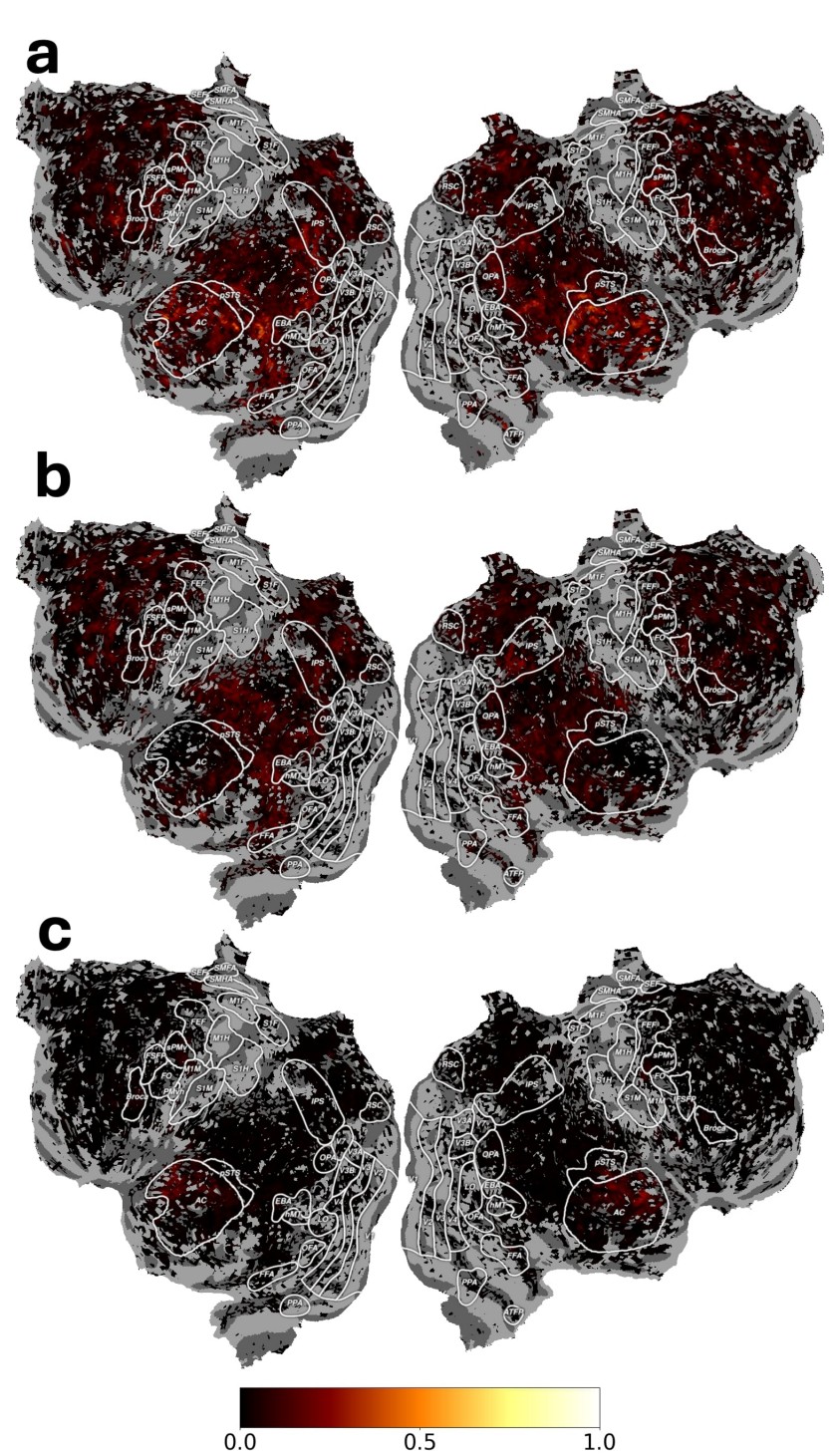

Figure 7: Voxelwise variance partitioning analysis showing the contributions of different feature types to prediction accuracy for a subject S2 using linear models. The flatmaps display (a) variance jointly explained by audio and semantic features, (b) variance uniquely explained by semantic features, and (c) variance uniquely explained by audio features. Values shown are normalized correlations ($CC_{norm}$) for voxels where the joint model achieved significant prediction ($q$(FDR) $< 0.01$).

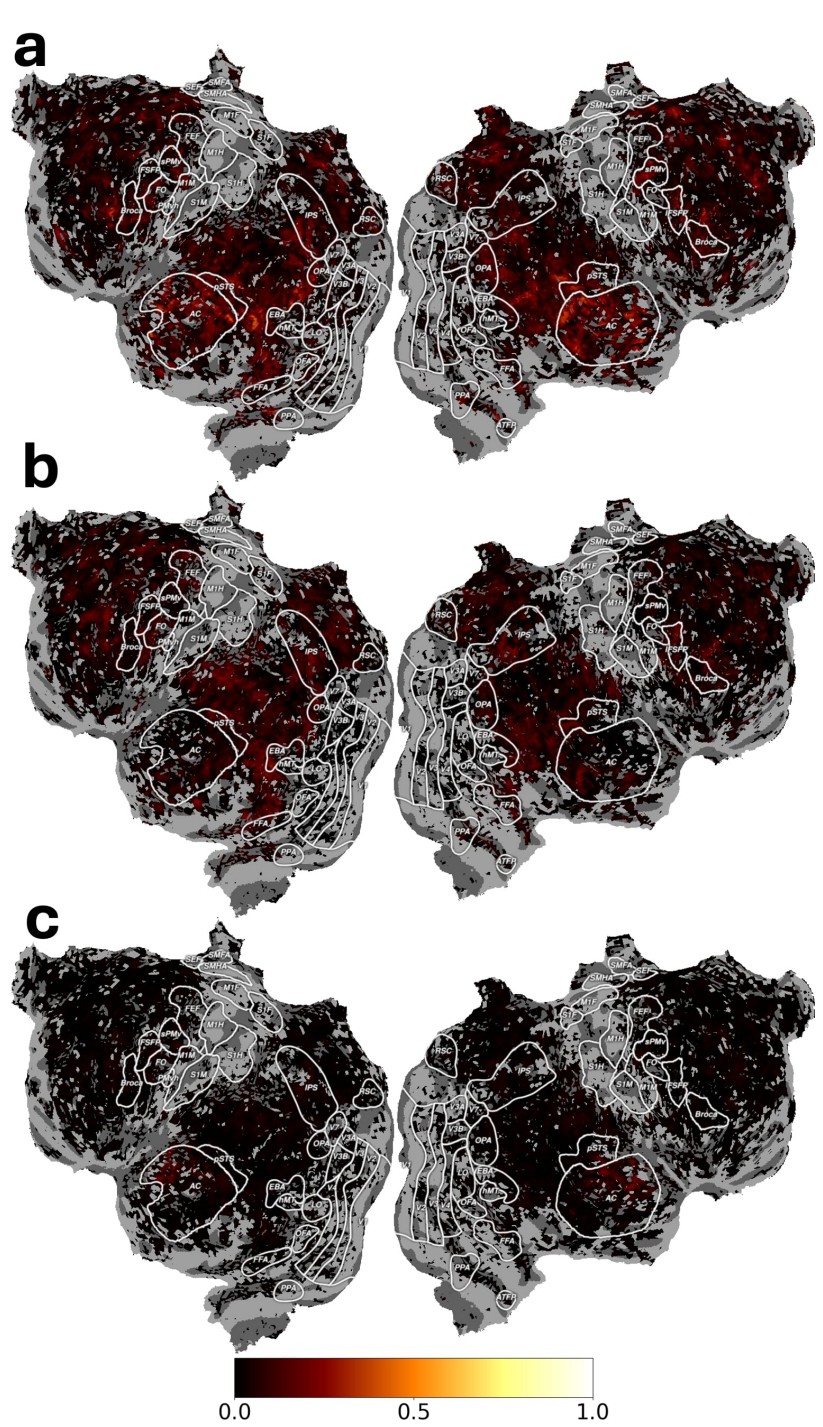

Figure 8: Voxelwise variance partitioning analysis showing the contributions of different feature types to prediction accuracy for a subject S2 using MLP models. The flatmaps display (a) variance jointly explained by audio and semantic features, (b) variance uniquely explained by semantic features, and (c) variance uniquely explained by audio features. Values shown are normalized correlations ($CC_{norm}$) for voxels where the joint model achieved significant prediction ($q$(FDR) $< 0.01$).

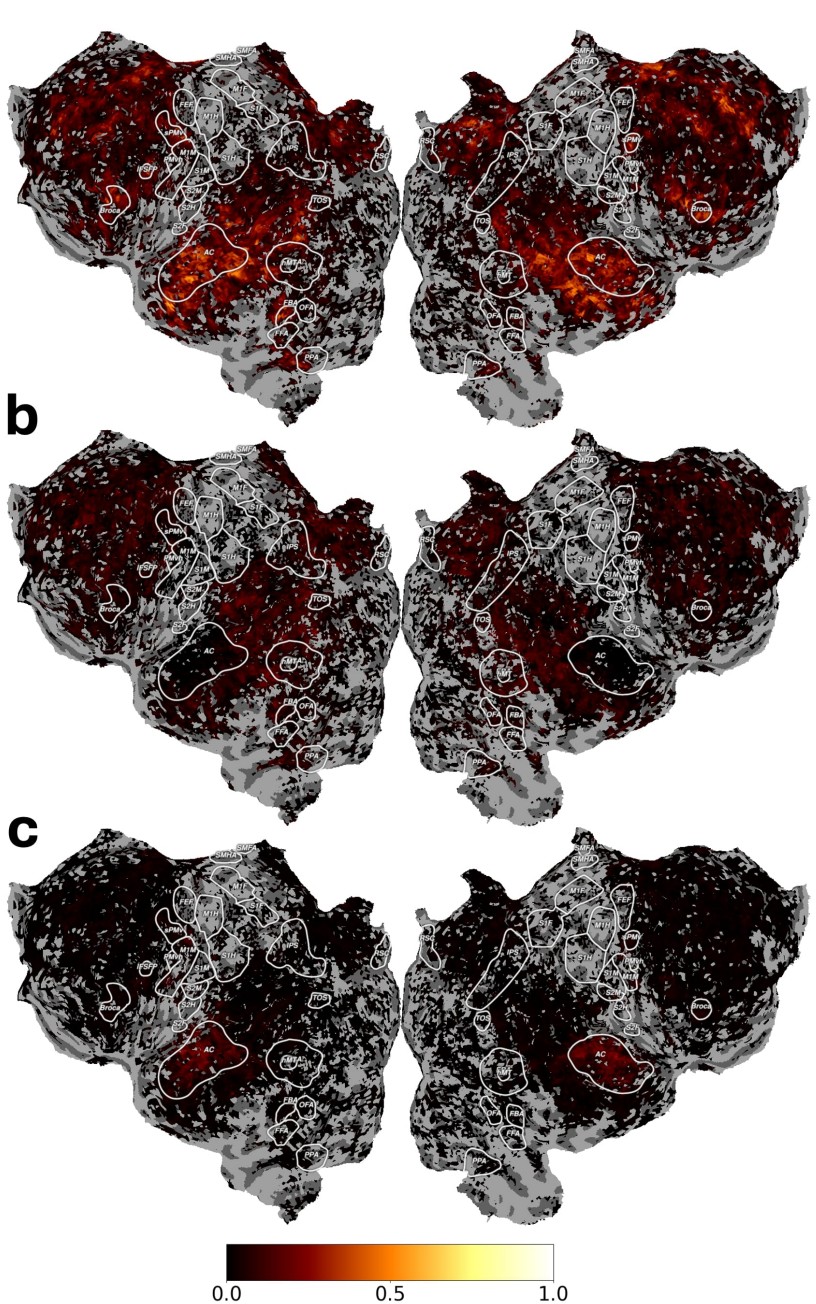

Figure 9: Voxelwise variance partitioning analysis showing the contributions of different feature types to prediction accuracy for a subject S3 using linear models. The flatmaps display (a) variance jointly explained by audio and semantic features, (b) variance uniquely explained by semantic features, and (c) variance uniquely explained by audio features. Values shown are normalized correlations ($CC_{norm}$) for voxels where the joint model achieved significant prediction ($q(\text{FDR}) < 0.01$).

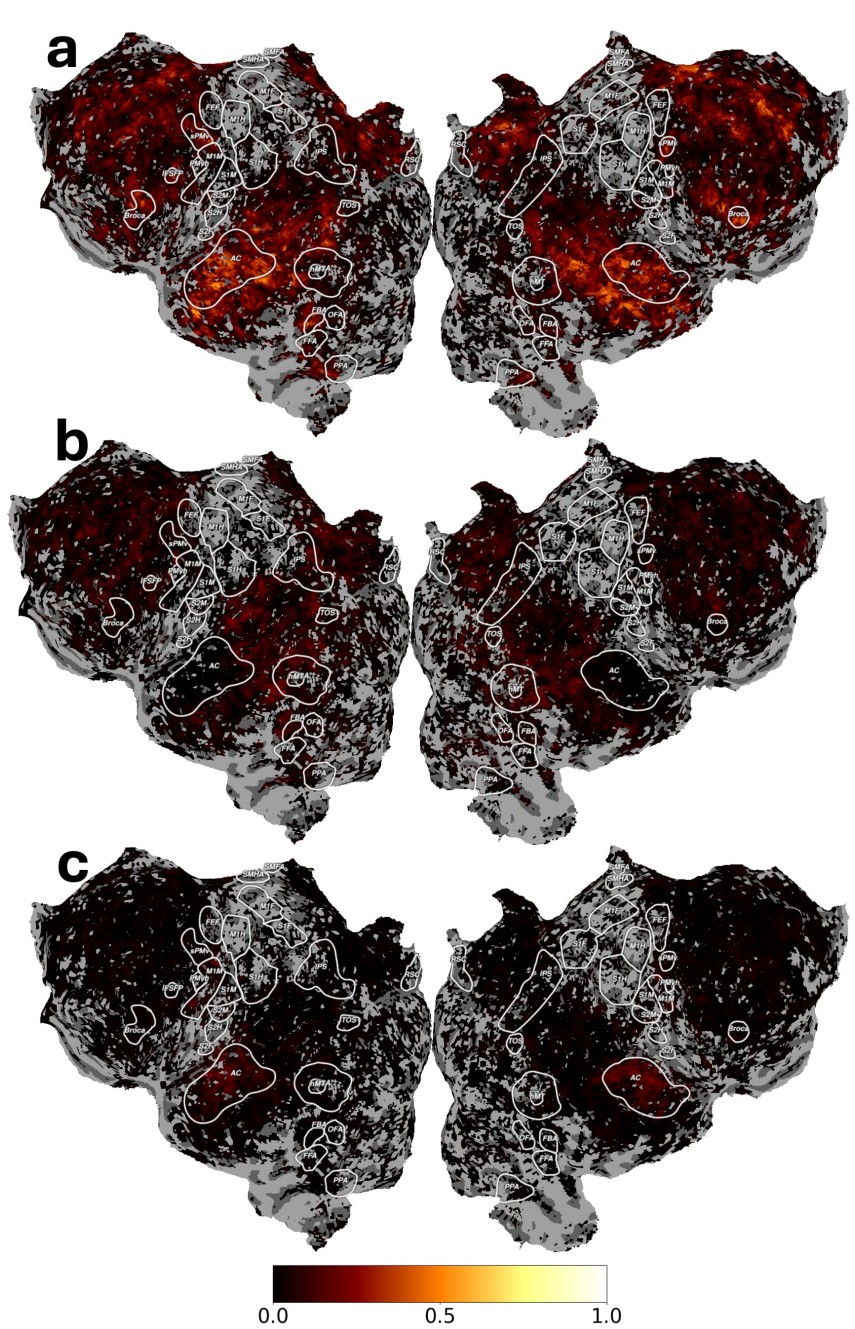

Figure 10: Voxelwise variance partitioning analysis showing the contributions of different feature types to prediction accuracy for a subject S3 using MLP models. The flatmaps display (a) variance jointly explained by audio and semantic features, (b) variance uniquely explained by semantic features, and (c) variance uniquely explained by audio features. Values shown are normalized correlations ($CC_{norm}$) for voxels where the joint model achieved significant prediction ($q(\text{FDR}) < 0.01$).

### A.7.3 LARGEST VARIANCE PARTITIONING FOR EACH VOXEL

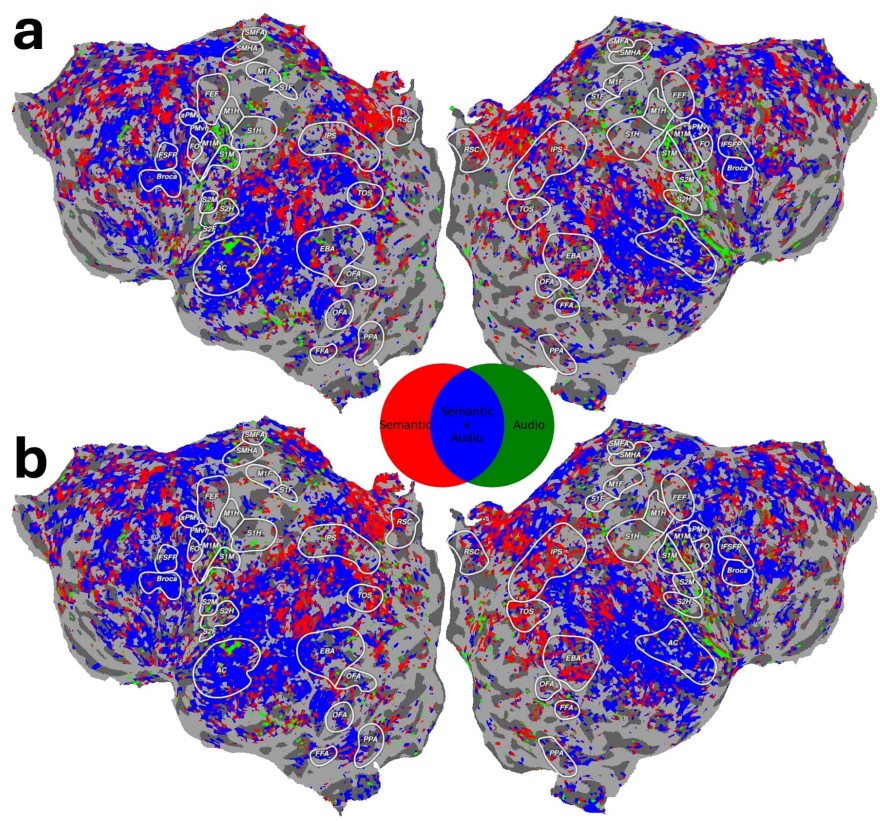

Figure 11: Voxelwise analysis showing the largest variance explained by each feature type for all significantly predicted voxels ($q(\text{FDR}) < 0.01$) for subject S1. The flatmaps display which feature partition (semantic in red, audio in green, or their combination in blue) best explains the variance in each cortical voxel using (a) linear and (b) MLP encoders, with outlined regions indicating key functional areas.

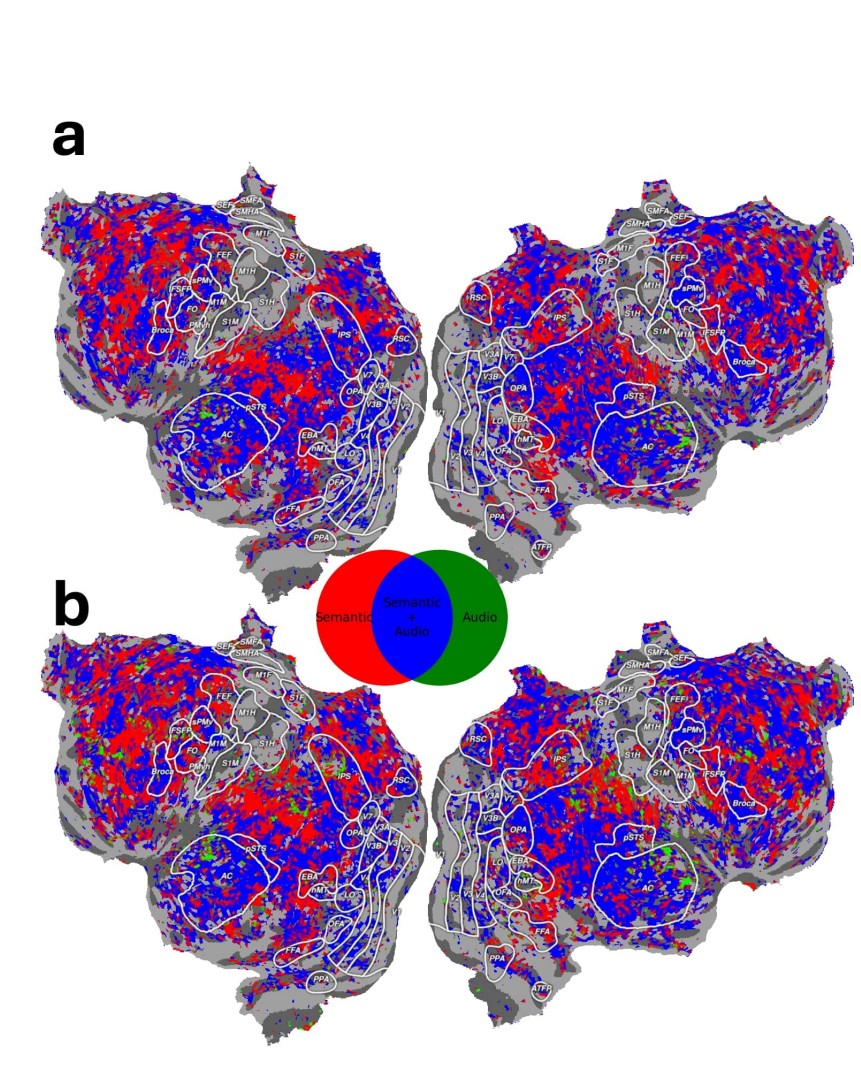

Figure 12: Same as Figure 11, but for subject S2

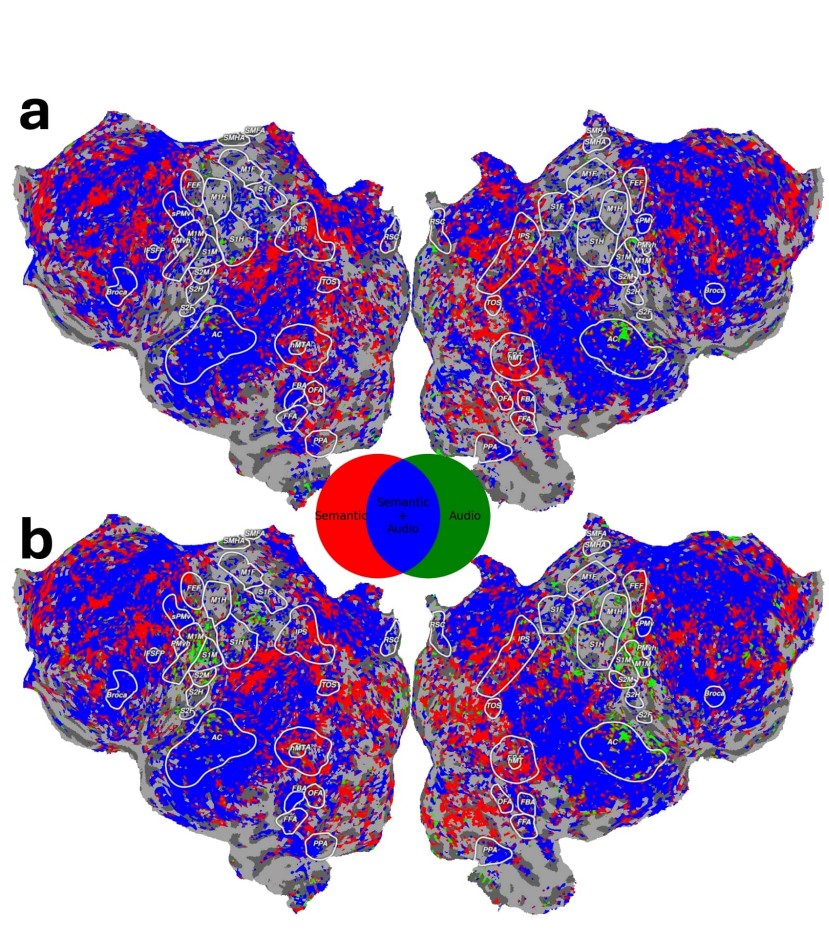

Figure 13: Same as Figure 11, but for subject S3

### A.7.4 VARIANCE PARTITIONING VENN DIAGRAM

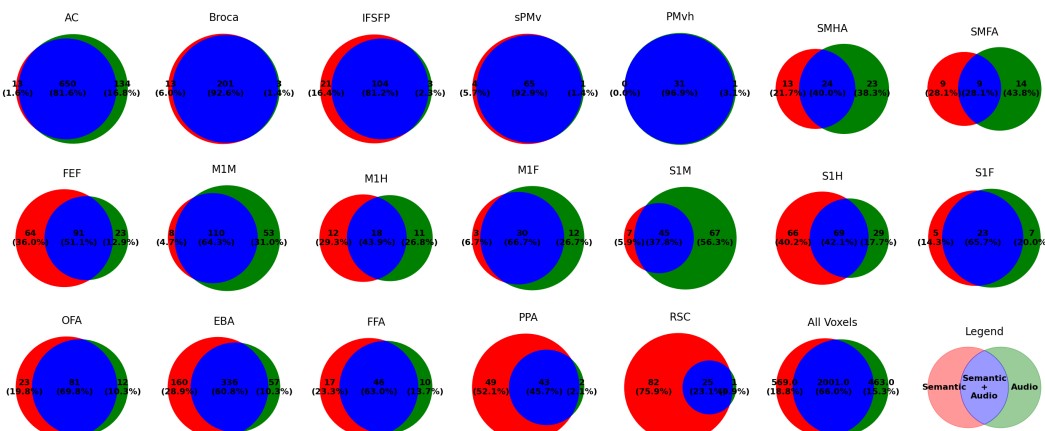

Figure 14: Venn diagrams showing the distribution of explained variance across different brain regions of interest (ROIs) for subject S1, using linear encoder. Each diagram displays the unique and shared variance explained by semantic features (red), audio features (green), and their overlap (blue). Values indicate the number of significantly predicted voxels and their percentages. Only the voxels that was predicted statistically significantly ($q(\text{FDR}) < 0.01$) was used in the analysis

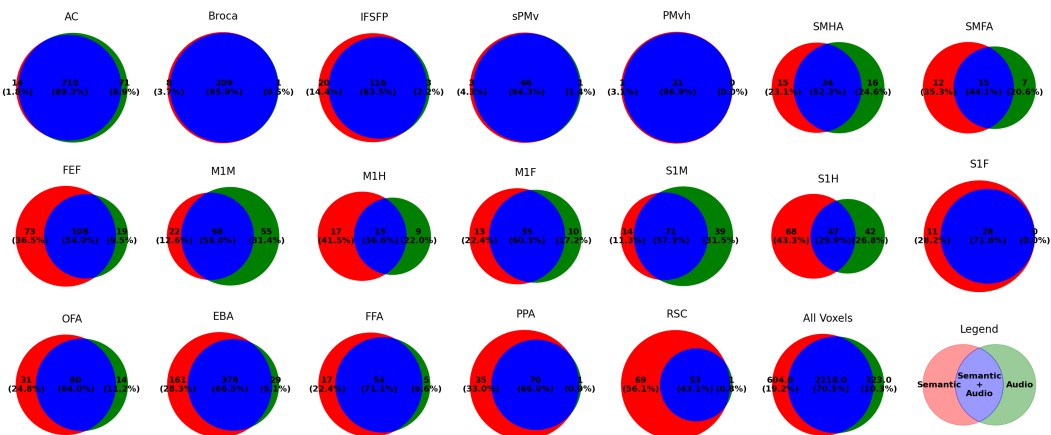

Figure 15: Venn diagrams showing the distribution of explained variance across different brain regions of interest (ROIs) for subject S1, using MLP encoder. Refer to Fig 14 for more detail.

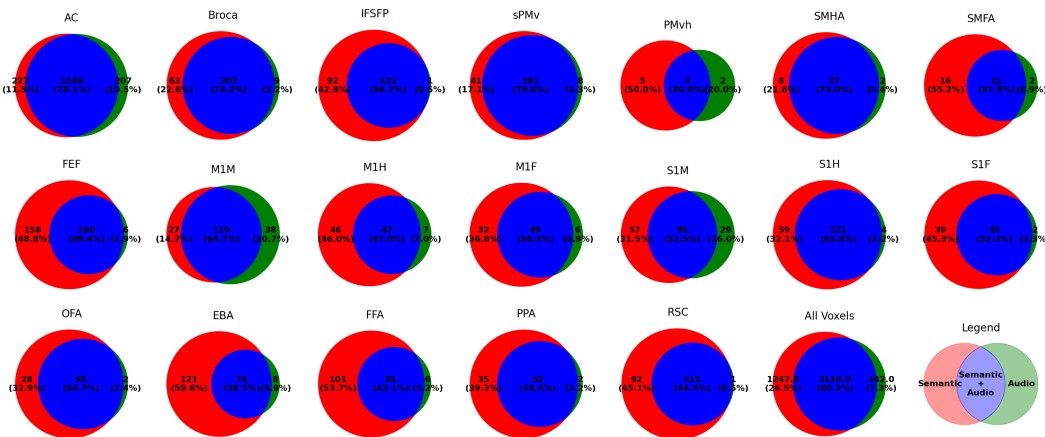

Figure 16: Venn diagrams showing the distribution of explained variance across different brain regions of interest (ROIs) for subject S2, using linear encoder. Refer to Fig 14 for more detail.

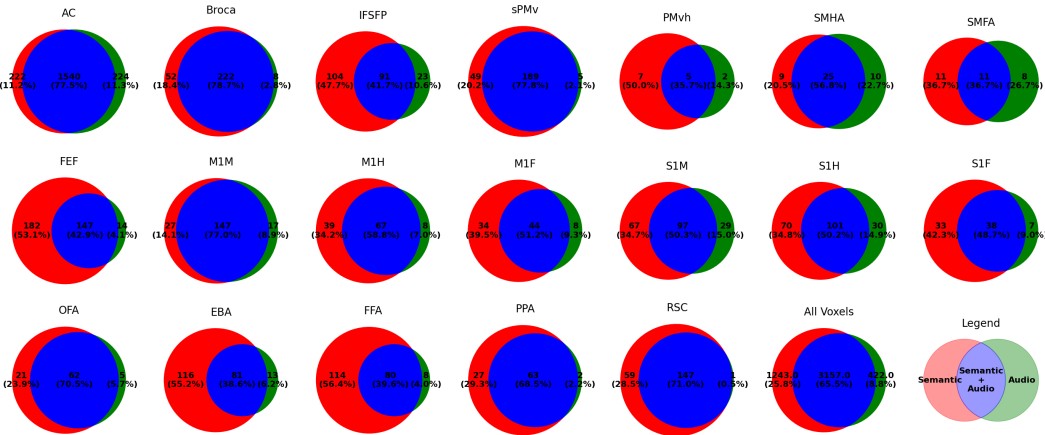

Figure 17: Venn diagrams showing the distribution of explained variance across different brain regions of interest (ROIs) for subject S2, using MLP encoder.Refer to Fig 14 for more detail.

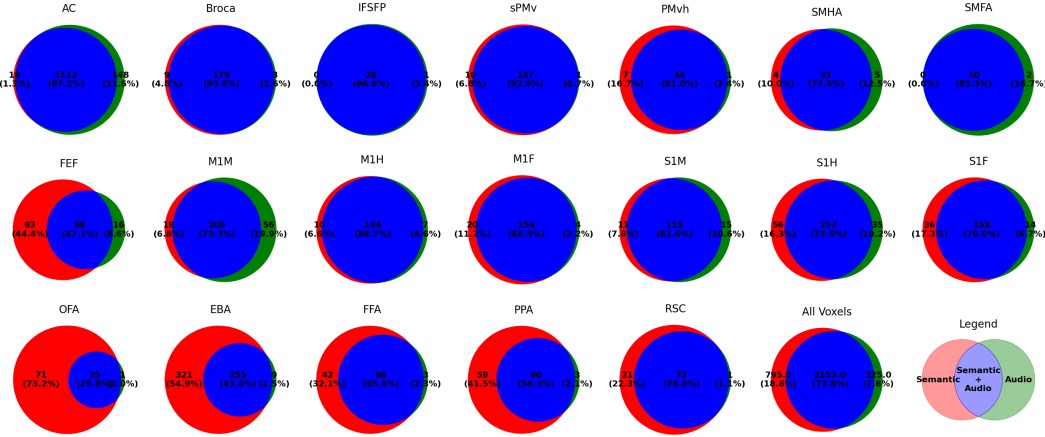

Figure 18: Venn diagrams showing the distribution of explained variance across different brain regions of interest (ROIs) for subject S3, using linear encoder. Refer to Fig 14 for more detail.

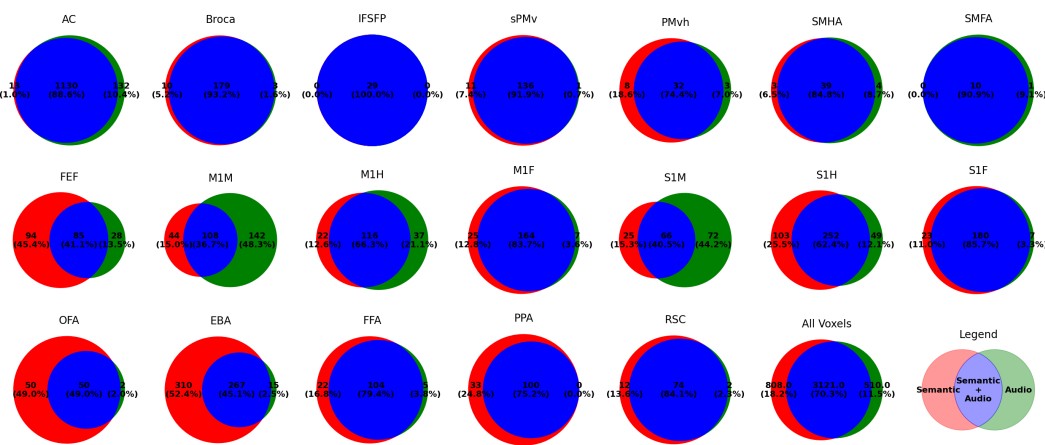

Figure 19: Venn diagrams showing the distribution of explained variance across different brain regions of interest (ROIs) for subject S3, using MLP encoder. Refer to Fig 14 for more detail.

## A.8 PERFORMANCE OF VARIOUS ENCODING MODELS USING DIFFERENT INPUTS

### A.8.1 VOXELWISE $r$ VALUES FROM DIFFERENT ENCODING MDOELS AND STIMULI

Figures 20, 21, and 22 each represent the voxelwise correlation ($r$) values using various encoders and inputs for subjects S1, S2, and S3, respectively.

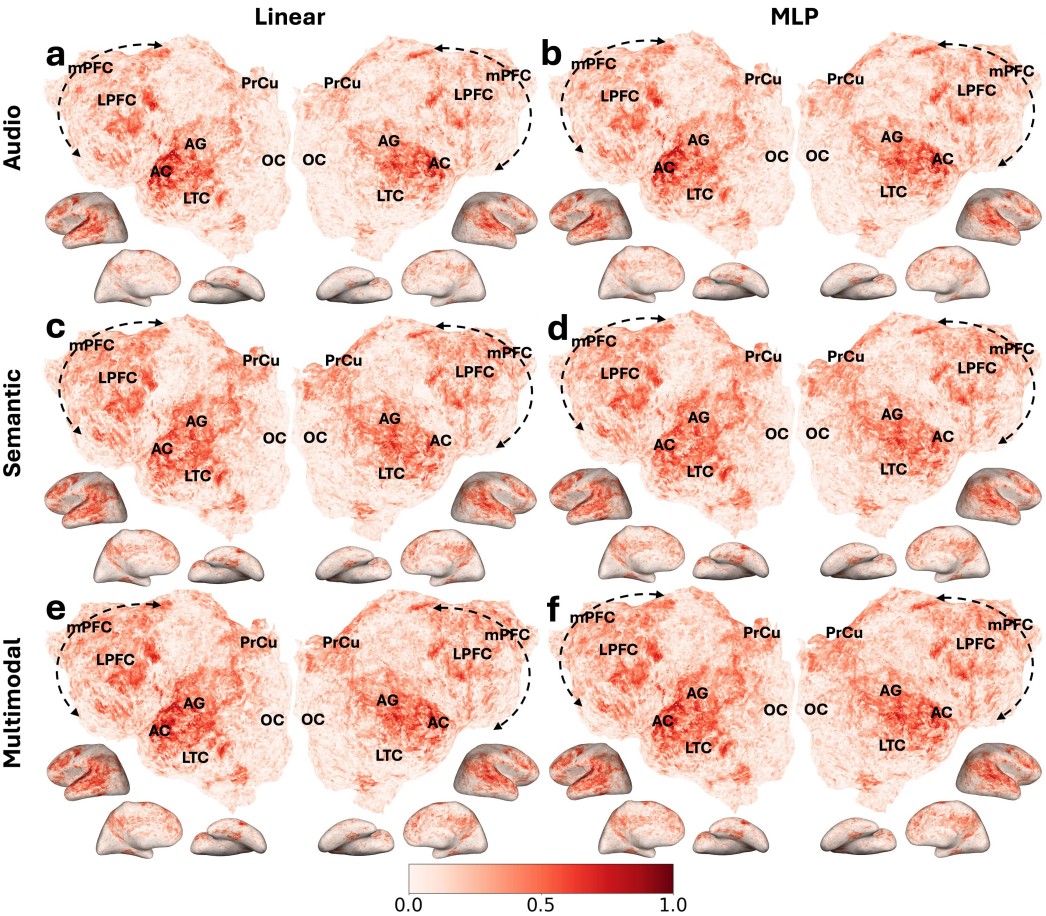

Figure 20: Voxelwise $r$ values for Subject S1 across different input modalities and encoding models. Rows show audio-only (a,b), semantic-only (c,d), and multimodal (e,f) inputs. Columns compare Linear (left) and MLP (right) encoders. Warmer colors indicate higher prediction accuracy.

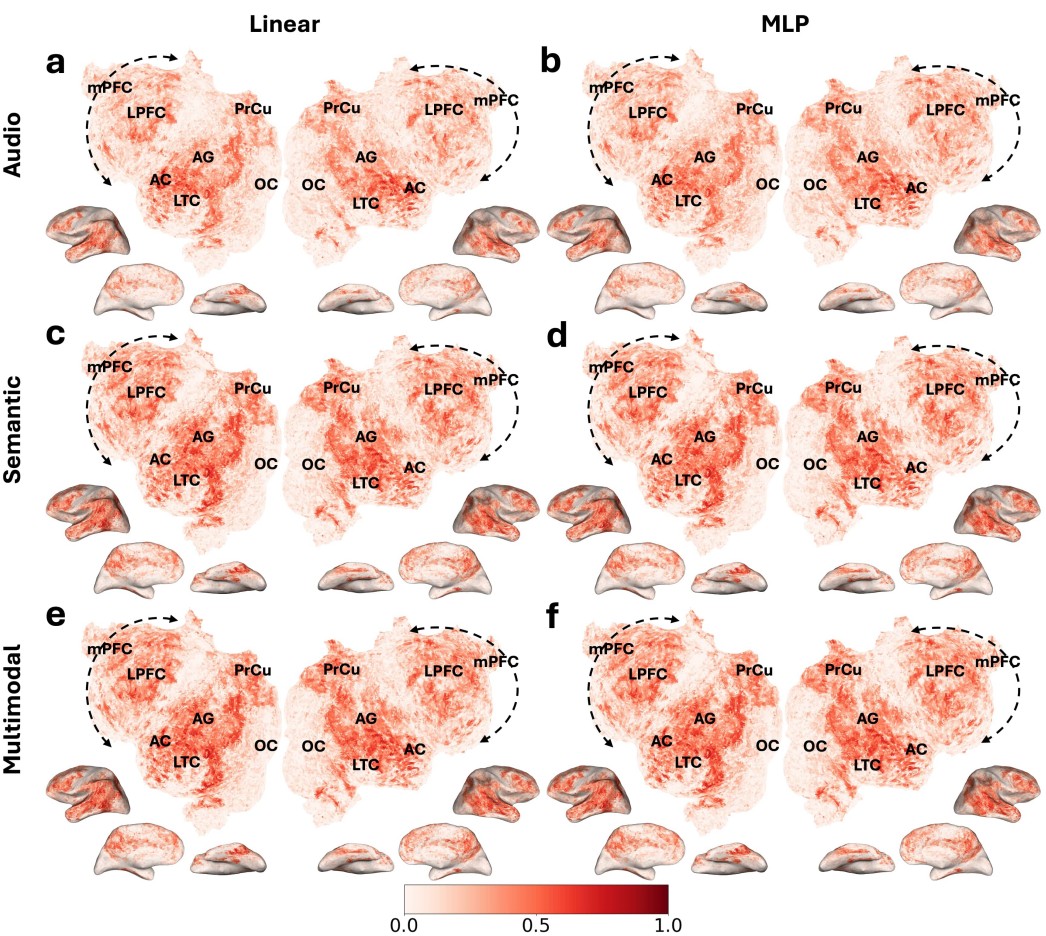

Figure 21: Voxelwise $r$ values for Subject S2 across different input modalities and encoding models. Rows show audio-only (a,b), semantic-only (c,d), and multimodal (e,f) inputs. Columns compare Linear (left) and MLP (right) encoders. Warmer colors indicate higher prediction accuracy.

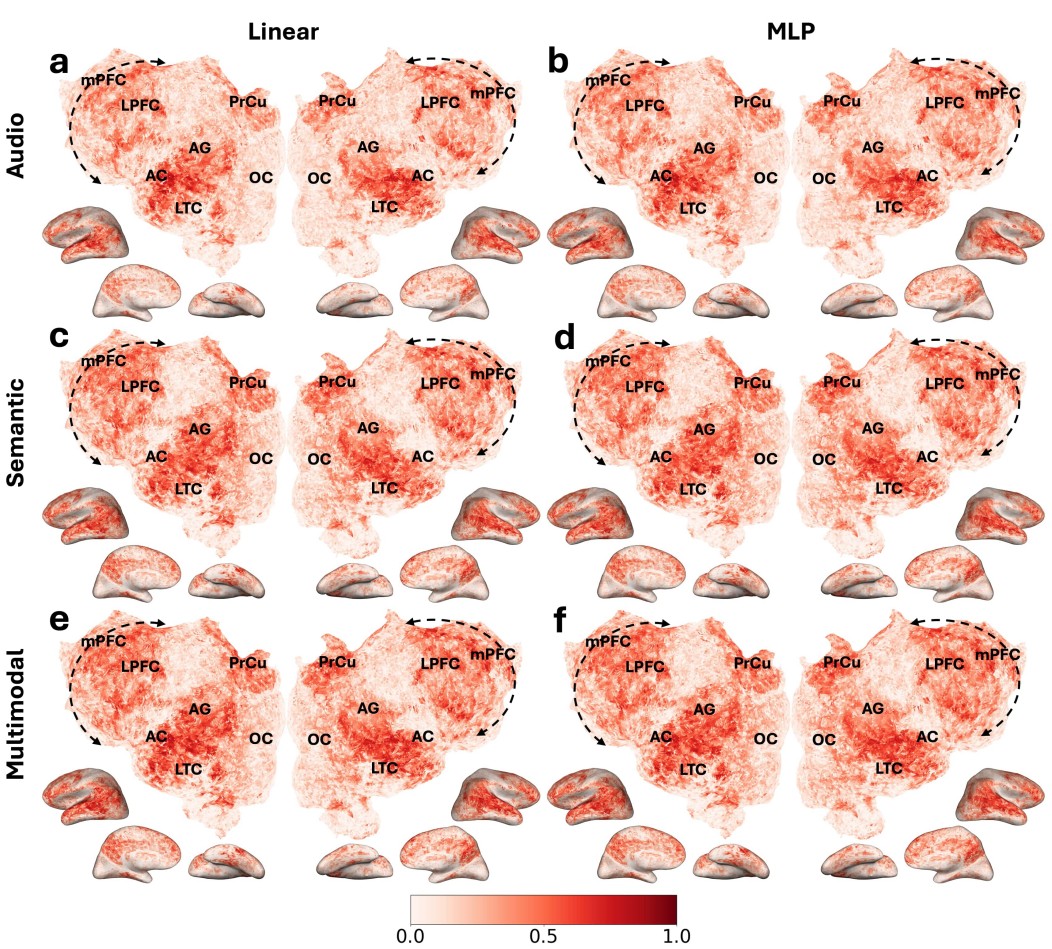

Figure 22: Voxelwise $r$ values for Subject S3 across different input modalities and encoding models. Rows show audio-only (a,b), semantic-only (c,d), and multimodal (e,f) inputs. Columns compare Linear (left) and MLP (right) encoders. Warmer colors indicate higher prediction accuracy.

### A.8.2  ROI-WISE $r$ VALUES FROM DIFFERENT ENCODING MODELS AND STIMULI

Figure 23 shows the $r$ value for different encoding models and stimuli averaged across subjects.

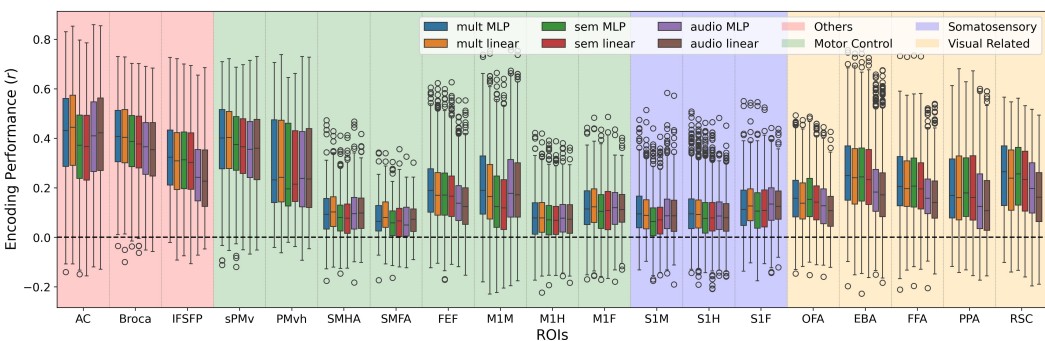

Figure 23: Box plot showing $r$ across different regions of interest (ROIs), where the $r$ values are aggregated over all subjects. *multi* refers to multimodal, and *sem* refers to semantic encoders. ROIs are grouped and color-coded by their functions.

### A.9  IMPROVEMENTS FROM NONLINEARITY

### A.9.1  LAYER-WISE PERFORMANCE INCREASES FROM MLP

Figure 24 shows that MLP improves encoding performance for both language and audio models, regardless of what layer is used for the MLP encoding model.

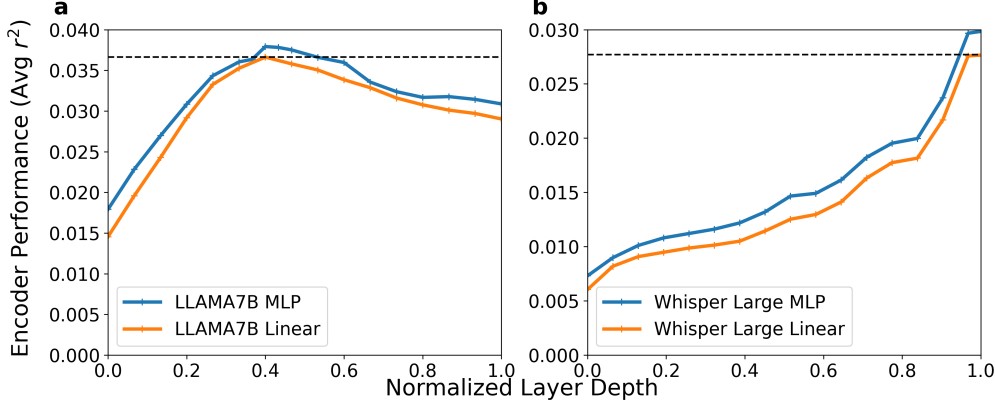

Figure 24: Average voxel-wise $r^2$ values, computed as the mean across three subjects, for each layer of the (a) language (LLAMA7B) and (b) audio (Whisper Large) models. Comparisons are shown between the MLP and linear encoders, and dashed black lines indicate the best performance for linear encoders

### A.9.2  VOXELWISE IMPROVEMENTS FROM MLP ($r$ ANALYSIS)

Figures 25, 26, and 27 each represent the performance improvements in voxelwise correlation values for semantic, audio, and multimodal inputs, respectively, for each subject.

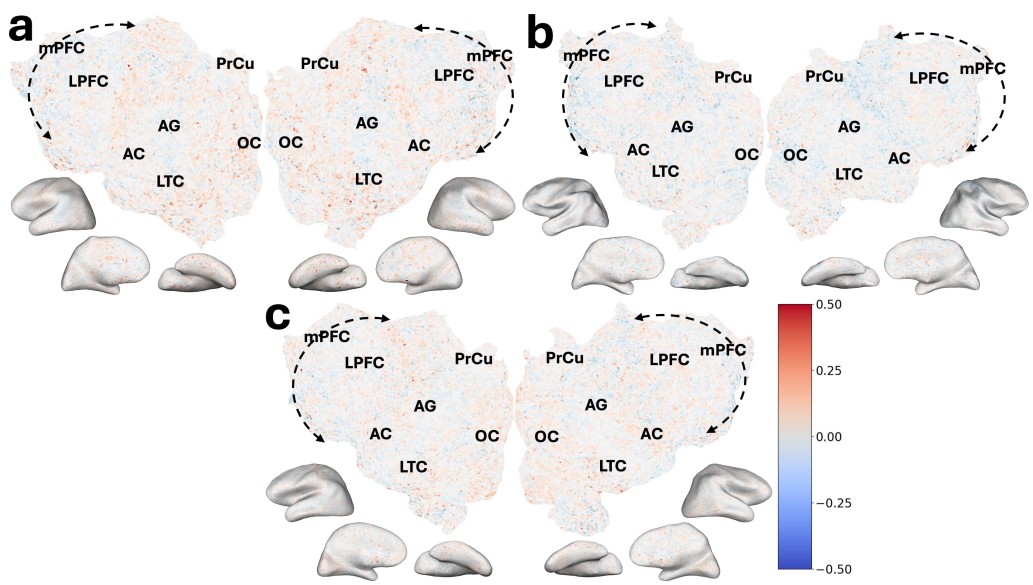

Figure 25: Encoding model performance improvements. (a-c) Voxelwise $\Delta r$ (MLP performance minus linear performance) for semantic input for subjects S1, S2, S3, respectively. Positive values indicate MLP outperformance.

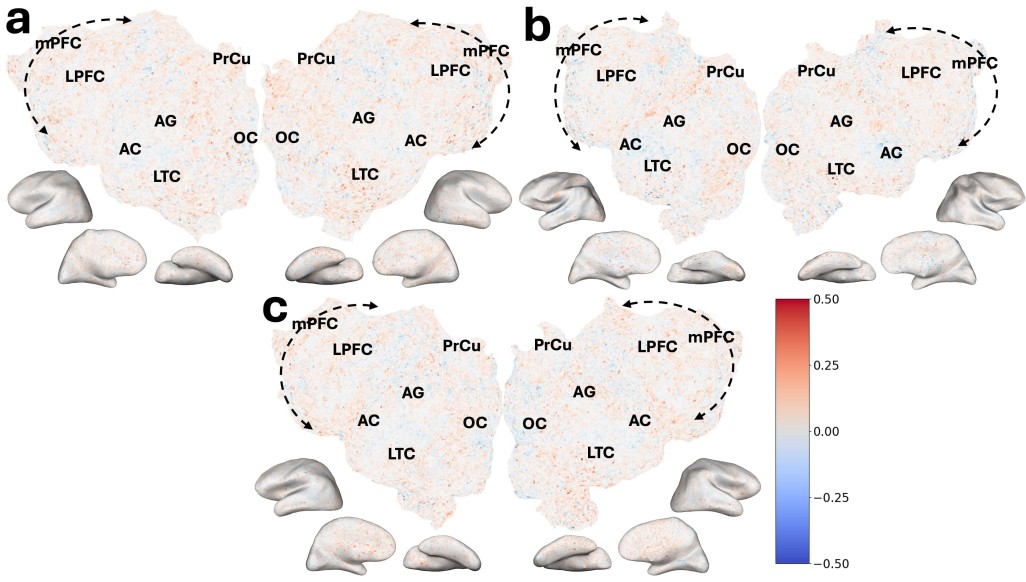

Figure 26: Encoding model performance improvements. (a-c) Voxelwise $\Delta r$ (MLP performance minus linear performance) for audio input for subjects S1, S2, S3, respectively. Positive values indicate MLP outperformance.

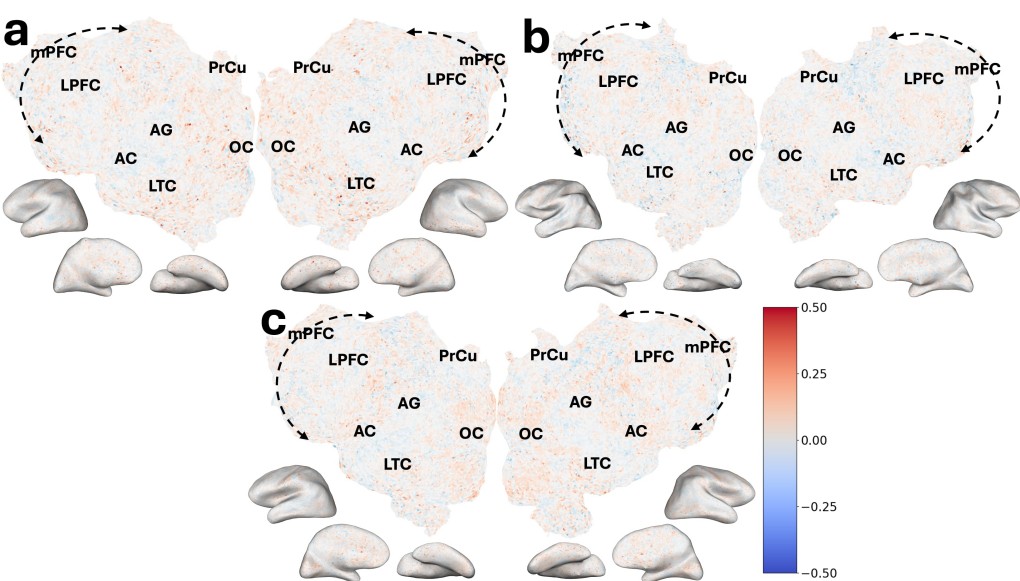

Figure 27: Encoding model performance improvements. (a-c) Voxelwise $\Delta r$ (MLP performance minus linear performance) for multimodal input for subjects S1, S2, S3, respectively. Positive values indicate MLP outperformance.

### A.9.3 VOXELWISE IMPROVEMENTS FROM MLP ($CC_{norm}$ ANALYSIS)

Figures 29, 28, and 30 each represent the performance improvements in voxelwise $CC_{norm}$ values for semantic, audio, and multimodal inputs, respectively, for each subject. The improvements are more pronounced with $CC_{norm}$ compared to $r$ as noise is taken into account.

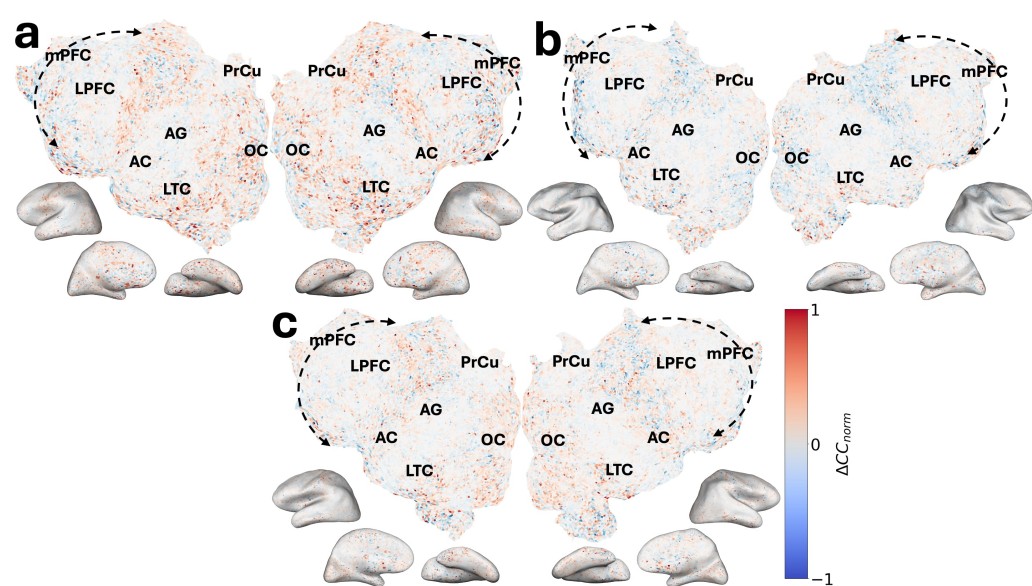

Figure 28: Encoding model performance improvements. (a-c) Voxelwise $\Delta CC_{norm}$ (MLP performance minus linear performance) for semantic input for subjects S1, S2, S3, respectively. Positive values indicate MLP outperformance.

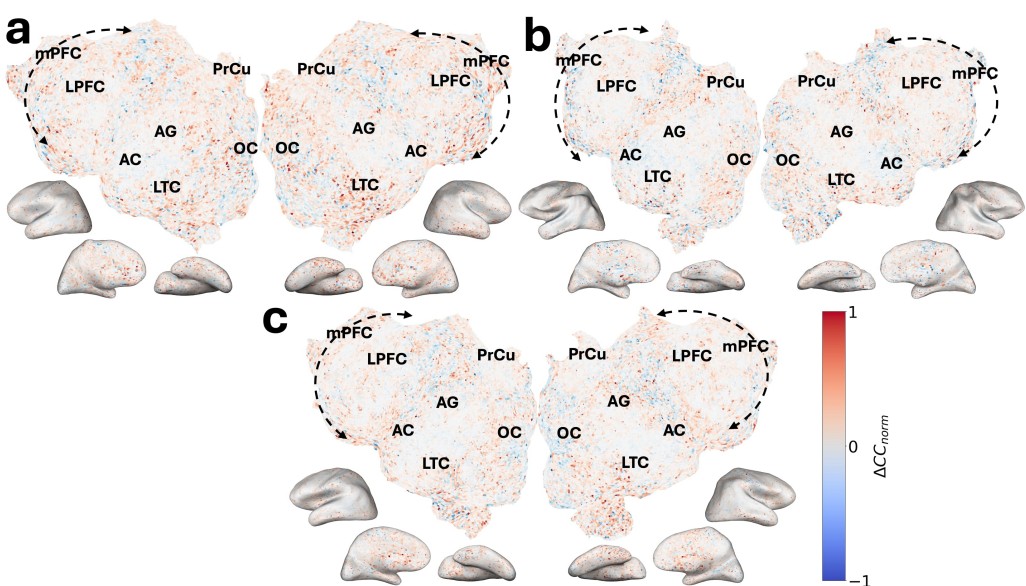

Figure 29: Encoding model performance improvements. (a-c) Voxelwise $\Delta CC_{norm}$ (MLP performance minus linear performance) for audio input for subjects S1, S2, S3, respectively. Positive values indicate MLP outperformance.

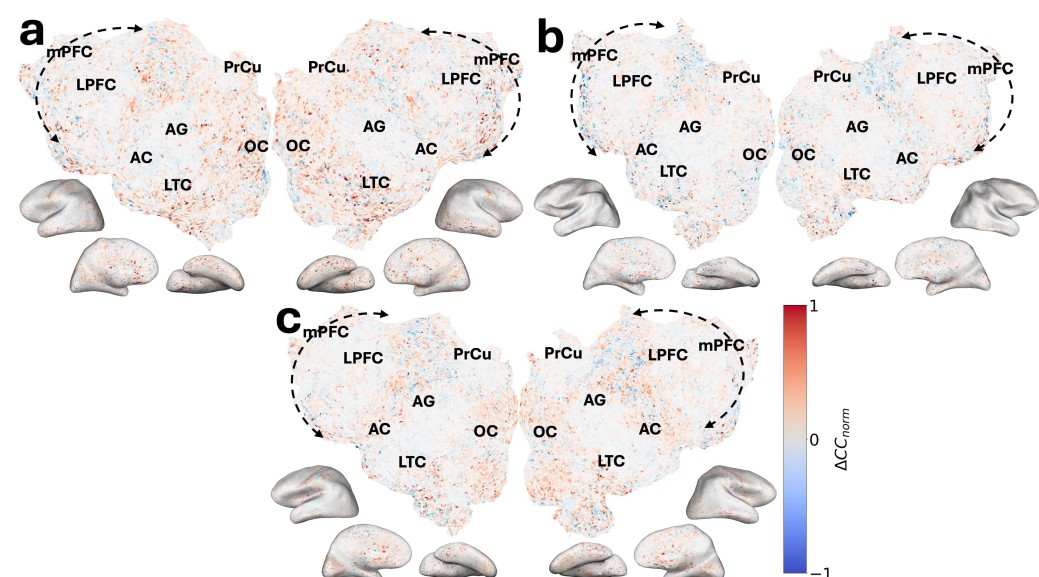

Figure 30: Encoding model performance improvements. (a-c) Voxelwise $\Delta CC_{norm}$ (MLP performance minus linear performance) for multimodal input for subjects S1, S2, S3, respectively. Positive values indicate MLP outperformance.

### A.9.4 BETTER SPATIO-TEMPORAL COMPARTMENTALIZATION OF BRAIN FUNCTION

To compare the performance between Whisper and LLAMA models, we define the Relative Error Difference (RED) for each voxel $v$ at time $t$ as:

$$\text{RED}(v,t) = |f_{\text{semantic}}(v,t) - y(v,t)| - |f_{\text{audio}}(v,t) - y(v,t)|$$

where $f_{\text{semantic}}(v,t)$ is the prediction from the semantic encoding model for voxel $v$ at time $t$, $f_{\text{audio}}(v,t)$ is the prediction from the audio encoding model for voxel $v$ at time $t$, and $y(v,t)$ represents the true value at voxel $v$ and time $t$. A positive RED value indicates that the audio model outperforms the semantic model at that specific voxel and time, while a negative value indicates that the semantic model performs better.

In this analysis, we computed the RED between Whisper and LLAMA models for each voxel $v$ at a given time $t$. For each region of interest (ROI), the average RED is calculated as:

$$\text{RED}_{\text{ROI}}(t) = \frac{1}{N} \sum_{v \in \text{ROI}} \text{RED}(v,t)$$

Where $N$ is the number of voxels in the ROI. The correlation matrices were then computed over these ROI time series for both linear and nonlinear (MLP) encoders (Figure 31 (b, c)). A high correlation between two ROIs indicates that their semantic/audio processing temporal dynamics are similar over time.

For comparison, functional connectivity (FC) was also computed using the average fMRI signal for each voxel (Figure 31 a). Hierarchical clustering was then performed on the correlation matrices, producing the dendrograms in panels (d-f).

As shown in Figure 31, panel (a) does not exhibit meaningful compartmentalization, indicating that the ROIs are not functionally clustered based on FC. However, the correlation matrices derived from RED (panels b, c) demonstrate clear block-diagonal structures, suggesting better functional compartmentalization. The dendrograms in panels (e, f) show that the ROIs cluster according to

their functional roles, where the somatosensory and motor areas, visual areas, and auditory areas are grouped (even lower levels are grouped well (M1H/S1H, M1M/S1M, M1F/S1F, SMHA/SMFA, Broca/sPMv are grouped)) with nonlinear (MLP) models (f) achieving more accurate clustering than linear models (e). Specifically, panel (e) incorrectly clusters SMFA with S1M and M1M, whereas panel (f) correctly clusters SMHA and SMFA together before clustering them with other sensory and motor-related regions.

This study presents a novel approach, as it is the first to use fMRI language encoding models to group ROIs based not only on spatial dynamics but also on their temporal processing dynamics. Traditionally, voxel-wise functional classification or grouping has been the norm in fMRI analysis, focusing solely on spatial relationships. However, here with the help of fMRI encoders, we incorporate both spatial and temporal information, allowing for a more comprehensive view of brain function, especially in the context of semantic and auditory encoding.

In summary, using nonlinear (MLP) models leads to better functional compartmentalization. In fact, modularity Q values further confirm this: FC (a) scored 0.068, linear encoders (b) scored 0.145, and nonlinear encoders (c) scored 0.155, highlighting the improved functional clustering achieved with better encoders.

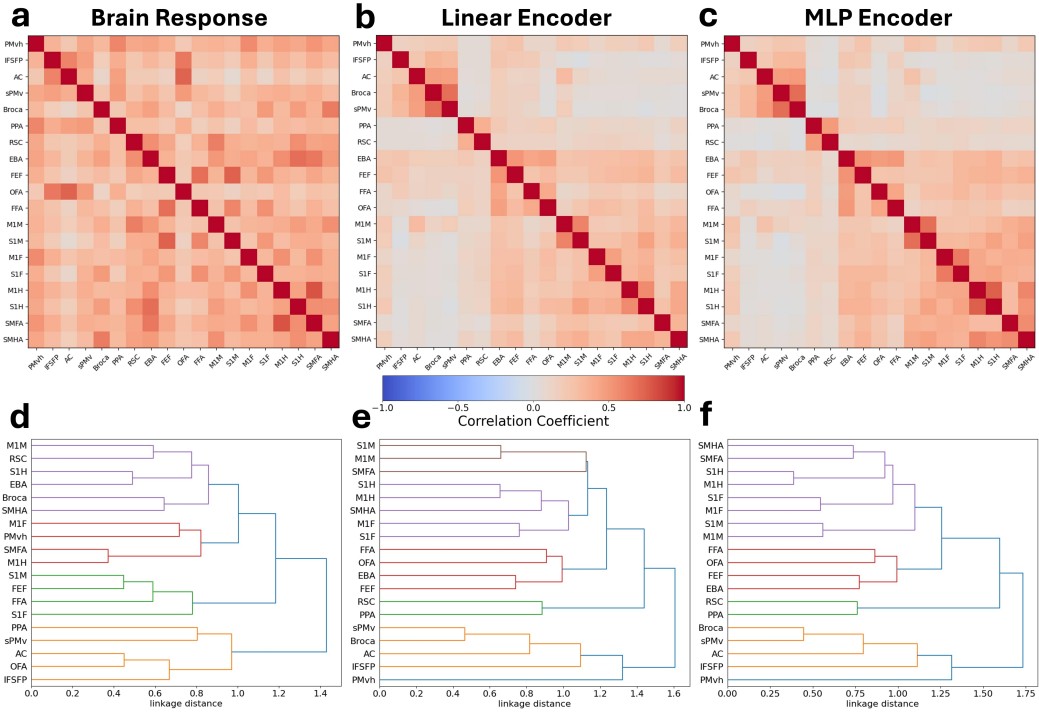

Figure 31: Spatio-temporal clustering based on Relative Error Difference (RED) between semantic and audio encoding models. Panels (a-c) display correlation matrices representing the temporal relationships between regions of interest (ROIs). Panel (a) shows the functional connectivity (FC) matrix, calculated from the average fMRI signals. Panel (b) presents the correlation matrix from Relative Error Difference between Whisper and LLAMA using linear encoders, while panel (c) uses nonlinear (MLP) encoders, showing better functional compartmentalization with stronger block-diagonal structures. Panels (d-f) depict hierarchical clustering dendrograms derived from the correlation matrices in panels (a-c). Panel (d), based on FC, shows no clear compartmentalization of ROIs. Panel (e), based on linear encoders, show almost perfect functional clustering, though with inaccuracies (e.g., SMFA clustered with S1M/M1M). Panel (f), based on nonlinear (MLP) encoders, achieves better functional clustering, correctly grouping motor-related regions. The modularity Q values confirm this improvement: FC (a) scored 0.068, linear encoders (b) scored 0.145, and nonlinear encoders (c) scored 0.155, highlighting the advantage of nonlinear encoders for functional organization.

### A.10 IMPROVEMENTS FROM MULTIMODALITY

#### A.10.1 VOXELWISE IMPROVEMENTS FROM MULTIMODALITY ($r$ ANALYSIS)

This section shows the subject-wise plots of voxelwise $\Delta r$ between multimodal linear/MLP and semantic/audio linear models (Figure 32, Figure 33). We observe consistent patterns of improvement when using multimodal models.

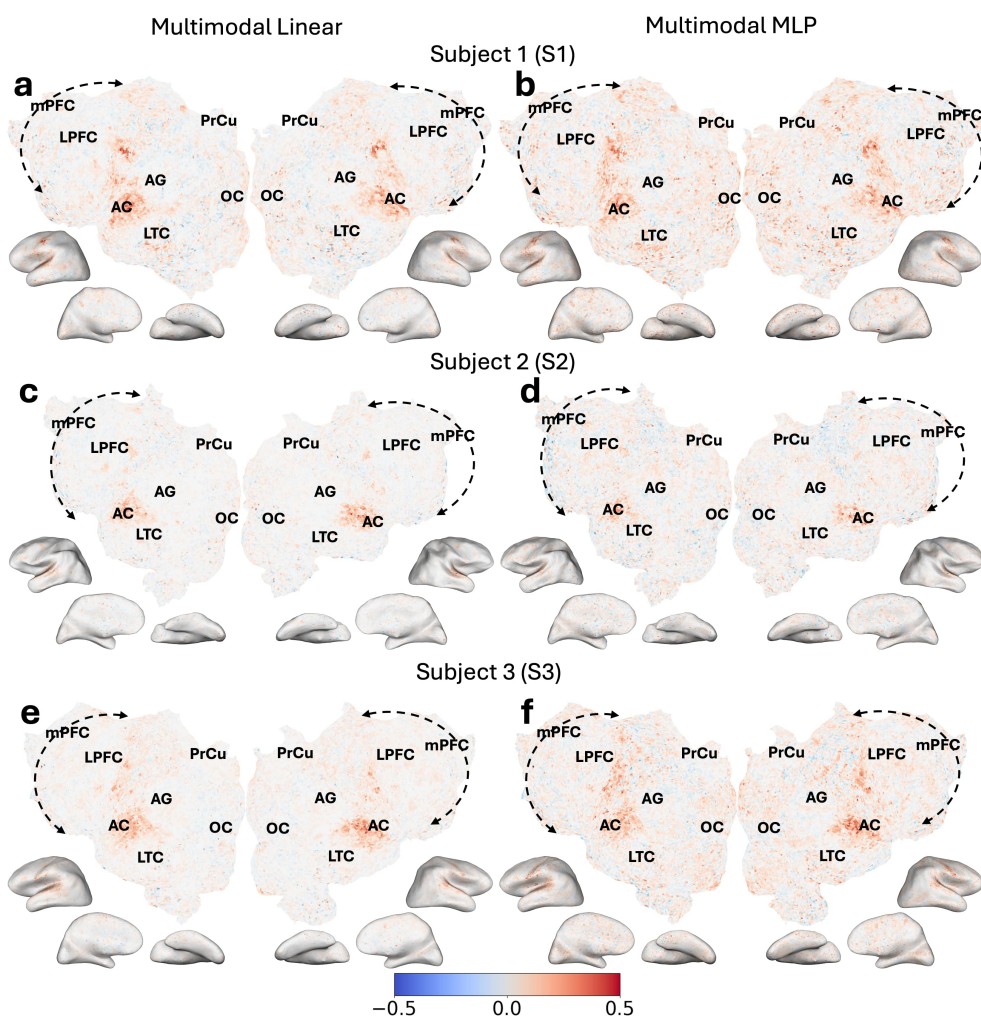

Figure 32: Subject-wise voxelwise $\Delta r$ plots of multimodal models compared to semantic models. Panels (a-f) display voxelwise $\Delta r$ values comparing multimodal and unimodal models across three subjects. Panels a, c, e show the difference between multimodal linear and semantic linear models, while panels b, d, f compare multimodal MLP and semantic linear models. Each row represents a different subject: Subject 1 (S1) in panels a-b, Subject 2 (S2) in panels c-d, and Subject 3 (S3) in panels e-f. Warmer colors indicate regions where the multimodal models outperform the unimodal linear models in prediction accuracy. The spatial patterns highlight enhanced encoding performance in key areas associated with semantic and auditory processing, such as the medial prefrontal cortex (mPFC), angular gyrus (AG), precuneus (PrCu), and lateral temporal cortex (LTC), emphasizing the benefits of multimodal models in capturing complex brain activity.

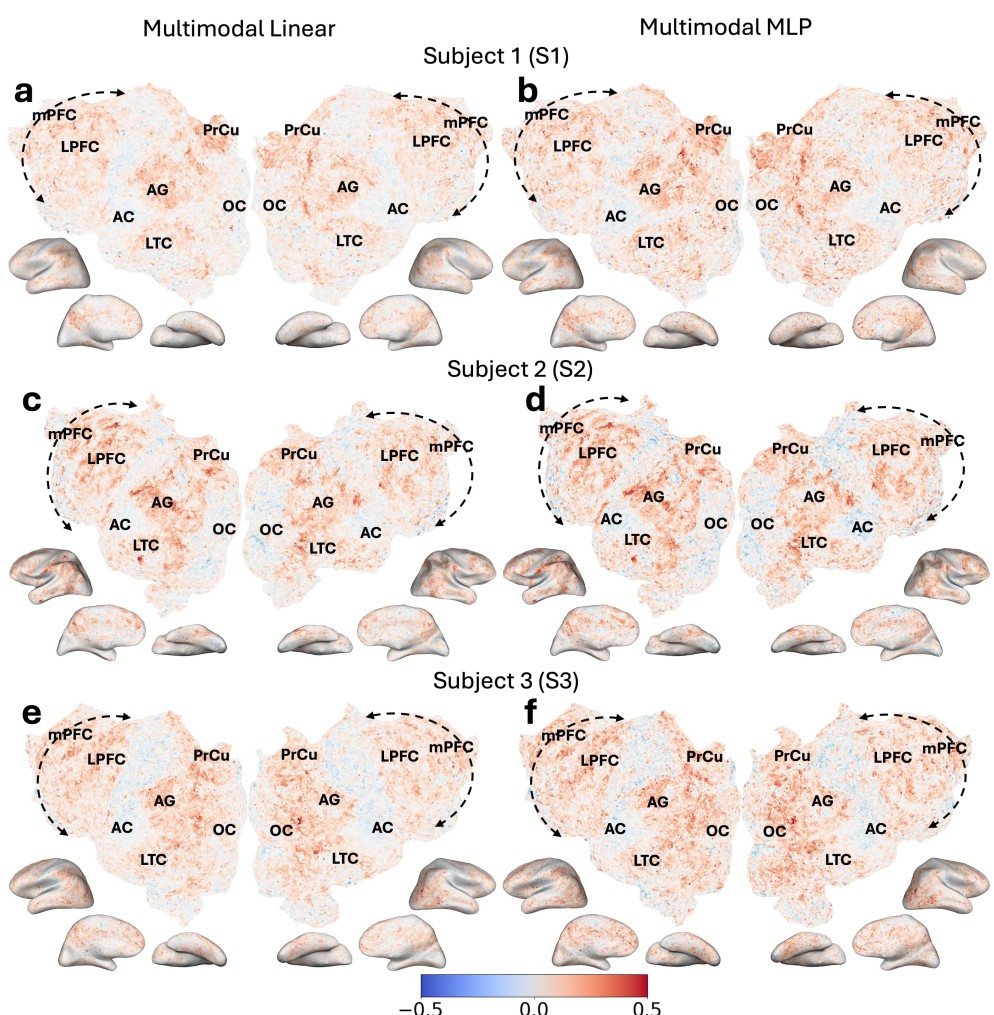

Figure 33: Subject-wise voxelwise $\Delta r$ plots of multimodal models compared to audio models. Panels (a-f) display voxelwise $\Delta r$ values comparing multimodal and unimodal models across three subjects. Panels a, c, e show the difference between multimodal linear and audio linear models, while panels b, d, f compare multimodal MLP and audio linear models. Each row represents a different subject: Subject 1 (S1) in panels a-b, Subject 2 (S2) in panels c-d, and Subject 3 (S3) in panels e-f. Warmer colors indicate regions where the multimodal models outperform the unimodal linear models in prediction accuracy.

### A.10.2 VOXELWISE IMPROVEMENTS FROM MULTIMODALITY ($CC_{norm}$ ANALYSIS)

This section shows the subject-wise plots of voxelwise $\Delta CC_{norm}$ between multimodal linear/MLP and semantic/audio linear models (Figure 35, Figure 35). We observe consistent patterns of improvement when using multimodal models. The improvements are more noticeable with $CC_{norm}$ compared to $r$ as noise is taken into account.

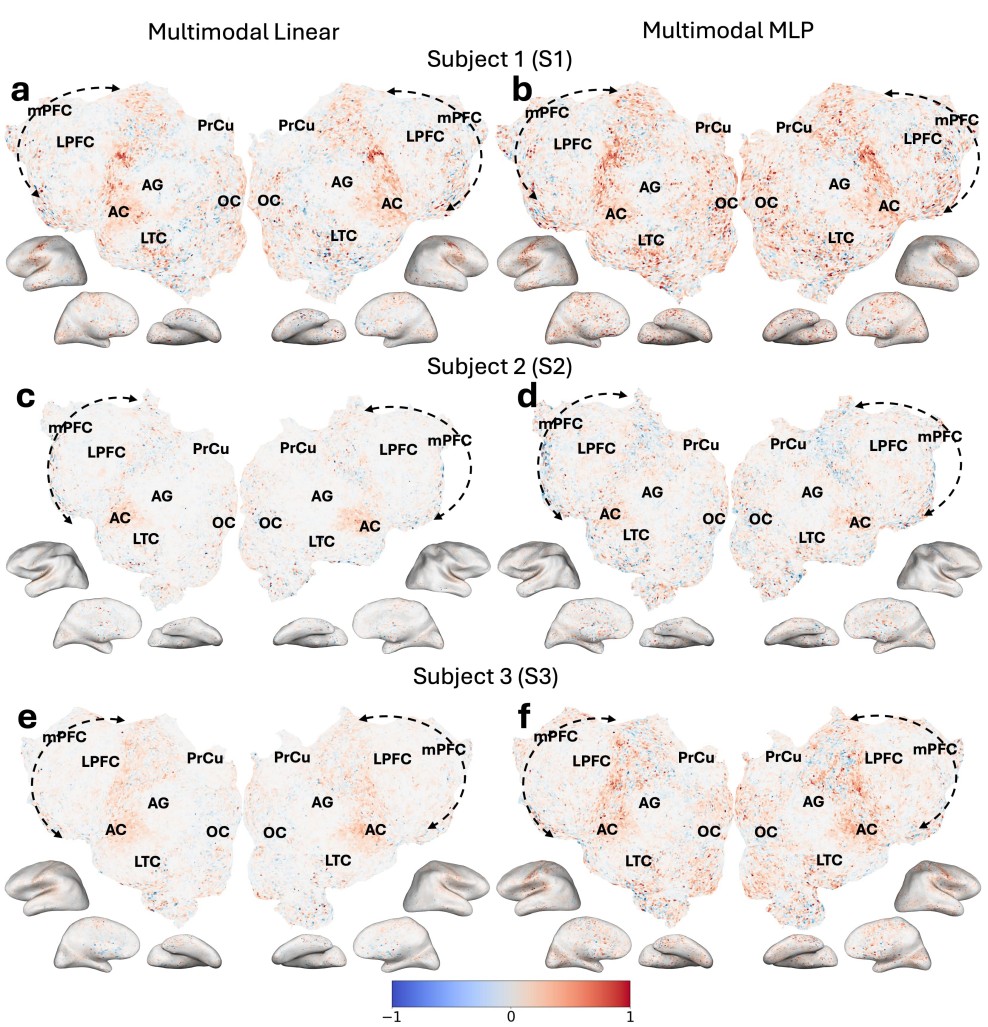

Figure 34: Subject-wise voxelwise $\Delta CC_{norm}$ plots of multimodal models compared to semantic models. Panels (a-f) display voxelwise $\Delta CC_{norm}$ values comparing multimodal and unimodal models across three subjects. Panels a, c, e show the difference between multimodal linear and audio linear models, while panels b, d, f compare multimodal MLP and audio linear models. Each row represents a different subject: Subject 1 (S1) in panels a-b, Subject 2 (S2) in panels c-d, and Subject 3 (S3) in panels e-f. Warmer colors indicate regions where the multimodal models outperform the unimodal linear models in prediction accuracy.

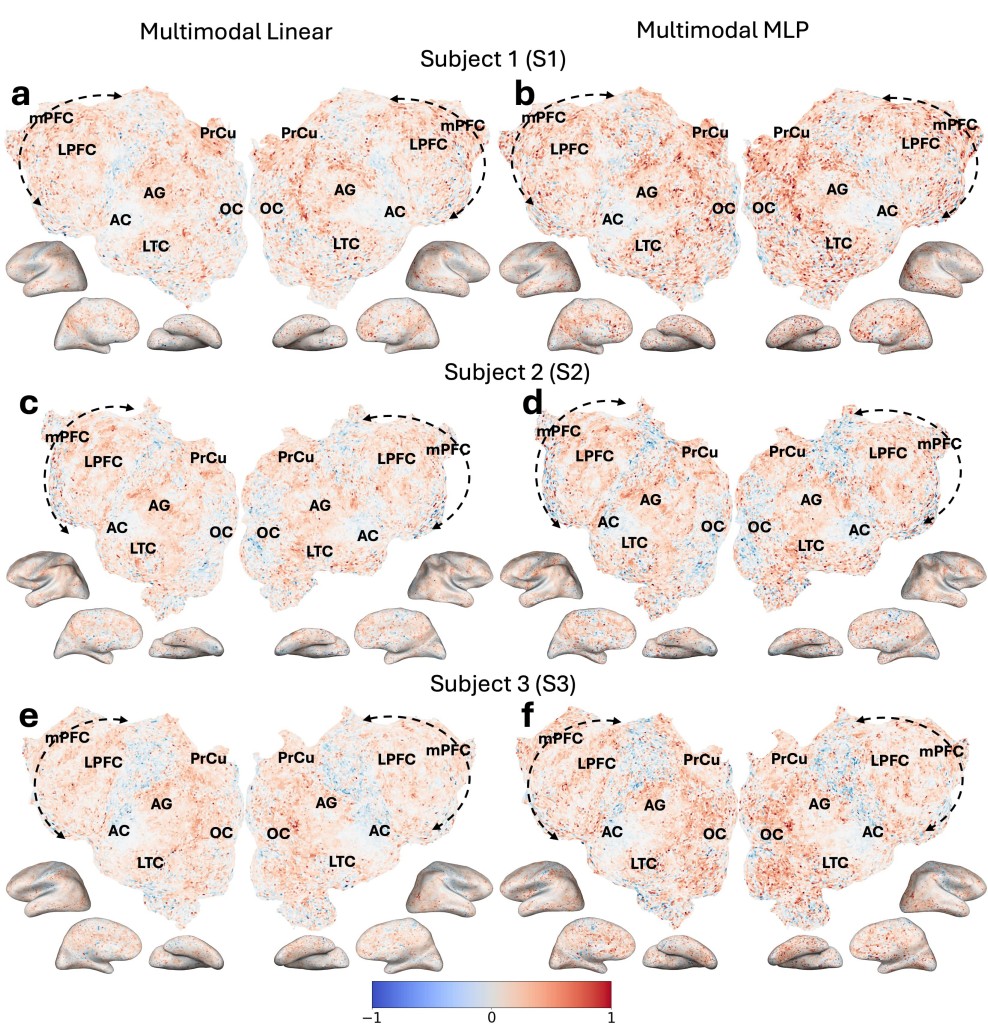

Figure 35: Subject-wise voxelwise $\Delta CC_{norm}$ plots of multimodal models compared to audio models. Panels (a-f) display voxelwise $\Delta CC_{norm}$ values comparing multimodal and unimodal models across three subjects. Panels a, c, e show the difference between multimodal linear and audio linear models, while panels b, d, f compare multimodal MLP and audio linear models. Each row represents a different subject: Subject 1 (S1) in panels a-b, Subject 2 (S2) in panels c-d, and Subject 3 (S3) in panels e-f. Warmer colors indicate regions where the multimodal models outperform the unimodal linear models in prediction accuracy.

## A.11 ROI PREDICTIONS IMPROVEMENTS FROM MULTIMODALITY

This section shows the ROI-wise improvements from using multimodal models (Figure 36)

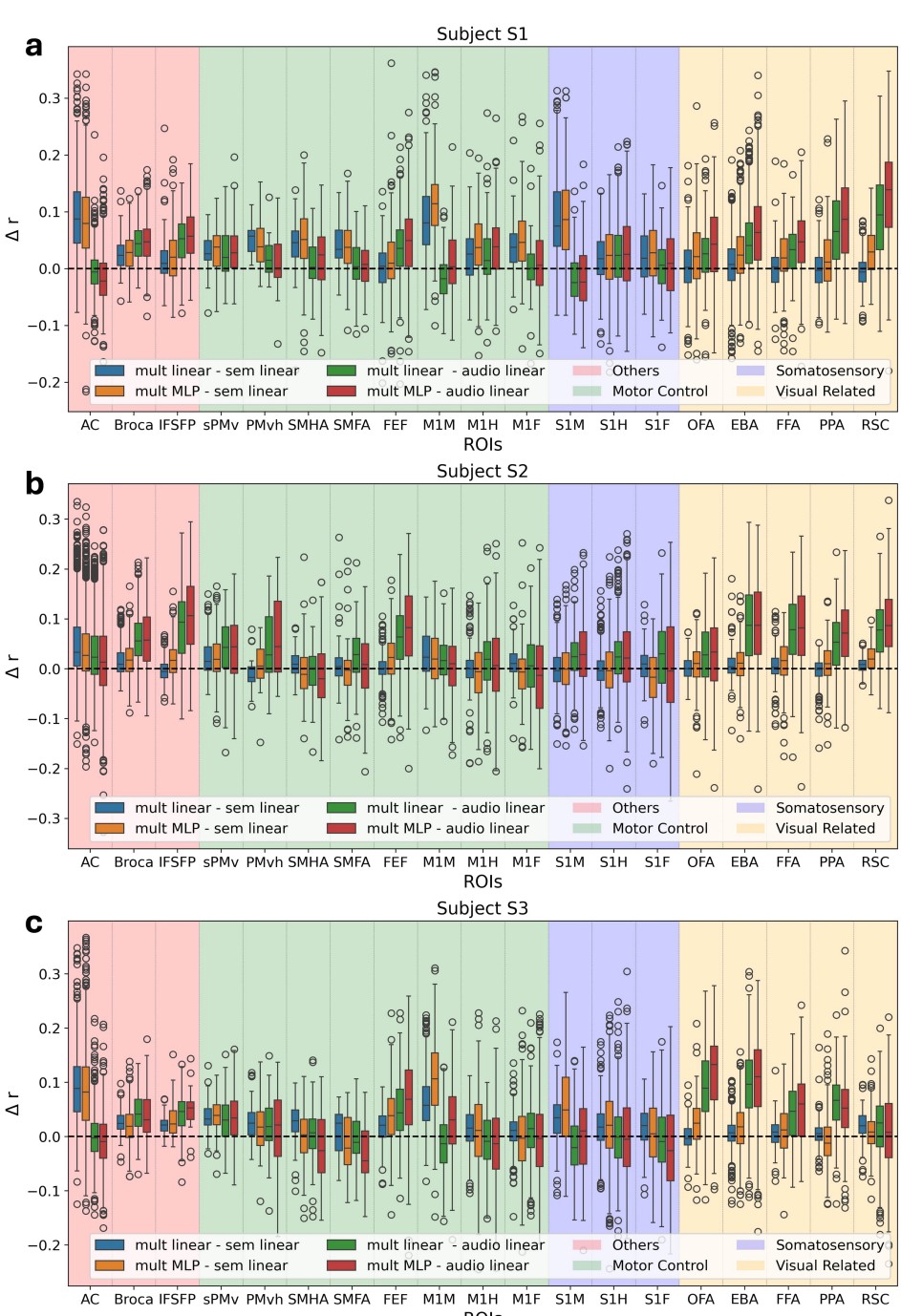

Figure 36: Subject-wise boxplots of performance differences ($\Delta r$) across different ROIs. The comparisons are made between different stimuli and encoding models: multimodal linear and multimodal MLP (mult MLP) models are compared against semantic (sem) and audio linear models. The ROIs are grouped into functional categories.

## A.12 IMPROVEMENTS FROM NONLINEARITY AND MULTIMODALITY

### A.12.1 VOXELWISE IMPROVEMENTS FROM DIMLP, AND ADDITIONAL IMPROVEMENTS FROM MLP ($r$ ANALYSIS)

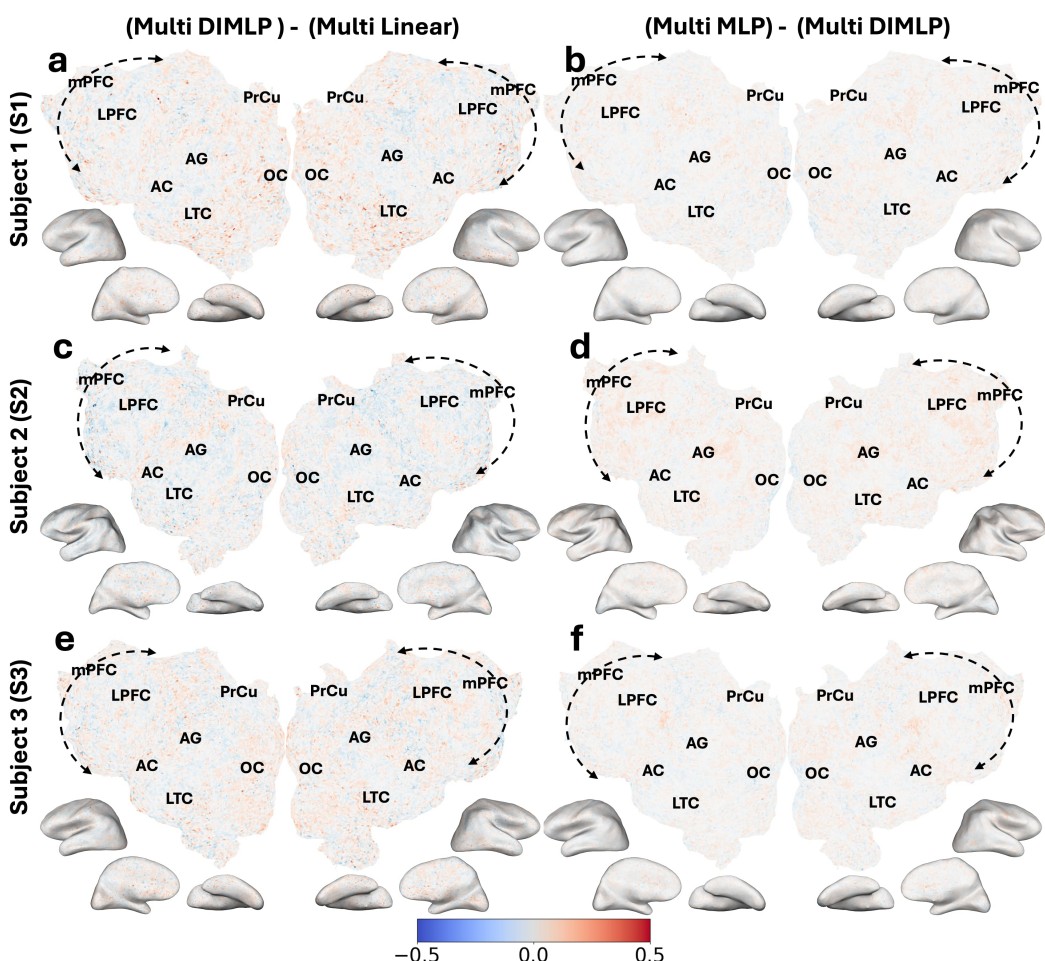

Figure 37: Nonlinearity Enhances Multimodal fMRI Predictions. Panels (a, c, e) show the voxelwise $\Delta r$ values (DIMLP minus linear model), illustrating the improvements achieved through nonlinear processing within each modality, while largely limiting cross-modal interactions. Panels (b, d, f) display voxelwise $\Delta r$ values (Multi MLP minus Multi DIMLP), highlighting the additional benefits of allowing nonlinear interactions between modalities ("Multi" denotes Multimodal). Each row represents the same subject: Subject 1 (S1) in panels a-b, Subject 2 (S2) in panels c-d, and Subject 3 (S3) in panels e-f. Warmer colors indicate regions where the nonlinear models outperform linear models.

### A.12.2 VOXELWISE IMPROVEMENTS FROM DIMLP, AND ADDITIONAL IMPROVEMENTS FROM MLP ($CC_{norm}$ ANALYSIS)

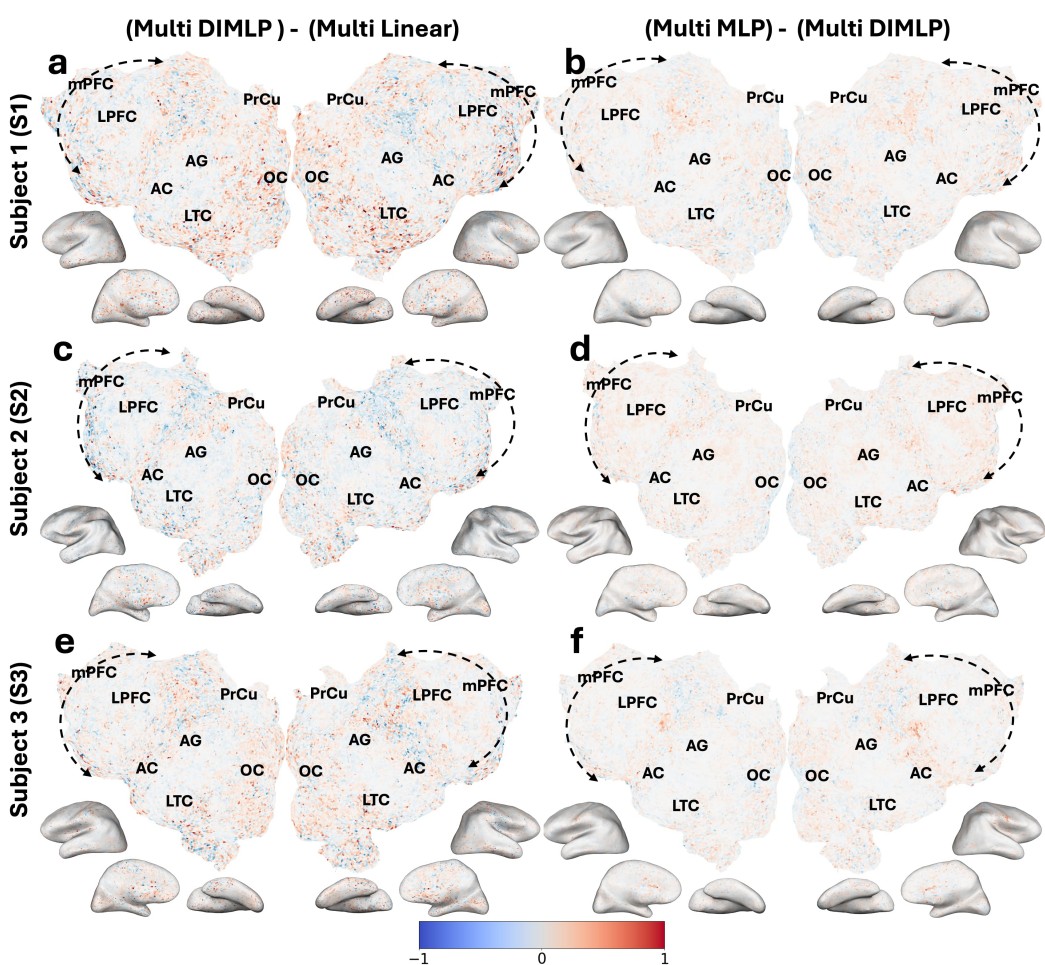

Figure 38: Nonlinearity Enhances Multimodal fMRI Predictions. Panels (a, c, e) show the voxelwise $\Delta CC_{norm}$ values (DIMLP minus linear model), illustrating the improvements achieved through nonlinear processing within each modality, while largely limiting cross-modal interactions. Panels (b, d, f) display voxelwise $\Delta CC_{norm}$ values (Multi MLP minus Multi DIMLP), highlighting the additional benefits of allowing nonlinear interactions between modalities ("Multi" denotes Multimodal). Each row represents the same subject: Subject 1 (S1) in panels a-b, Subject 2 (S2) in panels c-d, and Subject 3 (S3) in panels e-f. Warmer colors indicate regions where the nonlinear models outperform linear models.

### A.12.3 ROI-WISE IMPROVEMENTS OF MULTIMODAL DIMLP AND MLP FROM MULTIMODAL LINEAR MODEL

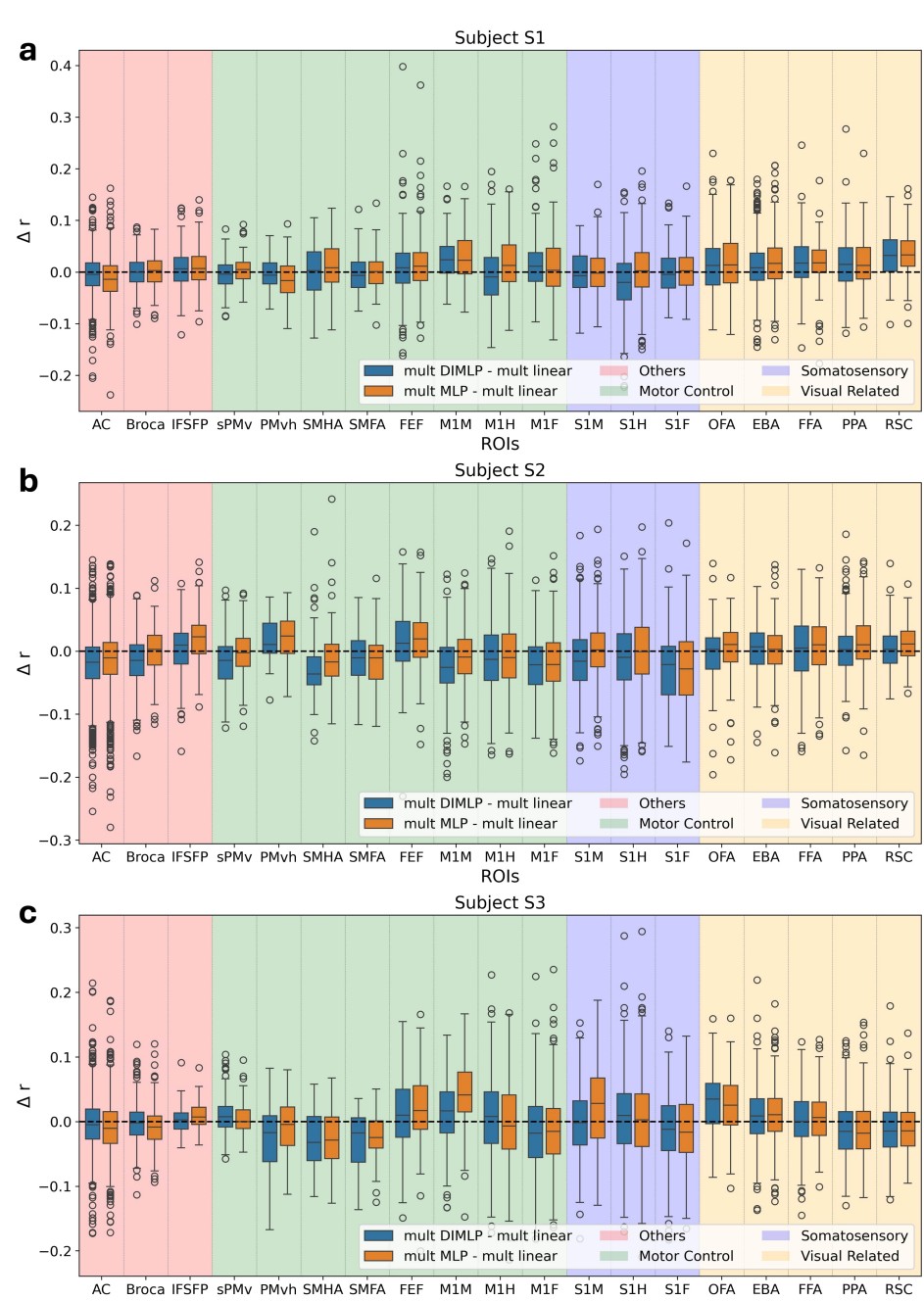

Figure 39: Subject-wise boxplots of voxel-wise differences ($\Delta r$) across different ROIs. The comparisons are made between different encoding models: multimodal MLP and multimodal DIMLP models are compared against multimodal linear models. The ROIs are grouped into functional categories.

## A.13 PERFORMANCE OF MULTIMODAL MLP MODEL WHEN MIXING DIFFERENT LAYERS

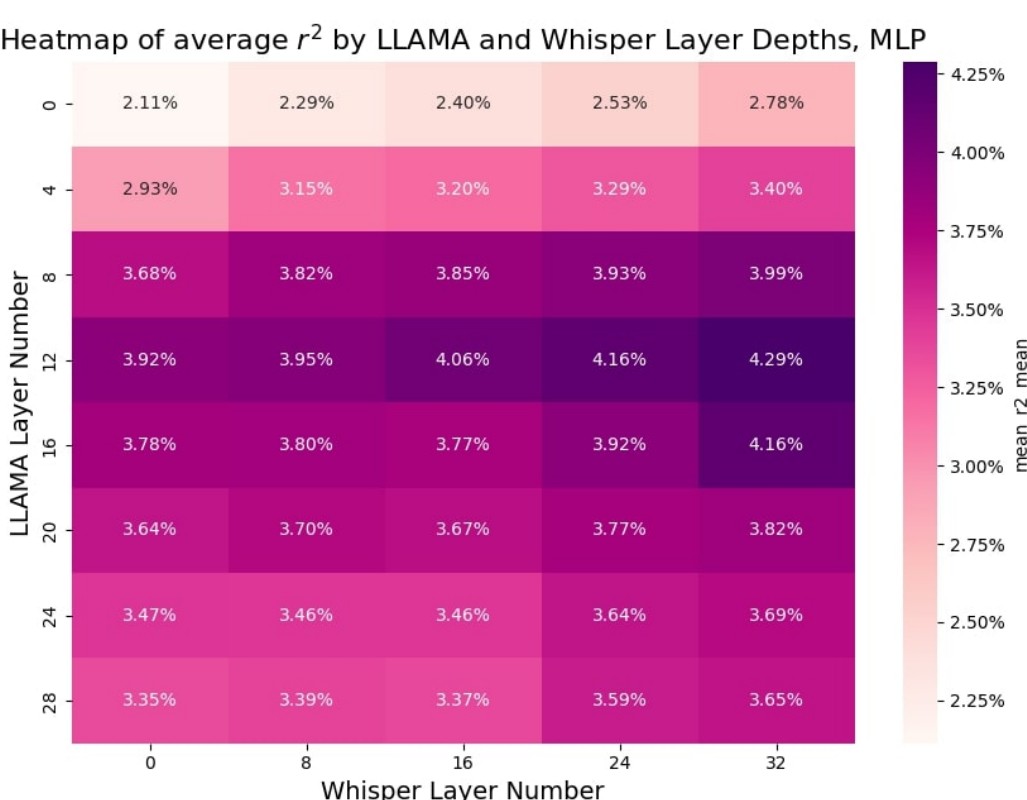

Figure 40: Heatmap showing average $r^2$ values for different combinations of LLAMA and Whisper layer depths using an MLP encoder. Darker colors represent higher performance, with the best results obtained when the best layers in the respective uni-modal encoding models were used.

