# OpenReview forum: "MIND THE GAP: ALIGNING THE BRAIN WITH LANGUAGE MODELS REQUIRES A NONLINEAR AND MULTIMODAL APPROACH"
_ICLR.cc/2025/Conference — Submitted to ICLR 2025_

### Official Review · Reviewer_6SyF · 2024-10-17

**Soundness:** 3
**Presentation:** 4
**Contribution:** 3
**Rating:** 8
**Confidence:** 5

**Summary:**

The paper examines various techniques for improving the state-of-the-art on the task of predicting fMRI-measured BOLD response to language. They propose a multimodal method of combining representations from audio and language models to outperform an established baseline. The method utilizes PCA to reduce the dimensionality of the responses prior to model fitting, as well as an MLP to learn nonlinearities of the data.

**Strengths:**

The proposed method substantially improves prediction performance on a task of extreme importance to neuroscience and neuroengineering. Encoding model prediction performance is directly related to decoding performance, and enables the use of in-silico testing. Nonlinear approaches to brain modelling are quite understudied and deserve more research.

I have currently marked the paper as a 5 owing to the concerns below, however if they are all successfully addressed or revised then I am willing to raise the score.

**Weaknesses:**

There are some omissions in the paper that I would hope to see addressed. For example, the "All voxels" condition seems to only have been tested for the Linear case as in the original cited baseline. This is somewhat puzzling, because for two of the conditions, both the semantic and the multimodal condition, the "PCA" condition actually performs worse than the "all voxels" condition. It's not clear to me why the authors use PCA here. The inclusion of a MLP+all voxels setting in the ablation table would be useful to clarify the reasoning for this choice.

The authors claim that their results do not support the claim in the original baseline (Antonello et al.) that encoding performance scales with model size for language models (line 964). This was somewhat surprising to me, as this relationship has been reproduced by Hong et al. as well as others. Looking deeper, the author's claim seems somewhat undersupported, as the authors only test models with no less than 7 billion parameters, even though the vast majority (>80%) of the improvement in the original baseline is claimed to occur when scaling from models of size 125 million to models of 13 billion parameters. If the authors intend to make the claim that scaling relationships do not exist for encoding models, they should test smaller feature extractors (~125M parameters) to fully validate their argument.

Additionally, the authors claim that their multimodal approach can explain regions outside of M1M and AC, however as far as I can tell the only regions that are red in the comparison flatmap (Figure 1a) are M1M and AC, so I'm not really sure where this is coming from. They specifically claim that medial prefrontal cortex and angular gyrus are better predicted, however, neither of these regions are red in the provided flatmap, and they are not included in the boxplots either. Please clarify this discrepancy.

I also find it somewhat odd that the multimodal model is only compared to the semantic baseline. It seems that the more appropriate comparison point would be the referenced stacked regression model when evaluating their MLP + PCA multimodal approach. The primary claim of a "17.2% increase in mean correlation across the cortex compared to the prior linear encoding models" therefore seems somewhat misleading.

The inclusion of motor regions (except for M1M) in the analyses are somewhat strange - these are typically poorly predicted overall and have little to do with language processing. It would be good in general for the authors to include at least one figure that shows absolute prediction performance rather than relative prediction performance. This is especially important because the provided flatmaps are changes in correlation coefficient and not r^2. An improvement in prediction performance from r=0.1 to r=0.2 explains substantially less additional variance than improvement from r=0.7 to r=0.8.

Also, is r^2 computed as r*|r| here? If so, this should be mentioned somewhere.

**Questions:**

See above.

---

> ### Author Response · Authors · 2024-11-25
>
> We appreciate the reviewer's insightful comments and will address each point below:
>
> **1. Testing MLP + All Voxels encoding model**
>
> We have now conducted the MLP + all voxels experiment and included the results in the revised Table 1 of the paper. This analysis shows that the MLP + all voxels model performs consistently worse than the MLP + PCA model. The model's poor performance when using all voxels (80-90,000) compared to using only the top 512 principal components (PCA) is attributed to overfitting, as the significantly larger number of output neurons (voxels) in the final layer increases the model's capacity to memorize the training data rather than learning generalizable features. This overfitting leads to poor generalization to unseen data. The superior performance of the PCA-reduced model validates the decision to use dimensionality reduction (PCA) before feeding data to the MLP. In essence, reducing the dimensionality prevents the model from becoming too complex and prone to overfitting.
>
>
> **2. LLM Model Scaling**
>
> We thank the reviewer for pointing out this potential misinterpretation. We have updated the text (Lines 168-171 and Appendix A.4) to better convey that:
> - Our observation specifically pertains to models ≥7B parameters, where we found performance plateaus regardless of model size or training data size (model version)
> - This aligns with the reviewer’s comment as well as recent findings showing scaling benefits diminish in larger models (Bonnasse-Gahot 2024). (Unfortunately, we failed to find the Hong paper the reviewer mentioned; providing the full citation would allow us to incorporate it into our discussion)
> - Our analysis was limited to models ≥7B parameters as this was the smallest available LLaMA model during our experiments.
>
> **3. Regarding the improvements in only M1M and AC**
>
> We apologize for the misidentification in our initial description. Upon closer inspection, we found that while subtle, the improvements appear in the occipital cortex and paracentral lobule, not the medial prefrontal cortex and angular gyrus as originally stated. To better visualize these subtle changes, we have added voxel-wise ΔCC_norm plots in Appendix A.10.2., where the differences are more visible due to CC_norm's accounting for noise ceiling effects. Regarding the boxplots, we did not have enough time to map the regions from structural MRI (T1) to pycortex. However, we believe the CC_norm visualizations demonstrate these improvements clearly enough without requiring additional quantitative analysis.
>
> **4. Regarding the absolute prediction performance figure**
>
> Thank you for the suggestion. In Appendix A.8.1. and A.8.2., we have included both voxelwise and ROI-wise absolute prediction performance of various encoding models with different stimuli.
>
> **5. Regarding Stacked Regression**
>
> While this would indeed be a valuable baseline, our attempts to replicate the stacked regression from Antonello et al. 2024 were unsuccessful due to NaN values in their provided weights. We are in communication with the authors to resolve this issue and will incorporate this comparison once we receive corrected weights.
>
> **6. Inclusion of motor regions except for M1M in the boxplot analysis**
>
> We agree with the reviewer that other motor regions are known to have little to do with language processing. However, we included these regions for the following reasons :
> * While less significant, other motor regions such as the sPMv, PMvh, and FEF show meaningful improvements (Δr) from incorporating multimodality.
> * While these regions do not have big Δr in the group-aggregated analysis in the main figure, in a subject level analysis, some of the regions show meaningful differences (Appendix A.11).
> * The spatio-temporal compartmentalization clustering analysis in Appendix A.9.4. shows that the other motor ROIs are also well clustered. This implies that these ROIs are not completely orthogonal to language processing.
>
> **7. How r^2 is computed**
>
> Thank you for pointing it out. While we did write that it is computed as |r|*r in the captions of Table 1, we will add it to the main text too for clarity (Line 250)
>
>
> **Reference** :
>
> **Bonnasse-Gahot 2024** : Laurent Bonnasse-Gahot and Christophe Pallier. fmri predictors based on language models of increasing complexity recover brain left lateralization. arXiv preprint arXiv:2405.17992, 2024.
>
> **Antonello 2024** : Richard Antonello, Aditya Vaidya, and Alexander Huth. Scaling laws for language encoding models in fmri. Advances in Neural Information Processing Systems, 36, 2024.

---

> > ### Comment · Reviewer_6SyF · 2024-11-25
> > **Updated my score**
> >
> > I have updated my score to reflect the largely satisfactory answers I have received here. I think the paper is practically useful and has value as a moderate impact improvement over the SOTA.

---

> ### Author Response · Authors · 2024-11-26
>
> Thank you for considering our responses and the updated assessment.
>
> Upon further discussion with the author of Antonello et al. (2024), we identified that while our work and Antonello et al. use three test stories for overall encoding performance, their stacked regression results specifically used two of these stories for validation (voxel-wise selection between stacked regression and semantic model) and only one story ("wheretheressmoke") for final testing. To ensure fair comparison, we will:
>
> * Add a new appendix section showing encoding performance on the single test story for all models in Table 1, including stacked regression.
> * Report our model's performance with and without their voxel selection method.
> * Quote the improvements of our models compared to the stacked regression using the single test story.
>
> **Reference**:
> * Antonello, R., Vaidya, A., & Huth, A. (2024). Scaling laws for language encoding models in fMRI. Advances in Neural Information Processing Systems, 36.

---

> ### Author Response · Authors · 2024-11-29
>
> We have now completed the promised comparison with the stacked regression baseline (Appendix A.3). Our multimodal MLP achieves a 14.4% improvement in $CC_{norm}$ and 7.7% improvement in average r² when evaluated on the single test story ("wheretheressmoke"). This analysis includes evaluations both with and without their voxel selection method, providing a complete and fair comparison framework.
>
> Additionally, we've added detailed variance partitioning analysis (Appendix A.7) that reveals joint audio-semantic features dominate the explainable variance across most cortical regions (approximately 65% of significantly predicted voxels). This supports our claim that multimodal integration occurs throughout the cortex rather than being limited to specific regions. The analysis includes:
> * Voxel-wise variance partition results showing individual feature contributions
> * Voxel-wise plots highlighting the largest variance partition for each voxel
> * ROI-wise Venn diagrams illustrating the distribution of feature contributions
>
> These analyses further validate our findings about the distributed nature of multimodal processing in the brain.
>
> *Note: We have updated our previous comments to reflect the revised appendix/figure numbering in the updated manuscript.*

---

### Official Review · Reviewer_V4NE · 2024-10-27

**Soundness:** 2
**Presentation:** 3
**Contribution:** 2
**Rating:** 5
**Confidence:** 5

**Summary:**

There is a large body of research focused on measuring the similarity between language processing in the brain and in language models. Recent studies have shown that representations from Transformer-based language models exhibit a higher degree of alignment with brain activity in language regions when using voxel-wise encoding models with linear mapping. However, the authors note that relying solely on linear mappings from language model representations falls short in capturing the complexity of nonlinear auditory signals and linguistic processing in the brain.
The primary aim of this paper is to investigate this question by using a nonlinear, multimodal encoding model that incorporates both audio and linguistic features to predict brain activity. To achieve this, the authors build both non-linear and linear voxel-wise encoding models and compare the encoding performance between representations from two language models (LLaMA:text-based and Whisper:speech-based) and brain recordings. Additionally, the authors create encoding models by combining the two representations and measure brain alignment. The experimental results demonstrate that there is a significant improvement in prediction accuracy with non-linear encoding models, achieving a 17.2% increase in mean correlation across the cortex compared to the prior linear encoding models.

**Strengths:**

1. The exploration of nonlinear voxel-wise encoding models, as well as the variations in these models' ability to predict brain activity compared to prior linear encoding approaches, provides valuable insights for the research community.
2. The significant improvement in prediction accuracy with nonlinear encoding models over linear models offers a promising future research direction for understanding what information is truly being captured in the nonlinear mapping of representations to brain activity.

**Weaknesses:**

1. While the main research question aims to investigate the benefits of using nonlinear encoding models over linear models, the insights into this question are not clearly presented. There are several significant weaknesses in this work:
- Previous brain encoding studies predominantly use linear mapping rather than nonlinear models for the sake of interpretability. This is because stimulus representations are complex, and these representations are derived from nonlinear AI models, while brain recordings often have a low signal-to-noise ratio (SNR). Linear mapping is advantageous in this context as it can provide clearer insights into the brain's processing or the model's representation. Therefore, I encourage the authors to elaborate on their motivation for using nonlinear models beyond the goal of improved prediction accuracy. Additionally, a discussion on how they intend to address the interpretability challenges associated with nonlinear approaches would strengthen the manuscript.
- Given that the authors extract representations from complex nonlinear AI models, it remains unclear whether the improvement observed with the nonlinear encoding model is due to the richness of the stimulus representations or the added complexity from introducing extra hidden layers in the voxel-wise encoding model.
- I recommend that the authors disentangle these factors by comparing their nonlinear model to a linear model with increased complexity (e.g., with more parameters). This approach would help isolate the specific impact of nonlinearity on model performance.
2. It is well-known that both LLaMA and Whisper are language models, with LLaMA being text-based and Whisper being speech-based. However, both models contain some degree of semantic information. As a result, integrating the two representations through concatenation may introduce redundancy or lead to one modality dominating the other. This raises the question of what information is actually contributing to predicting brain activity in the multimodal integration scenario, making it unclear which features are truly driving the prediction.
- In fact, a recent study by Oota et al. (2024) demonstrates that text-based language models exhibit high alignment in language regions due to brain-relevant semantics, whereas the alignment of speech-based language models in these regions is primarily driven by low-level stimulus features, indicating a lack of brain-relevant semantics in speech models.
- Consequently, it remains unclear which features are actually driving the prediction of brain activity in the concatenated scenario. Furthermore, this ambiguity extends to the nonlinear encoding model, where the specific contributions of different features are not well understood.

3. Using the concatenation of text-based and speech-based representations for multimodal information processing is not considered an ideal approach in the deep learning field. Moreover, while the authors discuss multimodal information processing in the brain, the experimental setup involves subjects listening to stories, which essentially pertains to language comprehension. In language comprehension, the early sensory processing differs—visual for reading and auditory for listening—while the semantic processing remains similar for both modalities [Deniz et al. 2019, Oota et al. 2024, Chen et al. 2024].
- As a result, it is unclear what constitutes "multimodal encoding" in this context if the focus is on language comprehension in the brain. If the authors intend to discuss multimodal information, it would be more appropriate to refer to it as "fusion" rather than "integration."

Deniz et al. 2019, The representation of semantic information across human cerebral cortex during listening versus reading is invariant to stimulus modality

Oota et al. 2024, Speech language models lack important brain-relevant semantics

Chen et al. 2024, The cortical representation of language timescales is shared between reading and listening

4. Several sentences are overstated: "We provide novel evidence for nonlinear multimodal integration in motor, sensory,
and visual brain regions, supporting existing theories in neurolinguistics while revealing new insights into the neural basis of speech comprehension."
- The authors used representations from language models but did not interpret which features are driving brain activity across different regions of interest (ROIs). Therefore, the claim that the models capture relevant brain activity patterns can only be substantiated if the authors analyze what information within these representations is truly predicting brain activity.
- I recommend incorporating techniques such as representational similarity analysis or feature importance measures. These methods could help identify which aspects of the language model representations are most predictive of brain activity across different regions, offering deeper insights into the model-brain alignment.
5. The feature extraction methods used for the two language models are not comparable. In LLaMA, the stimuli were presented in a dynamically sized context window that expanded up to 512 tokens, whereas the Whisper model used a fixed window of 16 seconds. This discrepancy is not ideal for comparing the performance of both models. Oota et al. (2024) addressed this issue by using a context of 20 tokens for text-based models and windows of 16 to 64 seconds for speech-based models to ensure a fair comparison. Consequently, with a 16-second window, it is unclear how much linguistic information is captured in Whisper compared to 512 tokens in LLaMA.
- Additionally, the feature extraction process in LLaMA is not clearly explained. The meaning of the context being "grown until 512 tokens and then reset to a new context of 256 tokens" needs further clarification.
- I suggest that the authors conduct an ablation study or sensitivity analysis to evaluate how varying context window sizes impact each model’s performance. This analysis could offer valuable insights into the influence of context window size as a methodological factor and potentially inform future improvements.
6. There is a lack of discussion on how the findings of the current paper might influence future model design in AI, particularly in developing cognitively inspired multimodal architectures, or how these results could contribute to neuroscientific theories of language processing.

**Questions:**

1. Line 67: The statement "While some studies have begun exploring multimodal models that combine linguistic features with visual information (Tang et al., 2024; Scotti et al., 2024)" is inaccurate. In Tang et al. (2024), the authors used a multimodal model (BridgeTower, which is pretrained on vision-language data) to extract representations, but they did not combine features directly. Instead, they provided either text or movie input separately to the model to obtain the representations. Therefore, the claim that these studies "combine features" is incorrect. Additionally, the authors have overlooked relevant studies that perform this kind of analysis, where multimodal integration or fusion is explicitly explored. These studies should be cited to provide a more accurate and comprehensive overview of the field.

Oota et al. 2022, Visio-linguistic brain encoding

Wang et al. 2023, Incorporating natural language into vision models improves prediction and understanding of higher visual cortex

2. Line 75: We demonstrate a substantial improvement in prediction accuracy, achieving a 17.2% increase in mean correlation across the cortex compared to the prior linear encoding models.
- The authors claim that the improvement is due to the use of nonlinear encoding models. However, it remains unclear what the hidden layer is capturing. Could the authors clarify what the 256 hidden neurons are learning that contributes to the increase in predicted brain activity? Understanding the specific features or patterns that these hidden neurons encode would help explain the observed improvement.
3. Line 291: Our findings suggest that nonlinear models are wellsuited for capturing this complex interplay of distributed neural activity, providing a more accurate and insightful representation of the brain’s functional organization compared to linear models.
- The authors should provide a stronger foundation for how the current study explains the brain's functional organization using nonlinear models in comparison to linear models. This would involve detailing how nonlinear models offer insights into the brain's functional architecture that linear models cannot capture, and specifying which aspects of brain organization are better explained by the nonlinear approach.
4. The insights presented in Section 3.3 are not clear. Please refer to the weaknesses mentioned in point 3, which highlight the lack of clarity regarding which features in the concatenated multimodal representations are actually driving the prediction of brain activity, especially given the differences in feature extraction and processing between text-based and speech-based models. Addressing this point would help clarify the insights in Section 3.3.
5. The surprisingly high prediction performance in visual regions during a listening task, as shown in Figures 1 and 2, requires further clarification.

---

> ### Author Response · Authors · 2024-11-25
>
> We appreciate the reviewer’s thoughtful points about nonlinearity and model interpretability. We addressed each concern:
>
>
> **Responses to point #1.**
> * “Elaborate on their motivation for using nonlinear models beyond the goal of improved prediction accuracy”
>   * Our choice of nonlinear models stems from several key motivations beyond simply improved prediction accuracy. First, we hypothesized that nonlinear relationships better capture the complex interactions inherent in neurobiological processes underlying language comprehension. This hypothesis is supported by existing studies (Tuller et al 2011, McGettigan 2012). Second, we anticipated that nonlinear models would offer a more nuanced understanding of the spatiotemporal dynamics of brain activity during language processing, potentially revealing finer-grained patterns of functional organization. This expectation is further substantiated by our findings in Appendix A.9.4., which demonstrate improved spatiotemporal compartmentalization and clustering of brain function using nonlinear methods. These theoretical considerations and anticipated advantages in understanding the underlying neurobiological mechanisms guided our methodological choices. We have further elaborated on these motivations in the revised manuscript in the introduction (Lines 63-85).
>
> * “Discussion on how they intend to address the interpretability challenges associated with nonlinear approaches”
>   * We acknowledge the reviewer's concern regarding the interpretability of nonlinear models. While this study primarily focuses on establishing the need for future research to explore nonlinear and multimodal approaches—demonstrating their superior predictive capabilities—we recognize the importance of addressing interpretability challenges. Therefore, we have added a discussion in the revised manuscript (Lines 522-535) outlining potential avenues for enhancing the interpretability of these models. This includes exploring machine learning interpretability techniques. While a comprehensive investigation of interpretability may be beyond the scope of this initial work, we believe our findings highlight the crucial need for future research to prioritize the development and application of appropriate interpretability methods within this burgeoning field of nonlinear neuroimaging analysis. We plan to address these challenges more directly in our future work.
>
> * “whether the improvement observed with the nonlinear encoding model is due to the richness of the stimulus representations or the added complexity from introducing extra hidden layers in the voxel-wise encoding model”, “disentangle these factors by comparing their nonlinear model to a linear model with increased complexity (e.g., with more parameters)”.
>   * The improved performance observed with the nonlinear encoding model arises from a combination of factors : the richer stimulus representations afforded by our multimodal approach and the increased model expressivity gained through nonlinearity. Table 1 demonstrates that the multimodal approach consistently improves performance regardless of model type (linear or nonlinear), indicating the inherent value of the richer stimulus representations. Furthermore, the direct comparison between our MLP+PCA model and the MLLinear+PCA model (a linear model with increased complexity, equivalent to an MLP without activation functions) highlights the additional performance gain attributable to the introduction of nonlinearity. To further address the reviewer’s concern, we have included additional baseline models in Table 1, including MLP + All Voxels (demonstrating the effectiveness of PCA in preventing overfitting when using a nonlinear model). These additional comparisons provide further evidence supporting the independent contributions of both multimodal representation and model nonlinearity to the observed performance improvements.
>
> **References** :
>
> * **Tuller et al 2011** : Tuller, Betty, et al. "Nonlinear dynamics in speech perception." Nonlinear Dynamics in Human Behavior(2011): 135-150.
>
> * **McGettigan 2012** : McGettigan, Carolyn, et al. "Speech comprehension aided by multiple modalities: behavioural and neural interactions." Neuropsychologia 50.5 (2012): 762-776.

---

> ### Author Response · Authors · 2024-11-25
>
> (Continued from previous comment)
>
> **Responses to point #2 and #3 (concatenation of language and audio models)**
>
> * “concatenation may introduce redundancy or lead to one modality dominating the other”
>   * We appreciate the reviewer's insightful comment regarding potential redundancy and dominance issues arising from the concatenation of speech and text modalities. While concatenation might introduce some redundancy, we believe the inputs contain substantial amounts of complementary information as they derive from different sources (speech spectrograms (Whisper) versus text (LLaMA)). Indeed, recent work by Oota et al. (2024) supports this, demonstrating that unlike text-based language models, speech models reveal brain activity patterns in auditory regions that are not fully explained by low-level stimulus features, providing unique information beyond what language models alone capture. Furthermore, the data-driven nature of the MLP encoding model should allow it to effectively learn to leverage this complementary information and weigh the contribution of each modality appropriately, mitigating the potential negative impact of redundancy. We have added descriptions about these complementary aspects of the modalities and their role in brain encoding in the Introduction (Lines 77-81).
>
> * “... making it unclear which features are truly driving the prediction”
>   * The reviewer raises a valid concern regarding the difficulty of attributing specific predictive power to individual features within our multimodal nonlinear model. While our primary focus is on testing the effectiveness of nonlinear methods in integrating complementary information from text and audio, even in the presence of potential redundancy, we acknowledge the limitations of our approach in precisely identifying the contribution of each individual feature. We have addressed this limitation in the revised discussion of the manuscript (Lines 522-535), including a discussion of feature attribution as a significant area for future investigation. This section makes note of Oota et al. (2024)'s approach to interpreting unimodal (text/speecn) models. We also suggest potential approaches for future work to disentangle the effects of each modality within nonlinear multimodal encoders, such as SHAP and LIME. The interactions within our nonlinear model make precise feature attribution challenging, but we believe that future research using more advanced feature attribution techniques will be crucial to further elucidate the individual and combined effects of the text and audio modalities.
>
> * “Concatenation … for multimodal information processing is not considered an ideal approach in the deep learning field”
>   * We agree that concatenation is not the most sophisticated approach for integrating text and speech representations. More advanced methods like cross-attention, learned fusion strategies, or hierarchical fusion are common in deep learning. However, given the limited size of available fMRI datasets (20 hours/subject), we deliberately chose concatenation as even simple two-layer MLPs performed poorly due to overfitting (Appendix A.4). In fact, while not presented in the paper, we found that even concatenating representations from just one additional whisper layer (i.e. one LLAMA layer + two whisper layers) decreased performance. More sophisticated fusion methods would introduce significantly more parameters, which would be impractical given our current data constraints. We view our concatenation approach as an important first step that demonstrates the potential of multimodal encoding, while acknowledging that more advanced fusion strategies could be explored when larger language fMRI datasets become available.
>
> * “While the author discusses multimodal information processing in the brain, the experimental setup involves subjects listening to stories“,“It would be more appropriate to refer to it as ‘fusion’ rather than ‘integration”
>   * We thank the reviewer for this insightful comment regarding our terminology. You are correct that while our work explores multimodal information processing, the experimental design uses auditory stimuli (stories). The term "integration" may indeed overstate the level of complete merging of information within our experimental setup. We agree that "fusion" more accurately reflects the combination of modalities in our experimental design and have revised the manuscript to use "fusion" instead of "integration" throughout. This change clarifies the nature of our multimodal approach.
>
>
> **References** :
>
> * **Oota et al 2024** : Oota, Subba Reddy, et al. "Speech language models lack important brain-relevant semantics." arXiv preprint arXiv:2311.04664 (2023).

---

> ### Author Response · Authors · 2024-11-25
>
> (Continued from previous comment)
>
> **Response to point #4  (Overstating of sentences)**
>
> * “did not interpret which features are driving brain activity across different regions of interest (ROIs)”, “ only be substantiated if the authors analyze what information within these representations is truly predicting brain activity.”
>   * The reviewer correctly points out that our current analysis does not delve into the specific features driving brain activity within different ROIs. We acknowledge that a more granular analysis of feature contributions would strengthen our conclusions. Although identifying precisely which features drive activity in various ROIs would provide a more detailed understanding, this level of analysis is beyond the scope of this initial investigation, which prioritizes establishing the benefits of nonlinear multimodal fusion. Our claims regarding the benefits of adding modalities, as supported by the work of Oota et al. (2024), remain valid; however, we agree that future work should investigate the precise contribution of specific features to brain activity predictions. We have added a discussion of this limitation and suggested future avenues for investigation in the revised manuscript (Lines 522-535), including approaches such as those outlined by Oota et al. (2024) and other feature attribution methods. We have also revised the relevant statements in the manuscript to avoid any overstatement of our findings.
>
> **Responses to point #5 (Window size discrepancies between modalities)**
> * “This discrepancy is not ideal for comparing the performance of both models”
>   * We acknowledge the reviewer's concern about different window sizes between LLaMA and Whisper. However, our goal was not direct model comparison but rather combining features using optimal parameters for each modality. Following Antello et al. (2024)'s established methodology, we used their validated parameters, and our own experiments confirm 16 seconds as the optimal window size for Whisper (results added to Appendix A.6).
> * “Feature extraction process in LLaMA is not clearly explained”
>   * We have revised Section 2.2 (Lines 155-161) to provide a detailed explanation of LLaMA's dynamic context window strategy, describing how the window grows to 512 tokens before resetting to 256 tokens to balance computational efficiency and contextual coherence. We also reference Antonello et al. (2024)'s original implementation for additional technical details.
> * “conduct an ablation study or sensitivity analysis to evaluate how varying context window sizes impact each model’s performance”
>   * We have added a new analysis in Appendix A.5 showing how Whisper’s window size affects encoding performance.
>
> **Responses to point #6 (Future model designs in AI)**
> * “...discussion on how the findings of the current paper might influence future model designs in AI, particularly in developing cognitively inspired multimodal architectures, or how these results could contribute to the neuroscientific theories of language processing.”
>   * We are grateful for this insightful comment. We included relevant discussion in our manuscript  (Lines 509-521).
>
> **References** :
>
> * **Oota et al 2024** : Oota, Subba Reddy, et al. "Speech language models lack important brain-relevant semantics." arXiv preprint arXiv:2311.04664 (2023).
>
> * **Antello et al 2024** : Richard Antonello, Aditya Vaidya, and Alexander Huth. Scaling laws for language encoding models in fmri. Advances in Neural Information Processing Systems, 36, 2024.

---

> ### Author Response · Authors · 2024-11-25
>
> (Continued from previous comment)
>
>
>
> **Answers for Questions 1 (incorrect reference regarding multimodal encoding models)**
> * We apologize for the ambiguous phrasing. What we meant to convey was that Tang et al 2024 use features from a multimodal feature extractor while Scottia et al 2024 mixed the features from multimodal sources. For consistency, we will remove the reference to Tang et al 2024 and add the two references you suggested. We sincerely thank the reviewer for the references.
>
> **Answers for Question 2 (what the hidden neurons encode)**
> * We agree with the reviewer. We have included interpretations as limitations and future works into the conclusion. (Lines 522-535)
>
> **Answers for Questions 3 (nonlinear models offering insights into the brain’s functional architecture that linear models cannot capture)**
> * We acknowledge that providing a stronger foundation for the benefits of nonlinear models would be valuable. However, we view this work as an initial exploration of a new analytical direction (using measures such as RED (relative error difference)) that is improved by nonlinear encoding models. The enhanced performance and refined functional compartmentalization achieved by the nonlinear MLP encoder suggest it is better equipped to capture the complex, distributed nature of speech comprehension. Our future research will delve deeper into understanding the mechanisms driving these improvements by leveraging interpretability methods.
>
> **Answers for Questions 4**
> * As in Question 2, we agree with the reviewer and have written it in the future works in the conclusion.
>
> **Answers for Questions 5**
> * To be honest, we were initially surprised by this finding too. However, as we explain in Section 3.2.3, this finding aligns with the convergence-divergence-zone theory, which proposes that semantic information from multiple modalities is integrated across the cortex to create unified representations of meaning. The improved predictions specifically occur in areas at the visual cortex border (OFA, EBA, FFA), consistent with prior fMRI studies (Popham et al 2021) showing these regions respond similarly to both visual and linguistic stimuli with matching semantic content.
>
> **References** :
>
> * **Popham et al 2021** : Popham, Sara F., et al. "Visual and linguistic semantic representations are aligned at the border of human visual cortex." Nature neuroscience 24.11 (2021): 1628-1636.

---

> ### Comment · Reviewer_V4NE · 2024-11-27
>
> I thank the authors for clarifying several questions, such as the extraction of features from the LLaMA and Whisper models. Additionally, the authors have majorly improved their presentation. However, some major issues remain unresolved.
>
> **Does concatenation introduce redundancy or lead to one modality dominating the other?**
>
> * The authors provided an explanation that the data-driven nature of the MLP encoding model should allow it to effectively leverage complementary information and appropriately weigh the contribution of each modality. However, this assumption needs to be thoroughly tested using methods such as variance partitioning or residual analysis. While the authors consider the concatenation-based model a joint encoding model, using separate encoding models for each feature space would better explain the unique variance of each feature space as well as the shared variance through partitioning or residual methods. Without empirical evidence, it remains unclear whether both features contribute equally in non-linear models.
>
> **Comparison with Oota et al. (2024)**
> * The authors’ approach differs from Oota et al. (2024), who did not combine two feature spaces. Instead, Oota et al. regressed out low-level features from semantic representations and examined the impact on brain alignment before and after the removal of low-level features. This approach isolates the contributions of different feature types, providing a clearer understanding of their individual roles in brain alignment.
>
> **making it unclear which features are truly driving the prediction**
>
> * The authors mentioned several libraries, such as SHAP and LIME, to disentangle the effects of each modality within non-linear multimodal models. However, this represents a significant weakness in the current work. While interpretability is more straightforward with simpler linear models, many studies have demonstrated which stimuli drive voxel responses using voxel-wise encoding model weights. The use of non-linear encoding models, coupled with reliance on libraries like SHAP and LIME for interpretation, appears unconvincing.
> * For example, a banded ridge regression model provides separate bands for each feature, allowing for the prediction performance of each feature to be analyzed individually. In the case of non-linear models, the question arises: Is there a method akin to banded regression that can learn distinct bands even in non-linear spaces? Such an approach could provide valuable insights into the contribution of each feature to brain alignment in non-linear models. Addressing this gap would significantly strengthen the interpretability and applicability of non-linear models in this context.
>
> I am raising my score based on the improved clarity of the presentation and their efforts in addressing several questions. Overall, I still believe that the current version of the paper requires major revisions to provide clearer results and a more robust analysis of the contributions of feature spaces.

---

> ### Author Response · Authors · 2024-11-29
>
> We thank the reviewer for their insightful feedback and the updated assessment.
>
> **Regarding Feature Dominance and Model Interpretability**
>
> - We have **conducted comprehensive variance partitioning analysis** (Appendix A.7) which includes :
>     - Voxel-wise variance partition results showing individual feature contributions
>     - Voxel-wise plots highlighting the largest variance partition for each voxel
>     - ROI-wise Venn diagrams illustrating the distribution of variance explained across brain regions
> - Key findings from this analysis are the following :
>     - **Joint audio-semantic features dominate** across most cortical regions (approximately 65% of significantly predicted voxels), demonstrating that our multimodal model effectively utilizes information from both modalities
>     - Core language-processing regions (AC, Broca's area, sPMv) show particularly strong attribution to joint representation (80-90% of voxels)
>     - Early AC and M1M areas show unique contributions from auditory features, consistent with early AC's role in processing low-level acoustic information and with our results in Figure 1 showing improved M1M predictions with audio features
> - While this analysis doesn't address all interpretability concerns, it provides strong evidence that **concatenation leads to genuine multimodal fusion** rather than single-modality dominance, supporting our claim of cortex-wide benefits from utilizing both audio and semantic features.
>
>
> **Comparison with Oota et al. (2024)**
>
> - We appreciate the clarification regarding Oota et al.'s approach. As mentioned in lines 525-529, we envision a multi-faceted approach to interpretability: multimodal-level analysis to understand feature integration (as demonstrated by our variance partitioning) and unimodal-level analysis to examine specific feature contributions (as in Oota et al. 2024).
>
> **Additional Updates**
>
> - Following another reviewer's feedback, we've added comparison with the (state-of-the-art) stacked regression baseline (Appendix A.3). Our multimodal MLP achieves a 14.4% improvement in $CC_{norm}$ and 7.7% improvement in average r² on the single test story ("wheretheressmoke").
> - While model interpretability remains crucial for neuroscientific research, we believe improved predictive performance itself has merit as it enables better in-silico experimentation, an emerging experimental paradigm for language neuroscience (Jain et al (2024)).
>
>
> **References**
> *  **Oota et al 2024** : Oota, Subba Reddy, et al. "Speech language models lack important brain-relevant semantics." arXiv preprint arXiv:2311.04664 (2023).
> * **Jain et al 2024** : Jain, Shailee, et al. "Computational language modeling and the promise of in silico experimentation." Neurobiology of Language 5.1 (2024): 80-106.
>
>
> *Note: We have updated our previous comments to reflect the revised appendix numbering in the updated manuscript.*

---

### Official Review · Reviewer_2Ry9 · 2024-11-08

**Soundness:** 2
**Presentation:** 4
**Contribution:** 2
**Rating:** 3
**Confidence:** 4

**Summary:**

This is an interesting paper, using embeddings from open source LLMs and Audio model compared to fMRI images of the human brain. The authors use 10 open fMRI datasets, from 3 subjects who listen to 20 hours of a podcast. They extract LLM embeddings from LLama3 and audio embeddings from Whisper. They carry out PCA on the fMRI to reduce the dimensionality of the space and then compare predictive models with the embeddings as inputs to predict the fMRI/PCA outputs.  The main claim of the study is that the combination of multimodality (LLM + audio embeddings) and nonlinearity (ML architecture comparisons) leads to a 17% increase in the accuracy of the predictions. The authors interpret this through the lens of cognitive science, and therefore " highlights the crucial role of nonlinearity and multimodality in accurately modeling the 491 brain’s complex processes during speech comprehension", and point to a set of theories in neurolinguistics.

**Strengths:**

The strength of the paper is the novelty of the question and the thoroughness of the analysis. The authors make use of open source ML models to create novel representation of input signals to the brain and fearlessly apply these to models predicting fMRI signals. There is precedence on this in the literature, both from ECOG data and fMRI data, but nonetheless taking this on is a highly nontrivial undertaking given the noise in the data and requires courage.

**Weaknesses:**

Unfortunately despite the claimed statistical significance Fig 1/2, I just don't believe the result. I grant the author that the changes in \Delta r might pass statistical tests, but these are extremely noisy measurements, the changes in the statistical indicator (r) are quite small and the error bars are enormous.  "Extraordinary claims require extraordinary evidence" and I just don't think we should be accepting papers that claim nonlinear processing in the brain without more evidence than this. Risk of overfitting here is very large and there are many choices the authors made that could have influenced their results unwittingly

I realize this is an impossible weakness to rebut -- dataset aquisition is expensive, open source datasets are few, and fMRI is noisy.  I appreciate the spirit of the analysis and would love to see this done at a larger scale to get more solid results -- but without better data I just don't think analysis llike this is warrented.

**Questions:**

If there were ways of rebutting this I'd be happy to listen but again given the nature of the results and the limitations in the data and the noisiness of the measurements I'm not sure what to ask. Are there more datasets to use or ways of strengthining evidence?

---

> ### Author Response · Authors · 2024-11-25
>
> We thank the reviewer for their evaluation of our work. We address their key concerns:
>
>
> ### **On Statistical Significance and Noisy Measurements**
> While we acknowledge the inherently noisy nature of fMRI measurements and small $\Delta r$ improvements, we argue our findings are meaningful and within the norms of previous research in this area. Here are the arguments :
>
> **(1) Regarding Large Error bars**
> * We wanted to make sure our Figure 1(e) was interpreted correctly. We just noticed that the reviewer mentioned "error bars", however, there are **no error bars in this figure as it is a boxplot**. We gently remind the reviewer that what appear to be **`"error bars" are actually boxplot whiskers`**, which extend to the most extreme values within 1.5 times the interquartile range (IQR). Assuming normal distribution, this corresponds to ±2.7σ (0.35th to 99.65th percentile).
> * We will revise the figure caption in future versions to clearly explain these are boxplots showing the distribution of improvements across voxels, where boxes indicate the interquartile range and whiskers extend to most extreme non-outlier values.
>
>
> **(2) Regarding statistical significance**
> * **Improvements are statistically significant (p < 0.05)**, following standard fMRI analysis practice in this field. (Oota 2022, King 2023)
> * Though not mentioned, we also used Benjamini-Hochberg (BH) correction. Also, **most of the ROI's p-values were much smaller than 0.05**. We will add the actual p-values to the Appendix in future versions.
>
>
> **(3) Regarding small $\Delta r$ improvements**
> * First, analyzing and deriving conclusions from **small $\Delta r$ improvements** are **common in language fMRI encoding field**, both using either ROI-wise and/or voxel-wise improvements.
> * Second, **our $\Delta r$ is actually quite large** when compared to previous research in this field.
> * Here we **list some papers** that analyze $\Delta r$ (both ROI-wise and voxel-wise), along **with their $\Delta r$ ranges, and compare them to ours**. We will divide them into two categories for clarity :
> * **(A) ROI-wise $\Delta r$ in other studies vs. ours**
>     * **Other studies**
>         * Figure 2(f) of King 2023: $\Delta r$ ranges -0.005 to 0.015
>         * Figure 4 of Lamarre 2022: $\Delta r$ values of approximately 0.025, 0.035, and 0.050 for AC, Broca, sPMv ROIs. Note: Lamarre 2022 did not perform ROI-wise statistical significance testing.
>         * Figure 3(D) of King 2021: ROI-wise $\Delta r$ in ranges of 0 to 0.015
>     * **Our study, by comparison** :
>         * Our ROI-wise $\Delta r$ plots in Figure 1(e) generally show **larger improvements** (AC : 0.06, Broca : 0.025\~0.050, IFSFP : 0.050\~0.075, and so on)
>         * While **$\Delta r$ in Figure 2** is smaller (FEF: 0.0125, M1M: 0.020), it remains **comparable to previous studies**
>         * **Subject-wise improvements** in Appendix A.11 and A.12.3 show **more pronounced effects** than main text figures (some ROIs with $\Delta r$ > 0.100)
> * **(B) Voxel-wise $\Delta r$ in other studies vs. ours**
>     * **Other studies**
>        * Figure 4 of Loong 2023: range of -0.2~0.2
>        * Figure 3 of Jain 2018: range of -0.2~0.2
>        * King 2021: ranges from -0.008\~0.008 to -0.06\~0.06
>        * King 2023: "forecast score" ranges 0.004~0.020 and relative gains of 0 to 5%
>     * **Our study, in comparison**
>         * All voxel-wise $\Delta r$ plots use range of -0.5~0.5 with visible improvements
>         * 17.2% increase in average $r^2$ compared to semantic-only linear models represents substantial improvement
> * In all the papers we mentioned above, ROI-wise/voxel-wise **$\Delta r$, however small, played a pivotal role** in deriving their conclusions. Given that **our $\Delta r$ are more pronouned**, we believe **our research is grounded**, notwithstanding passing statistical significance test.
>
> **(4) Other**
> * Our claims are based not solely on ROI-wise improvements, but **other analysis** too : **(1) cortex-wide voxel-wise improvements, (2) average voxel-wise $r^2$ improvements** (17.2% improvement compared to semantic linear models, 14.4%, 7.7% improvement compared to state-of-the-art.) **(3) Variance partitioning analysis**, and **(4) Better spatio-temporal compartmentalization** of brain function.
> * Our method **adopts pre-existing methodologies and performance metrics** from many established works regarding fMRI encoding models (Antonello 2024, Heer 2017, Oota 2024, Jain 2018, Jain 2024, Lamarre 2022, King 2021, King 2023)

---

> ### Author Response · Authors · 2024-12-01
>
> (continued)
>
> ### **Responses to Other Comments**
>
> **On Overfitting Concerns**:
>
> We employed various methods so no overfitting or data leaked occurred during training:
> * Tested on completely independent fMRI test sessions (stories) never seen during training/validation
> * We computed the PCA weight matrix only using training data.
> * Used cross-validation during training
> * We would greatly appreciate if the reviewer specified the type of overfitting that is concerning.
>
>
> **On Dataset Scale** :
>
> We share the reviewer's perspective that larger-scale studies would be valuable. We believe that this is the very reason our research is needed. In vision encoding/decoding, the introduction of large datasets like the Natural Scenes Dataset (NSD) (Allen 2022) with long (40+ hours with ~90,000 time points per subject) and high resolution MRI scans (7T) has enabled the transformation of fMRI vision encoding/decoding models, enabling nonlinear/cross-subject/multimodal encoding/decoding models. However, speech encoding represents an arguably more complex challenge than vision - requiring prediction across the entire cortex and engaging higher-level semantic processing beyond perceptual features. Despite this scientific significance, the field lacks datasets of comparable scale to NSD, preventing a similar transformation in language encoding models. Our work, using the largest available language fMRI dataset (20+ hours of audio per subject, ~33,000 timepoints), demonstrates both the potential and current limitations of this transformation: while we show promising results with simple nonlinear models, more complex architectures (two-layer MLPs, RNNs, or Transformers) performed poorly due to data constraints. We view this as empirical justification for developing larger and more comprehensive language fMRI datasets.
>
> **Regarding the reviewer's concern about "many choices that could have influenced the results" :**
>
> We would welcome more specific feedback about which methodological choices warrant further examination or justification.
>
> We appreciate the constructive feedback and view our work as an important step towards understanding nonlinear and multimodal processing in the brain.
>
> **Reference** :
>
> *	**Antonello 2024** : Richard Antonello, Aditya Vaidya, and Alexander Huth. Scaling laws for language encoding models in fmri. Advances in Neural Information Processing Systems, 36, 2024.
> *	**Heer 2017** : Wendy A de Heer, Alexander G Huth, Thomas L Griffiths, Jack L Gallant, and Fr´ed´eric E Theunissen. The hierarchical cortical organization of human speech processing. Journal of Neuroscience, 37(27):6539–6557, 2017.
> *	**Allen 2022** : Allen, Emily J., et al. "A massive 7T fMRI dataset to bridge cognitive neuroscience and artificial intelligence." Nature neuroscience 25.1 (2022): 116-126.
> Note: We have updated our previous comments to reflect the revised appendix numbering in the updated manuscript.
> *	**Jain 2024** : Jain, Shailee, et al. "Computational language modeling and the promise of in silico experimentation." Neurobiology of Language 5.1 (2024): 80-106.
> *	**Jain 2018** : Jain, Shailee, and Alexander Huth. "Incorporating context into language encoding models for fMRI." Advances in neural information processing systems 31 (2018).
> * **Oota 2024** : Oota, Subba Reddy, et al. "Speech language models lack important brain-relevant semantics." arXiv preprint arXiv:2311.04664 (2023).
> * **Lamarre 2022** : Lamarre, Mathis, Catherine Chen, and Fatma Deniz. "Attention weights accurately predict language representations in the brain." bioRxiv (2022): 2022-12.
> * **Oota 2022** : Oota, Subba Reddy, et al. "Neural language taskonomy: Which NLP tasks are the most predictive of fMRI brain activity?." arXiv preprint arXiv:2205.01404 (2022).
> * **Loong 2023** : Aw, Khai Loong, and Mariya Toneva. "Training language models to summarize narratives improves brain alignment." arXiv preprint arXiv:2212.10898 (2022).
> * **King 2023** : Caucheteux, Charlotte, Alexandre Gramfort, and Jean-Rémi King. "Evidence of a predictive coding hierarchy in the human brain listening to speech." Nature human behaviour 7.3 (2023): 430-441.
> * **King 2021** : Millet, Juliette, and Jean-Remi King. "Inductive biases, pretraining and fine-tuning jointly account for brain responses to speech." arXiv preprint arXiv:2103.01032 (2021).

---

> ### Author Response · Authors · 2024-12-01
>
> We noticed our response to your thoughtful review has been pending for quite a while.
>
> We've significantly updated our response to address your concerns, particularly regarding statistical significance and measurement reliability.
>
> We would greatly appreciate your feedback when you have a chance to review the updates.
>
> Please let us know if you need any clarification on our response.
>
> Best,
>
> The Authors

---

### Author Response · Authors · 2024-11-30
**Overview of changes made**

We sincerely thank all reviewers for their thoughtful feedback that has helped us significantly strengthen our paper. We have made several major revisions across two rounds:

### 1. First Revision
**1. Statistical Analysis Clarifications**
* Addressing Reviewer 2Ry9's concerns about statistical interpretation :
    * Clarified through comment that figures show boxplots with whiskers (not error bars), extending to extreme values ($\pm$1.5×IQR from Q1 and Q3)
    * Referenced many papers in comment showing that our $\Delta r$ are on par or larger than other studies that use $\Delta r$ to draw conclusions

**2. Comprehensive Model Analysis**
* Addressing Reviewer V4NE :
    * Added results of the encoding performance from using different Whisper window sizes (New Appendix A.6)
* Addressing Reviewer 6SyF :
    * Added results of MLP + all voxels condition to Table 1, demonstrating PCA's importance for preventing overfitting

**3. Enhanced Visualizations**
* Addressing Reviewer 6SyF :
    * Added both voxelwise and ROI-wise absolute prediction performance visualizations (New Appendix A.8)
    * Added CCnorm visualizations (New Appendix A.10.2) better highlighting improvements across brain regions.

**4. Methodology and Discussion Improvements**
* Addressing Reviewer V4NE :
    * Expanded nonlinear modeling motivations (Lines 63-85)
    * Added description about the complemtary information features audio and semantic models capture (Oota et al 2024) (Lines 77- 81)
    * Revised terminology from "multimodal integration" to "multimodal fusion"
    * Clarified LLaMA context window strategy during feature extraction (lines 155-164)
    * Added interpretability discussion (Lines 522-535) and future directions, regarding multimodal LLMs (Lines 510-521)
    * Clarified the referenceing regarding multimodal encoding models, and included two more references suggested by the reviewer (Lines 74-76)
* Addressing Reviewer 6SyF
    * Clarified description on LLaMA scaling behavior (lines 170-173, Appendix A.5)
    * Added explicit description of r² computation method (|r|*r) description

---

> ### Author Response · Authors · 2024-12-03
> **(continued) Overview of changes made**
>
> ### 2. Second Revision
>
> **1. Stacked Regression**
> * Addressing Reviewer 6SyF :
>     * Added detailed comparison with state-of-the-art (Antonello et al., 2024) stacked regression baseline (New Appendix A.3)
>         * Our model achieves 14.4% improvement in CCnorm and 7.7% improvement in average r² on single-story evaluation
>         * Result also show simple linear model beating stacked regression, bolstering our claim that allowing the interaction between the two modalities is critical
>         * Demonstrated clear advantages of direct modality interaction:
>             * Even linear concatenation outperforms stacked regression (SR) by 4.5%, showcasing benefits of allowing direct feature interaction
>             * Performance hierarchy (MLP > Linear > SR) suggests both architectural choices - direct fusion and nonlinearity - contribute independently
>             * Our models learn effective feature selection implicitly, performing better without explicit validation-based masking
>         * Included analysis with/without voxel selection for fair comparison
>
> **2. Variance Partitioning Analysis**
> * Addressing Reviewer V4NE's concerns about feature attribution :
>     * Added extensive variance partitioning analysis (New Appendix A.7) showing :
>         * Joint audio-semantic features dominate (~65% of significant voxels)
>         * Core language regions show strong attribution to joint representation (80-90%)
>         * Validates that our multimodal approach achieves genuine feature fusion rather than single-modality dominance
>
> ### Current Status:
> * Reviewer 2Ry9: We look forward to engaging in a constructive discussion regarding our detailed explanations, particularly on our methodology and the statistical significance of our findings, to ensure clarity and mutual understanding.
> * Reviewer V4NE: Following our first revision, raised concerns about potential single-modality dominance in concatenated features and interpretability aspects of nonlinear models. We addressed these concerns in our second revision through comprehensive variance partitioning analysis (Appendix A.7), demonstrating that joint audio-semantic features dominate (~65% of voxels) and core language regions show strong joint representation (80-90%). We look forward to their evaluation of this additional analysis and are happy to provide any further clarifications if needed.
> * Reviewer 6SyF: Has expressed satisfaction with the revisions we have made, which we deeply appreciate.
>
> These revisions maintain our core contribution while providing substantially stronger empirical support, particularly through the new variance partitioning analysis and comparisons with state-of-the-art models added in our second revision.
>
> We are again grateful to the reviewers for their insightful suggestions that have helped greatly improve both the clarity and comprehensiveness of our work. We are happy to provide any additional clarifications if needed.
>
> Best regards,
>
>
> The Authors

---

### Meta-Review · Area_Chair_QoTE · 2024-12-23

**Metareview:**

This paper investigates the use of nonlinear and linear embeddings from open source LLMs and audio models to predict the PCA components of BOLD responses as measured on fMRI voxel images of the human brain, and addresses the question whether linear or nonlinear mappings from language model representations could capture the complexity of nonlinear auditory signals and linguistic processing in the brain activity. The authors also create encoding models by combining the two representations and measure brain alignment. Their experimental results demonstrate a significant improvement in prediction accuracy with non-linear encoding models, achieving a 17.2% increase in mean correlation across the cortex compared to the prior linear encoding models, on the fMRI of 3 subjects listening to 20 hours of text. Results are really extensively analysed (the paper is 52 pages long, with a long appendix).

Strengths:
Exploration of nonlinear encoding models, thoroughness of the evaluation with very extensive experiments, scientific significance for the understanding of nonlinear relationships in the human brain.

Weaknesses:

The scores were very divided (3, 5, 8). Reviewer (2Ry9) who gave a score of 3 did no longer engage with the review after rebuttals, arguing that "despite the claimed statistical significance Fig 1/2, I just don't believe the result [...] Risk of overfitting here is very large and there are many choices the authors made that could have influenced their results unwittingly [...] I realize this is an impossible weakness to rebut"
Reviewer V4NE updated their score to 5 after discussion but still believed that a major rewrite was necessary. This paper is an edge case  but it seems it would help to revise and resubmit this at a different venue, and I wish the authors best of luck.

**Additional Comments On Reviewer Discussion:**

Two reviewers asked for and obtained very extensive revisions. The remaining questions from V4NE are about the lack of clarity in the presentation of results, and a request for more robust analysis of the contributions of text and speech-based LLM features in the feature space.

---

### Decision · Program_Chairs · 2025-01-22

Reject